# Ultrahyperbolic Neural Networks

**Marc T. Law**

NVIDIA

## Abstract

Riemannian space forms, such as the Euclidean space, sphere and hyperbolic space, are popular and powerful representation spaces in machine learning. For instance, hyperbolic geometry is appropriate to represent graphs without cycles and has been used to extend Graph Neural Networks. Recently, some pseudo-Riemannian space forms that generalize both hyperbolic and spherical geometries have been exploited to learn a specific type of nonparametric embedding called ultrahyperbolic. The lack of geodesic between every pair of ultrahyperbolic points makes the task of learning parametric models (e.g., neural networks) difficult. This paper introduces a method to learn parametric models in ultrahyperbolic space. We experimentally show the relevance of our approach in the tasks of graph and node classification.

## 1 Introduction

Riemannian manifolds of constant curvature are the most common representation spaces in machine learning. They include the Euclidean space (of constant zero curvature), the $d$-sphere (of constant positive curvature) and the hyperbolic space (of constant negative curvature). The choice of a geometry to represent data mainly depends on the kind of relationship that needs to be described. For instance, Gromov [10] showed the relevance of hyperbolic geometry to represent trees (i.e., graphs without cycles). Since many hierarchies can be described as trees, hyperbolic representations have been used to represent hierarchical relationships (e.g., hypernymy between words [19]). Nonetheless, in many domains (e.g., social networks or protein structures), hierarchical graphs contain cycles.

In hyperbolic geometry, the considered manifold is not a vector space and is not equipped with the standard dot product. Therefore, most hyperbolic neural networks [5, 8, 18, 27] represent the weights of their last layer in the tangent space of some reference point. That tangent space is equipped with a positive definite metric tensor and the learned model can then be optimized with Riemannian gradient descent [1, 4]. In particular, since there exists a geodesic between any pair of points, the parameters are often optimized by using parallel transport (also called parallel translation) or the logarithm map. The Riemannian gradients are then parallel translated to the reference tangent space in which the model parameters lie. We refer the reader to [23] for a recent survey on hyperbolic neural networks.

Recently, Law & Stam [15] proposed *ultrahyperbolic* embeddings. They are a type of embedding that lies on a pseudo-Riemannian manifold of constant nonzero curvature [2, 21, 30]. Pseudo-Riemannian manifolds (also called semi-Riemannian manifolds) are generalizations of Riemannian manifolds where the nondegenerate metric tensor is not constrained to be positive definite [16]. In particular, when the metric tensor is not positive definite (e.g., when it is indefinite), the negative of the (pseudo-Riemannian) gradient is not a descent direction [9]. Law & Stam [15] proposed an efficient method to calculate a descent direction and learn ultrahyperbolic (nonparametric) embeddings. The main motivation of representing data on an ultrahyperbolic manifold is that it contains hyperbolic and spherical parts (see Fig. 1 and supp. material for details). It can then describe relationships specific to hyperbolic and spherical geometries (e.g., to represent parts of a graph that are trees or cycles) and is more flexible. Ultrahyperbolic embeddings were experimentally shown to be more appropriate than hyperbolic embeddings to represent hierarchical graphs with cycles on several datasets [15].

35th Conference on Neural Information Processing Systems (NeurIPS 2021).

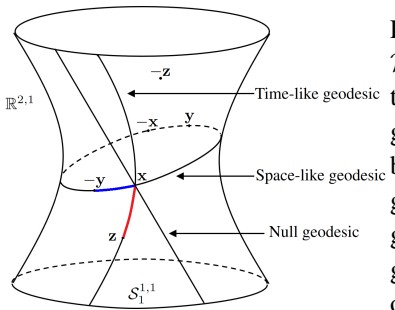

Figure 1: Geodesics of the pseudo-Riemannian quotient manifold $\mathcal{P}_1^{1,1} = \mathcal{S}_1^{1,1}/\pm 1$ embedded in $\mathbb{R}^{2,1}$. The point $[\mathbf{x}]$ of $\mathcal{P}_1^{1,1}$ is the pair $\{\mathbf{x}, -\mathbf{x}\}$. Any pair of points of $\mathcal{P}_1^{1,1}$ can be joined by a geodesic of $\mathcal{P}_1^{1,1}$. On the other hand, $\mathbf{x}$ and $-\mathbf{z}$ cannot be joined by an (unbroken) geodesic of $\mathcal{S}_1^{1,1}$. The length of the minimizing geodesic of $\mathcal{P}_1^{1,1}$ joining $[\mathbf{x}]$ and $[\mathbf{y}]$ is the length of the minimizing geodesic of $\mathcal{S}_1^{1,1}$ joining $\mathbf{x}$ and $-\mathbf{y}$ (in blue). The length of the geodesic of $\mathcal{P}_1^{1,1}$ joining $[\mathbf{x}]$ and $[\mathbf{z}]$ is the length of the geodesic of $\mathcal{S}_1^{1,1}$ joining $\mathbf{x}$ and $\mathbf{z}$ (in red). See details in the supp. material.

However, since there exist pairs of points on the ultrahyperbolic manifold considered in [15] that cannot be joined by an (unbroken) geodesic, gradients might not be parallel translated via a geodesic and the logarithm map joining two given points might not be defined. Directly extending hyperbolic neural networks [5, 8, 18, 27] to ultrahyperbolic space is then problematic.

In this paper, we propose a method to learn *ultrahyperbolic* representations with neural networks. Unlike [15], we consider the pseudo-Riemannian quotient manifold defined such that every point $\mathbf{x} = (x_0, \ldots, x_d)^\top$ is equivalent to its antipodal point $-\mathbf{x} = (-x_0, \ldots, -x_d)^\top$. In this way, for any other point $\mathbf{y}$, there always exists at least one geodesic joining $(\mathbf{x}, \mathbf{y})$ or $(-\mathbf{x}, \mathbf{y})$. We provide sufficient conditions to minimize a function defined on our quotient manifold. Since tangent vectors (hence gradients) of quotient manifolds are abstract objects, we explain how the function can be optimized with the *horizontal lift* operator. Our optimization framework is general, so we also introduce an extension to Graph Neural Networks (GNNs) [18] such that the activation representations at each layer of our GNN lie in ultrahyperbolic space. We then obtain a deep ultrahyperbolic model to represent graphs given as input. We evaluate our approach in different graph classification tasks.

## 2 Pseudo-sphere and Quotient Manifold

We extend the ultrahyperbolic manifold described in [15] (denoted by $\mathcal{S}_r^{p,q}$) to a quotient manifold denoted by $\mathcal{P}_r^{p,q}$ where $(p, q)$ is the metric signature (see page 343 of [16]) of the pseudo-Riemannian manifold and $1/r^2$ is its curvature. The motivation is that any pair of points of $\mathcal{P}_r^{p,q}$ can be joined by at least one geodesic, which allows us to optimize parametric models. We consider three pseudo-Riemannian manifolds $\mathcal{P}_r^{p,q} \subset \mathcal{S}_r^{p,q} \subset \mathbb{R}^{p+1,q}$ that we define below. We explain how $\mathcal{P}_r^{p,q}$ generalizes elliptic and hyperbolic geometries in the special cases where $q = 0$ and $p = 0$, respectively.

**Notation.** We denote points on a smooth manifold $\mathcal{M}$ [16] by boldface Roman characters $\mathbf{x} \in \mathcal{M}$. $[\mathbf{x}] := \{\mathbf{x}, -\mathbf{x}\}$ is a pair of antipodal points. $T_\mathbf{x}\mathcal{M}$ is the tangent space of $\mathcal{M}$ at $\mathbf{x}$ and we write tangent vectors $\boldsymbol{\xi} \in T_\mathbf{x}\mathcal{M}$ in boldface Greek fonts. $\mathbb{R}^d$ is the $d$-dimensional Euclidean space equipped with the (positive definite) dot product $\langle \cdot, \cdot \rangle$ defined as $\langle \mathbf{x}, \mathbf{y} \rangle := \mathbf{x}^\top \mathbf{y}$. $\mathbf{I}$ is the identity matrix. The inverse function of the cosine (resp. hyperbolic cosine) is denoted by $\cos^{-1}$ (resp. $\cosh^{-1}$).

**Ambient space $\mathbb{R}^{p+1,q}$.** Our ambient space $\mathbb{R}^{p+1,q}$ is a vector space of dimensionality $d + 1 = p + q + 1 \in \mathbb{N}$ called *pseudo-Euclidean space* [21]. It is equipped with the following scalar product (i.e., nondegenerate symmetric bilinear form) of signature $(p + 1, q)$:

$$\forall \mathbf{x} = (x_0, \ldots, x_d)^\top, \ \mathbf{y} = (y_0, \ldots, y_d)^\top, \ \langle \mathbf{x}, \mathbf{y} \rangle_q := \sum_{i=0}^{p} x_i y_i - \sum_{j=p+1}^{d} x_j y_j = \mathbf{x}^\top \mathbf{G} \mathbf{y}, \quad (1)$$

where the signature matrix $\mathbf{G} = \mathbf{G}^{-1} = \mathbf{I}_{p+1,q}$ is the $(d + 1) \times (d + 1)$ diagonal matrix with the first $p + 1$ diagonal elements equal to 1 and the remaining $q$ equal to $-1$. Following general relativity literature and spacetime terminology [7], $\mathbb{R}^{p+1,q}$ has $p + 1$ space dimensions and $q$ time dimensions. Since it is a vector space, we can identify its tangent space to the space itself by means of the natural isomorphism $T_\mathbf{x}\mathbb{R}^{p+1,q} \approx \mathbb{R}^{p+1,q}$. Finally, the Euclidean space $\mathbb{R}^{d+1}$ is the special case of $\mathbb{R}^{d+1,0}$ which contains zero time dimension, and where $\mathbf{G} = \mathbf{I}_{d+1,0} = \mathbf{I}$.

**Total space $\mathcal{S}_r^{p,q}$.** Our total space $\mathcal{S}_r^{p,q}$ is a pseudo-sphere of radius $r > 0$ embedded in $\mathbb{R}^{p+1,q}$. It is the following hypersurface:

$$\mathcal{S}_r^{p,q} := \left\{ \mathbf{x} \in \mathbb{R}^{p+1,q} : \langle \mathbf{x}, \mathbf{x} \rangle_q = r^2 \right\}, \quad (2)$$

It is equivalent to work with the pseudo-hyperboloid $\mathcal{Q}_r^{q,p} := \{\mathbf{x} \in \mathbb{R}^{q,p+1} : \langle \mathbf{x}, \mathbf{x} \rangle_{p+1} = -r^2\}$ and the pseudo-sphere $\mathcal{S}_r^{p,q}$ as they are anti-isometric to each other (see supp. material). Moreover, the radius $r > 0$ plays a role of scaling factor so we consider it to be 1 although it can be learned [5, 14]. Finally, both $\mathbf{x} \in \mathcal{S}_r^{p,q}$ and its antipodal point $-\mathbf{x}$ lie on $\mathcal{S}_r^{p,q}$ since $\langle \mathbf{x}, \mathbf{x} \rangle_q = \langle -\mathbf{x}, -\mathbf{x} \rangle_q$.

**Quotient manifold $\mathcal{P}_r^{p,q}$.** We consider as equivalence relation the two-element group $\{\pm 1\}$ consisting of the identity map $\mathbf{x} \mapsto \mathbf{x}$ and the antipodal map $\mathbf{x} \mapsto -\mathbf{x}$. This means that two points $\mathbf{x} \in \mathcal{S}_r^{p,q}$ and $\mathbf{y} \in \mathcal{S}_r^{p,q}$ are equivalent iff $\mathbf{y} = \mathbf{x}$ or $\mathbf{y} = -\mathbf{x}$. We define the following projective space:

$$\mathcal{P}_r^{p,q} := \mathcal{S}_r^{p,q} / \pm 1 = \mathcal{S}_r^{p,q} / \pm \mathbf{I} = \{\{\mathbf{x}, -\mathbf{x}\} : \mathbf{x} \in \mathcal{S}_r^{p,q}\}. \tag{3}$$

Every point of $\mathcal{P}_r^{p,q}$ is an unordered pair that we denote by $[\mathbf{x}] := \{\mathbf{x}, -\mathbf{x}\}$. Since $\mathcal{P}_r^{p,q}$ is a projective space, every point $[\mathbf{x}] \in \mathcal{P}_r^{p,q}$ can be interpreted as the intersection of the pseudo-sphere $\mathcal{S}_r^{p,q}$ with a line passing through the origin of $\mathbb{R}^{p+1,q}$. In some cases, it might be easier to interpret points of $\mathcal{P}_r^{p,q}$ as lines through the origin, and to study their properties when they intersect the pseudo-sphere. Each point $[\mathbf{x}] \in \mathcal{P}_r^{p,q}$ is also a submanifold of $\mathcal{S}_r^{p,q}$ and a discrete space.

In the following, we explain how $\mathcal{P}_r^{p,q}$ extends spherical geometry to elliptic geometry (i.e., when $q = 0$), or naturally describes the hyperboloid model of hyperbolic geometry (i.e., when $p = 0$).

**Elliptic geometry ($q = 0$).** In spherical geometry, points lie on the unit $d$-sphere $\mathcal{S}^d := \mathcal{S}_1^{d,0} = \{\mathbf{x} \in \mathbb{R}^{d+1} : \langle \mathbf{x}, \mathbf{x} \rangle = 1\}$. The geometry of the projective $d$-space $\mathcal{P}^d := \mathcal{S}^d / \pm 1$ is called elliptic geometry [24, 30]. Geodesic distances of $\mathcal{P}^d$ naturally account for the fact that they compare sets. Let $\bar{\mathsf{d}}_{\overline{\gamma}} : \mathcal{S}^d \times \mathcal{S}^d \to \mathbb{R}$ be the geodesic distance of $\mathcal{S}^d$ (i.e., spherical distance). The geodesic distance between $[\mathbf{x}] \in \mathcal{P}^d$ and $[\mathbf{y}] \in \mathcal{P}^d$ is $\mathsf{d}_\gamma([\mathbf{x}], [\mathbf{y}]) = \min_{\mathbf{a} \in [\mathbf{x}], \mathbf{b} \in [\mathbf{y}]} \bar{\mathsf{d}}_{\overline{\gamma}}(\mathbf{a}, \mathbf{b})$. We then have:

$$\mathsf{d}_\gamma([\mathbf{x}], [\mathbf{y}]) := \min\{\bar{\mathsf{d}}_{\overline{\gamma}}(\mathbf{x}, \mathbf{y}), \bar{\mathsf{d}}_{\overline{\gamma}}(-\mathbf{x}, \mathbf{y})\} = \cos^{-1}(|\langle \mathbf{x}, \mathbf{y} \rangle|) = \cos^{-1}(|\langle \mathbf{x}, \mathbf{y} \rangle_q|), \tag{4}$$

which is a distance metric. The fact that the spherical geometry is antipodally symmetric (i.e., every point can be inverted w.r.t. the origin) leads to a duplication of geometric information [24]. Identifying each pair of antipodal points to one point eliminates the antipodal duplication in spherical geometry.

**The hyperboloid model of hyperbolic geometry** is similar to the geometry of $\mathcal{P}_1^{0,q}$ ($p = 0$). The $q$-dimensional manifold $\mathcal{S}_1^{0,q} \subset \mathbb{R}^{1,q}$ contains two separate sheets (i.e., two connected components) and is anti-isometric to the hyperboloid of two sheets $\mathcal{Q}_1^{q,0}$. Pairs of antipodal points lying on different sheets of $\mathcal{S}_1^{0,q}$ are considered as a single point of $\mathcal{P}_1^{0,q}$. Let $\mathbf{x} \in \mathcal{S}_1^{0,q}$ and $\mathbf{z} \in \mathcal{S}_1^{0,q}$ be two points lying on the same sheet of $\mathcal{S}_1^{0,q}$, there exists no geodesic joining $\mathbf{x}$ and $-\mathbf{z}$. Their geodesic distance with respect to $\mathcal{S}_1^{0,q}$ can then be considered to be $\bar{\mathsf{d}}_{\overline{\gamma}}(\mathbf{x}, -\mathbf{z}) = +\infty$, and we have:

$$\mathsf{d}_\gamma([\mathbf{x}], [\mathbf{z}]) := \min\{\bar{\mathsf{d}}_{\overline{\gamma}}(\mathbf{x}, \mathbf{z}), +\infty\} = \bar{\mathsf{d}}_{\overline{\gamma}}(\mathbf{x}, \mathbf{z}) = \cosh^{-1}(\langle \mathbf{x}, \mathbf{z} \rangle_q) = \cosh^{-1}(|\langle \mathbf{x}, \mathbf{z} \rangle_q|), \tag{5}$$

which is similar to the hyperbolic distance metric of the hyperboloid model studied in [20].

**Ultrahyperbolic geometry (or indefinite elliptic geometry).** In this paper, we propose a parametric model that learns representations lying on the quotient manifold $\mathcal{P}_r^{p,q}$. When both $p$ and $q$ are positive, the metric tensor of $\mathcal{P}_r^{p,q}$ is nondegenerate (see page 343 of [16]) and indefinite. This means that the manifold is pseudo-Riemannian but not Riemannian due to the lack of positive definiteness of the metric tensor. $\mathcal{P}_r^{p,q}$ is also called an *indefinite elliptic space* [30] in the literature. We refer the reader to Chapters 11 and 12 of [30] or Chapter 7 of [21] for details. As an example, Fig. 1 illustrates the manifold $\mathcal{P}_1^{1,1}$. Our main motivation for considering $\mathcal{P}_r^{p,q}$ is that it is more flexible than hyperbolic and elliptic geometries since it contains hyperbolic and elliptic parts (i.e., time-like and space-like geodesics in Fig. 1). This flexibility allows us to better represent graphs that are not entirely trees or cycles, but that contain tree-like or cycle subgraphs. We experimentally verify our intuition.

## 3 Optimization on Ultrahyperbolic Quotient Manifolds

Our ultrahyperbolic representations lie on the quotient manifold $\mathcal{P}_r^{p,q}$. In this section, we provide differential geometry tools to optimize some differentiable function $f : \mathcal{P}_r^{p,q} \to \mathbb{R}$. To this end, we need the formulation of geodesics of $\mathcal{P}_r^{p,q}$. In Section 3.1, we explain how to formulate tangent vectors of $\mathcal{P}_r^{p,q}$ as a function of tangent vectors of $\mathcal{S}_r^{p,q}$ via the *horizontal lift* operator. This operator allows us to formulate geodesics of $\mathcal{P}_r^{p,q}$ as a function of geodesics of $\mathcal{S}_r^{p,q}$ in Section 3.2. In Section 3.3, we state the properties that the function $f$ has to satisfy due to the quotient nature of $\mathcal{P}_r^{p,q}$. In Section 3.4, we illustrate how to optimize a standard neural network. Our deep GNN that maps activation representations in ultrahyperbolic space at each layer is introduced in Section 4.

## 3.1 Representing tangent vectors of $\mathcal{P}_r^{p,q}$ only by horizontal tangent vectors of $\mathcal{S}_r^{p,q}$

It can be difficult to work numerically with the tangent space $T_{[\mathbf{x}]}\mathcal{P}_r^{p,q}$ of $\mathcal{P}_r^{p,q}$ at $[\mathbf{x}]$ since $[\mathbf{x}] = \{\mathbf{x}, -\mathbf{x}\}$ is an equivalence class. We now present some differential geometry tools to define tangent vectors of $\mathcal{S}_r^{p,q}$ as a function of tangent vectors of $\mathcal{P}_r^{p,q}$, and vice versa. Their general definitions can be found in Chapter 7 of [21]. We also refer the reader to [4] for details on optimization on quotient manifolds. Our contribution in this subsection is that we give their formulation for $\mathcal{P}_r^{p,q}$. We first give the formulation of tangent spaces of $\mathcal{S}_r^{p,q}$ and then provide tools to identify tangent vectors of $\mathcal{P}_r^{p,q}$. These tools will be essential to construct geodesics of $\mathcal{P}_r^{p,q}$ and represent them via $\mathcal{S}_r^{p,q}$.

**The tangent space** $T_{\mathbf{x}}\mathcal{S}_r^{p,q}$ of $\mathcal{S}_r^{p,q}$ at $\mathbf{x}$ can be defined as: $T_{\mathbf{x}}\mathcal{S}_r^{p,q} := \{\boldsymbol{\xi} \in \mathbb{R}^{p+1,q} : \langle \boldsymbol{\xi}, \mathbf{x} \rangle_q = 0\}$.

**The canonical map (or natural map [21])** $\pi : \mathcal{S}_r^{p,q} \to \mathcal{P}_r^{p,q}$ is defined as: $\forall \mathbf{x} \in \mathcal{S}_r^{p,q}, \pi(\mathbf{x}) := [\mathbf{x}] = \{\mathbf{x}, -\mathbf{x}\}$. Its differential at $\mathbf{x}$ is denoted by $\mathrm{d}\pi_{\mathbf{x}} : T_{\mathbf{x}}\mathcal{S}_r^{p,q} \to T_{[\mathbf{x}]}\mathcal{P}_r^{p,q}$.

**The horizontal space** $\mathcal{H}_{\mathbf{x}}$ **and the vertical space** $\mathcal{V}_{\mathbf{x}}$ at $\mathbf{x} \in \mathcal{S}_r^{p,q}$ are subspaces of $T_{\mathbf{x}}\mathcal{S}_r^{p,q}$ defined such that $T_{\mathbf{x}}\mathcal{S}_r^{p,q} = \mathcal{H}_{\mathbf{x}} \oplus \mathcal{V}_{\mathbf{x}}$ is a direct sum of linear spaces, and $\mathcal{V}_{\mathbf{x}}$ is the following kernel: $\mathcal{V}_{\mathbf{x}} := \ker(\mathrm{d}\pi_{\mathbf{x}})$. From Proposition 5.38 of [16], we find $\ker(\mathrm{d}\pi_{\mathbf{x}}) = T_{\mathbf{x}}([\mathbf{x}]) = \mathbf{0}$ because $[\mathbf{x}]$ is a discrete space so $[\mathbf{x}]$ and its tangent spaces are 0-dimensional. We then have $\mathcal{H}_{\mathbf{x}} = T_{\mathbf{x}}\mathcal{S}_r^{p,q}$. Elements of $\mathcal{H}_{\mathbf{x}}$ are called horizontal vectors, and all the tangent vectors of $\mathcal{S}_r^{p,q}$ are horizontal.

**The horizontal lift** (see §29 of [29]) at $\mathbf{x} \in \mathcal{S}_r^{p,q}$ of the tangent vector $\boldsymbol{\xi} \in T_{[\mathbf{x}]}\mathcal{P}_r^{p,q}$ is the unique horizontal vector denoted by $\overline{\boldsymbol{\xi}}_{\mathbf{x}} = \mathrm{lift}_{\mathbf{x}}(\boldsymbol{\xi}) \in \mathcal{H}_{\mathbf{x}}$ such that $\mathrm{d}\pi_{\mathbf{x}}(\overline{\boldsymbol{\xi}}_{\mathbf{x}}) = \boldsymbol{\xi}$. Since $\mathcal{H}_{\mathbf{x}} = T_{\mathbf{x}}\mathcal{S}_r^{p,q}$, the $\mathrm{lift}_{\mathbf{x}}$ operator is bijective so tangent vectors in $T_{[\mathbf{x}]}\mathcal{P}_r^{p,q}$ can be equivalently represented by horizontal vectors in $\mathcal{H}_{\mathbf{x}}$. During optimization, we will exploit this bijection and consider only some specific horizontal space to represent and update the weights of our neural network. The fact that $\mathcal{H}_{\mathbf{x}} = T_{\mathbf{x}}\mathcal{S}_r^{p,q}$ is convenient since it implies that any tangent vector in $T_{\mathbf{x}}\mathcal{S}_r^{p,q}$ can be represented in $T_{[\mathbf{x}]}\mathcal{P}_r^{p,q}$. We can then construct a geodesic of $\mathcal{P}_r^{p,q}$ from any geodesic of $\mathcal{S}_r^{p,q}$ as discussed below.

## 3.2 Geodesic of $\mathcal{P}_r^{p,q}$, exponential map and parallel transport

To optimize over $\mathcal{S}_r^{p,q}$ and $\mathcal{Q}_r^{q,p}$, Gao *et al.* [9] and Law & Stam [15] define tools such as the geodesic, parallel transport, exponential map, logarithm map and the *geodesic distance* $\overline{\mathrm{d}}_{\overline{\gamma}} : \mathcal{S}_r^{p,q} \times \mathcal{S}_r^{p,q} \to \mathbb{R}$ (see formulations in the supp. material). Our contribution in this subsection is that we extend all of the above differential geometry tools to $\mathcal{P}_r^{p,q}$. Their details can be found in the supp. material.

**The geodesic** $\overline{\gamma}_{\mathbf{x} \to \overline{\boldsymbol{\xi}}_{\mathbf{x}}} : \mathbb{R} \to \mathcal{S}_r^{p,q}$ of $\mathcal{S}_r^{p,q}$ is the curve defined such that its initial point is $\overline{\gamma}_{\mathbf{x} \to \overline{\boldsymbol{\xi}}_{\mathbf{x}}}(0) = \mathbf{x} \in \mathcal{S}_r^{p,q}$, its initial velocity is $\overline{\gamma}'_{\mathbf{x} \to \overline{\boldsymbol{\xi}}_{\mathbf{x}}}(0) = \overline{\boldsymbol{\xi}}_{\mathbf{x}} \in T_{\mathbf{x}}\mathcal{S}_r^{p,q}$ and its acceleration is zero. When the initial conditions are clear from the context, we denote the geodesic by $\overline{\gamma}$ and ignore its indices. Since every geodesic $\overline{\gamma}$ of $\mathcal{S}_r^{p,q}$ satisfies $\forall t, \overline{\gamma}'(t) \in \mathcal{H}_{\overline{\gamma}(t)}$, it is called horizontal and $\gamma := \pi \circ \overline{\gamma} : \mathbb{R} \to \mathcal{P}_r^{p,q}$ is a geodesic of $\mathcal{P}_r^{p,q}$. By the chain rule, we have $\forall t, \gamma'(t) = \mathrm{d}\pi_{\overline{\gamma}(t)}(\overline{\gamma}'(t))$, which implies $\forall t, \mathrm{lift}_{\overline{\gamma}(t)}(\gamma'(t)) = \overline{\gamma}'(t)$. We then have $\forall t \in \mathbb{R}, \gamma_{[\mathbf{x}] \to \boldsymbol{\xi}}(t) = \{\overline{\gamma}_{\mathbf{x} \to \overline{\boldsymbol{\xi}}_{\mathbf{x}}}(t), \overline{\gamma}_{-\mathbf{x} \to \overline{\boldsymbol{\xi}}_{-\mathbf{x}}}(t)\}$, and we find $\overline{\boldsymbol{\xi}}_{\mathbf{x}} = -\overline{\boldsymbol{\xi}}_{-\mathbf{x}}$ to preserve the equivalence between antipodal points: $\overline{\gamma}_{\mathbf{x} \to \overline{\boldsymbol{\xi}}_{\mathbf{x}}}(t) = -\overline{\gamma}_{-\mathbf{x} \to \overline{\boldsymbol{\xi}}_{-\mathbf{x}}}(t)$.

**Exponential and logarithm map.** The exponential map of $\mathcal{P}_r^{p,q}$ at $[\mathbf{x}]$ is the differentiable mapping $\exp_{[\mathbf{x}]} : T_{[\mathbf{x}]}\mathcal{P}_r^{p,q} \to \mathcal{P}_r^{p,q}$ defined such that $\exp_{[\mathbf{x}]}(\boldsymbol{\xi}) := \gamma_{[\mathbf{x}] \to \boldsymbol{\xi}}(1) = \{\overline{\gamma}_{\mathbf{x} \to \overline{\boldsymbol{\xi}}_{\mathbf{x}}}(1), \overline{\gamma}_{-\mathbf{x} \to \overline{\boldsymbol{\xi}}_{-\mathbf{x}}}(1)\}$.

We denote the exponential map of $\mathcal{S}_r^{p,q}$ at $\mathbf{x}$ by $\overline{\exp}_{\mathbf{x}} : T_{\mathbf{x}}\mathcal{S}_r^{p,q} \to \mathcal{S}_r^{p,q}$. It is defined as $\overline{\exp}_{\mathbf{x}}(\overline{\boldsymbol{\xi}}_{\mathbf{x}}) := \overline{\gamma}_{\mathbf{x} \to \overline{\boldsymbol{\xi}}_{\mathbf{x}}}(1)$, and we have $\exp_{[\mathbf{x}]}(\boldsymbol{\xi}) = [\overline{\exp}_{\mathbf{x}}(\overline{\boldsymbol{\xi}}_{\mathbf{x}})]$. In practice, we select some reference point $\mathbf{x}$ and only work with the exponential map $\overline{\exp}_{\mathbf{x}}$. The logarithm map is the inverse function of the exponential map (i.e., $\log_{[\mathbf{x}]} := \exp_{[\mathbf{x}]}^{-1}$). Their exact formulation can be found in the supp. material.

**Parallel transport on $\mathcal{S}_r^{p,q}$.** Given the minimizing (unbroken) geodesic $\overline{\gamma}$ (i.e., minimizing the arc length) from $\mathbf{x} = \overline{\gamma}(0)$ to $\mathbf{y} = \overline{\gamma}(1)$, the parallel transport $P_{\mathbf{x} \curvearrowright \mathbf{y}}^{\overline{\gamma}} : T_{\mathbf{x}}\mathcal{S}_r^{p,q} \to T_{\mathbf{y}}\mathcal{S}_r^{p,q}$ is a linear isometry such that $\forall \overline{\boldsymbol{\xi}}_{\mathbf{x}}, \overline{\boldsymbol{\zeta}}_{\mathbf{x}}, \langle \overline{\boldsymbol{\xi}}_{\mathbf{x}}, \overline{\boldsymbol{\zeta}}_{\mathbf{x}} \rangle_q = \langle P_{\mathbf{x} \curvearrowright \mathbf{y}}^{\overline{\gamma}}(\overline{\boldsymbol{\xi}}_{\mathbf{x}}), P_{\mathbf{x} \curvearrowright \mathbf{y}}^{\overline{\gamma}}(\overline{\boldsymbol{\zeta}}_{\mathbf{x}}) \rangle_q$ (see page 66 of [21]). The parallel transport along $\overline{\gamma}$ from $\mathbf{x}$ to $\mathbf{y}$ (where $\mathbf{x}$ and $\mathbf{y}$ satisfy $\langle \mathbf{x}, \mathbf{y} \rangle_q > -r^2$) is:

$$P_{\mathbf{x} \curvearrowright \mathbf{y}}^{\overline{\gamma}}(\overline{\boldsymbol{\xi}}_{\mathbf{x}}) := \overline{\boldsymbol{\xi}}_{\mathbf{x}} - \frac{\langle \mathbf{y}, \overline{\boldsymbol{\xi}}_{\mathbf{x}} \rangle_q}{\langle \mathbf{x}, \mathbf{y} \rangle_q + r^2}(\mathbf{y} + \mathbf{x}) \tag{6}$$

**Minimizing geodesic of $\mathcal{P}_r^{p,q}$.** Our parallel transport on $\mathcal{P}_r^{p,q}$ depends on a minimizing geodesic $\gamma$ whose arc length (that we call *geodesic distance* $\mathsf{d}_\gamma$) from $[\mathbf{x}] = \gamma(0)$ to $[\mathbf{y}] = \gamma(1)$ is:

$$\forall [\mathbf{x}] \in \mathcal{P}_r^{p,q}, [\mathbf{y}] \in \mathcal{P}_r^{p,q}, \; \mathsf{d}_\gamma([\mathbf{x}], [\mathbf{y}]) = \begin{cases} r \cosh^{-1}(|\frac{\langle \mathbf{x}, \mathbf{y} \rangle_q}{r^2}|) & \text{if } |\frac{\langle \mathbf{x}, \mathbf{y} \rangle_q}{r^2}| \geq 1 \\ r \cos^{-1}(|\frac{\langle \mathbf{x}, \mathbf{y} \rangle_q}{r^2}|) & \text{otherwise.} \end{cases} \tag{7}$$

and we have $\overline{\mathsf{d}}_{\overline{\gamma}}(\mathbf{x}, \mathbf{y}) < \overline{\mathsf{d}}_{\overline{\gamma}}(-\mathbf{x}, \mathbf{y})$ iff $\langle \mathbf{x}, \mathbf{y} \rangle_q > 0$. See details in the supp. material.

**The parallel transport $P_{[\mathbf{x}] \frown [\mathbf{y}]}^\gamma$ on $\mathcal{P}_r^{p,q}$** can be horizontally lifted on $\mathcal{H}_\mathbf{y}$ as discussed above:

$$\forall \boldsymbol{\xi} \in T_{[\mathbf{x}]} \mathcal{P}_r^{p,q}, \; \text{lift}_\mathbf{y}(P_{[\mathbf{x}] \frown [\mathbf{y}]}^\gamma(\boldsymbol{\xi})) = \begin{cases} P_{\mathbf{x} \frown \mathbf{y}}^{\overline{\gamma}}(\overline{\boldsymbol{\xi}}_\mathbf{x}) & \text{if } \langle \mathbf{x}, \mathbf{y} \rangle_q > 0 \\ P_{-\mathbf{x} \frown \mathbf{y}}^{\overline{\gamma}}(\overline{\boldsymbol{\xi}}_{-\mathbf{x}}) & \text{if } \langle \mathbf{x}, \mathbf{y} \rangle_q < 0. \end{cases} \tag{8}$$

If $\langle \mathbf{x}, \mathbf{y} \rangle_q = 0$, we have $\overline{\mathsf{d}}_{\overline{\gamma}}(\mathbf{x}, \mathbf{y}) = \overline{\mathsf{d}}_{\overline{\gamma}}(-\mathbf{x}, \mathbf{y})$ and there exist two minimizing geodesics joining $[\mathbf{x}]$ and $[\mathbf{y}]$. In practice, we arbitrarily choose one of these two geodesics when $\langle \mathbf{x}, \mathbf{y} \rangle_q = 0$.

## 3.3 Optimized function $f : \mathcal{P}_r^{p,q} \to \mathbb{R}$

Our goal is to minimize some differentiable function $f : \mathcal{P}_r^{p,q} \to \mathbb{R}$. We now describe the two properties that $f$ has to satisfy. We first recall that every $[\mathbf{x}] \in \mathcal{P}_r^{p,q}$ is a set of equivalent elements that should preserve invariance. To simplify explanations, we consider the function $\overline{f} : \mathcal{S}_r^{p,q} \to \mathbb{R}$ defined such that $\overline{f} := f \circ \pi$. We then have $\forall \mathbf{x} \in \mathcal{S}_r^{p,q}, \; \overline{f}(\mathbf{x}) = f([\mathbf{x}])$.

**Property 1.** Since $\mathbf{x}$ and $-\mathbf{x}$ are equivalent, the first property that $f$ has to satisfy is $\overline{f}(\mathbf{x}) = \overline{f}(-\mathbf{x})$.

**Property 2.** Let $\nabla \overline{f}(\mathbf{x}) := (\partial \overline{f}(\mathbf{x})/\partial x_0, \ldots, \partial \overline{f}(\mathbf{x})/\partial x_d)^\top$ be the Euclidean gradient of $\overline{f}$ at $\mathbf{x} = (x_0, \ldots, x_d)^\top$. The pseudo-Riemannian gradient of $\overline{f}$ at $\mathbf{x} \in \mathcal{S}_r^{p,q}$ is $D\overline{f}(\mathbf{x}) := \Pi_\mathbf{x}(\mathbf{G}^{-1} \nabla \overline{f}(\mathbf{x})) = \Pi_\mathbf{x}(\mathbf{G} \nabla \overline{f}(\mathbf{x})) \in T_\mathbf{x} \mathcal{S}_r^{p,q}$ where $\Pi_\mathbf{x}(\mathbf{z}) := \mathbf{z} - \frac{\langle \mathbf{z}, \mathbf{x} \rangle_q}{\langle \mathbf{x}, \mathbf{x} \rangle_q} \mathbf{x}$ is the orthogonal projection of $\mathbf{z}$ onto $T_\mathbf{x} \mathcal{S}_r^{p,q}$.

Let $Df([\mathbf{x}]) \in T_{[\mathbf{x}]} \mathcal{P}_r^{p,q}$ be the pseudo-Riemannian gradient of $f$ at $[\mathbf{x}] \in \mathcal{P}_r^{p,q}$. By applying the chain rule, the second property that $f$ has to satisfy is $\text{lift}_\mathbf{x}(Df([\mathbf{x}])) = D\overline{f}(\mathbf{x}) = -D\overline{f}(-\mathbf{x})$.

## 3.4 Optimization of parametric models

We now explain how to minimize some function $f : \mathcal{P}_r^{p,q} \to \mathbb{R}$ that takes as input the ultrahyperbolic representation returned by some parametric model $\varphi_\theta$ (e.g., a neural network with parameters $\theta$) that we want to learn. We exploit the fact that, due to the properties of the (affine) Levi-Civita connection [6, 17] of $\mathcal{P}_r^{p,q}$, the metric of the manifold $\mathcal{P}_r^{p,q}$ is preserved when we work with its tangent spaces via the exponential map (see page 61 of [21]).

**Forward pass.** Let us consider the positive pole $\mathbf{p} = (r, 0, \ldots, 0)^\top \in \mathcal{S}_r^{p,q}$ defined such that only its first element $r > 0$ is nonzero. The horizontal space of $\mathbf{p}$ can be defined as the following vector space $\mathcal{H}_\mathbf{p} = T_\mathbf{p} \mathcal{S}_r^{p,q} = \{0\} \times \mathbb{R}^{p,q}$. The mapping $\varphi_\theta : \mathcal{X} \to \mathcal{H}_\mathbf{p}$ maps any input data $\boldsymbol{x} \in \mathcal{X}$ to $\mathcal{H}_\mathbf{p}$ and the resulting horizontal vector is mapped to $\mathcal{S}_r^{p,q}$ with the exponential map as follows: $\mathbf{x} := \overline{\exp}_\mathbf{p}(\varphi_\theta(\boldsymbol{x})) \in \mathcal{S}_r^{p,q}$. As mentioned above, working with the vector space $\mathcal{H}_\mathbf{p}$ greatly simplifies computations and preserves the metric thanks to the Levi-Civita connection of $\mathcal{P}_r^{p,q}$.

Note that for standard neural networks that map to $\mathbb{R}^d$, the tangent space is identified to the space itself by the natural isomorphism $T_\mathbf{x} \mathbb{R}^d \approx \mathbb{R}^d$ so the network weights also implicitly lie in the tangent space. Our approach extends Euclidean neural networks to $\mathcal{P}_r^{p,q}$.

**Backward pass.** We assume that the function $\overline{f} : \mathcal{S}_r^{p,q} \to \mathbb{R}$ satisfies the properties mentioned in Section 3.3. By exploiting Eq. (8), the horizontal lift of the parallel translate of the gradient $Df([\mathbf{x}])$ can be formulated as follows:

$$\boldsymbol{\lambda}_{[\mathbf{x}],\mathbf{p}} := \text{lift}_\mathbf{p}\left(P_{[\mathbf{x}] \frown [\mathbf{p}]}^\gamma(Df([\mathbf{x}]))\right) = \begin{cases} P_{\mathbf{x} \frown \mathbf{p}}^{\overline{\gamma}}(D\overline{f}(\mathbf{x})) & \text{if } \langle \mathbf{x}, \mathbf{p} \rangle_q \geq 0 \\ P_{-\mathbf{x} \frown \mathbf{p}}^{\overline{\gamma}}(-D\overline{f}(\mathbf{x})) & \text{otherwise.} \end{cases} \tag{9}$$

**Descent direction.** When the metric tensor of the manifold is not positive definite, the manifold is not Riemannian and the negative of $\boldsymbol{\lambda}_{[\mathbf{x}],\mathbf{p}}$ is not a descent direction [9]. We show in the supp. material that the negative of $\mathbf{G}\boldsymbol{\lambda}_{[\mathbf{x}],\mathbf{p}} \in \mathcal{H}_\mathbf{p}$ is a descent direction that can be used to optimize the parameters of $\varphi_\theta$ with standard descent algorithms. We illustrate one such example in Section 5.1.

**Complexity.** Our optimizer exploits efficient closed-form expressions on $\mathcal{S}_r^{p,q}$ by considering $\mathbf{x}$ or its antipodal point $-\mathbf{x}$ depending on its "geodesic distance" with the positive pole $\mathbf{p}$. This geodesic distance depends only on the sign of $\langle \mathbf{x}, \mathbf{p} \rangle_q$, which is also the sign of the first element of $\mathbf{x} = (x_0, \ldots, x_d)^\top$ (i.e., , $\overline{\mathsf{d}}_{\overline{\gamma}}(\mathbf{x}, \mathbf{p}) < \overline{\mathsf{d}}_{\overline{\gamma}}(-\mathbf{x}, \mathbf{p}) \iff \langle \mathbf{x}, \mathbf{p} \rangle_q > 0 \iff x_0 > 0$). Our operators generalize tools used in hyperbolic space and are then as efficient as hyperbolic approaches.

## 4 Ultrahyperbolic Graph Convolutional Network (GCN)

We now extend the hyperbolic graph neural networks introduced in [18] to $\mathcal{P}_r^{p,q}$.

**Graph Neural Networks.** We first provide some background on Graph Neural Networks (GNNs) which can be interpreted as parametric models performing message passing between nodes of a graph. We recall the formulation of Graph Convolutional Networks (GCNs) [13] and rewrite them in our formalism with quotient manifolds. Let $G = (V, E)$ be a undirected graph containing $n = |V|$ nodes and $m = |E|$ edges. Its adjacency matrix is denoted by $\mathbf{A} \in \mathbb{R}^{n \times n}$. To account for self-loops, Liu *et al.* [18] consider the matrix $\tilde{\mathbf{A}} = \mathbf{D}^{-1/2}(\mathbf{A} + \mathbf{I})\mathbf{D}^{-1/2}$ where $\mathbf{D}$ is the diagonal degree matrix defined such that $\mathbf{D}_{ii} = \sum_j (\mathbf{A}_{ij} + \mathbf{I}_{ij})$. The vector representation of node $v$ at step $k$ is denoted by $\mathbf{h}_v^k \in \mathbb{R}^d$, and $\mathbf{h}_v^0$ is given. $\mathbf{W}^k$ is a matrix whose elements are the trainable parameters of the $k$-th layer. The information in the Euclidean GCN propagates as: $\mathbf{h}_u^{k+1} = \sigma\left(\sum_{v \in \mathcal{I}(u)} \tilde{\mathbf{A}}_{uv} \mathbf{W}^k \mathbf{h}_v^k\right)$ where $\mathcal{I}(u)$ is the set of in-neighbors of $u \in V$ (i.e., $u$ and $v$ are joined by an edge) and $\sigma$ is a nonlinear activation function such as the element-wise Rectified Linear Unit (ReLU) or its variants.

**Ultrahyperbolic GNN.** Let us now consider that $\forall v, k, \mathbf{h}_v^k \in \mathcal{P}_r^{p,q}$. Since $\mathcal{P}_r^{p,q}$ is not a vector space, the operation $\mathbf{W}^k \mathbf{h}_v^k$ is not defined, and the activation function $\sigma$ has to be adapted. As in Section 3.4, we exploit properties of the Levi-Civita connection to work with the tangent spaces of $\mathcal{P}_r^{p,q}$ via the exponential map and its inverse (i.e., logarithm map). The propagation is then extended to $\mathcal{P}_r^{p,q}$ by:

$$\mathbf{h}_u^{k+1} := \sigma\left(\left[\overline{\exp}_{\mathbf{p}}\left(\sum_{v \in \mathcal{I}(u)} \tilde{\mathbf{A}}_{uv} \mathbf{W}^k \operatorname{lift}_{\mathbf{p}}\left(\log_{[\mathbf{p}]}(\mathbf{h}_v^k)\right)\right)\right]\right) \in \mathcal{P}_r^{p,q}, \tag{10}$$

where $\mathbf{p} = (r, 0, \ldots, 0)^\top$ is the positive pole and we exploit the logarithm map to map points of $\mathcal{P}_r^{p,q}$ to a single tangent space. As explained in Section 3, in practice, we use the horizontal lift operator so that the exponential and logarithm maps only consider the horizontal space $\mathcal{H}_{\mathbf{p}}$ during optimization (see supp. material for details). The hyperbolic GNN [18] corresponds to the special case where $\mathcal{P}_r^{p,q} = \mathcal{P}_1^{0,q}$ (i.e., $p = 0$). We now give the formulation of the activation function $\sigma$.

**Activation function via stereographic projection.** For simplicity of exposition, we now consider that the radius of $\mathcal{S}_r^{p,q}$ is $r = 1$. To enforce nonlinearity between the different layers of the hyperbolic graph neural network, Liu et al. [18] formulate their activation function as the result of a steoreographic projection onto the negative pole $-\mathbf{p}$ from the hyperboloid model to the Poincaré ball, followed by a ReLU activation (in the Poincaré ball) and an inverse steoreographic projection from the Poincaré ball to the hyperboloid. We explain below how to generalize $\sigma$ to pseudo-spheres.

Let us note $\varepsilon \in \{-1, 1\}$. The pole $\varepsilon\mathbf{p} = (\varepsilon, 0, \ldots, 0)^\top$ is positive if $\varepsilon = 1$, and negative if $\varepsilon = -1$. Let us consider a point $\mathbf{x} = (x_0, x_1, \ldots, x_d)^\top \in \mathcal{S}_1^{p,q}$ with $x_0 > 0$ (i.e., lying on the positive hemisphere). The stereographic projection of $\mathbf{x}$ onto $\varepsilon\mathbf{p}$ is $\mathbf{a} = \omega_\varepsilon(\mathbf{x}) := \frac{1}{1 - \varepsilon x_0}(x_1, x_2, \ldots, x_d)^\top$. If $x_0 < 0$, we equivalently consider that $\mathbf{a} = \omega_\varepsilon(-\mathbf{x}) = -\omega_{-\varepsilon}(\mathbf{x})$ instead of $\omega_\varepsilon(\mathbf{x})$ due to the quotient nature of $\mathcal{P}_r^{p,q}$ and to account for the fact that $[\mathbf{x}]$ is projected onto the pole of different hemisphere if $\varepsilon = -1$, or same hemisphere if $\varepsilon = 1$. The inverse projection of $\mathbf{a} = (a_1, \ldots, a_d)^\top \in \mathbb{R}^{p,q}$ is:

$$\omega_\varepsilon^{-1}(\mathbf{a}) := \frac{1}{1 + \langle \mathbf{a}, \mathbf{a} \rangle_q}\begin{pmatrix} \varepsilon(\langle \mathbf{a}, \mathbf{a} \rangle_q - 1) \\ 2\mathbf{a} \end{pmatrix} \in \mathcal{S}_1^{p,q} \quad \text{where } \langle \mathbf{a}, \mathbf{a} \rangle_q := \sum_{i=1}^{p} a_i^2 - \sum_{j=p+1}^{d} a_j^2. \tag{11}$$

We formulate $\sigma([\mathbf{x}]) := [\omega_\varepsilon^{-1}(\operatorname{ReLU}(\omega_\varepsilon(\mathbf{x})))]$ if $x_0 \geq 0$, and $\sigma([\mathbf{x}]) := [\omega_\varepsilon^{-1}(\operatorname{ReLU}(\omega_\varepsilon(-\mathbf{x})))]$ otherwise, where ReLU (or one of its variants such as LeakyRelu) is applied element-wise only on the $q$ time dimensions of the input vector, which avoids having a zero denominator in Eq. (11). As in [18], we consider $\varepsilon = -1$. It is worth noting that Liu *et al.* [18] work with the upper sheet of the hyperboloid $\mathcal{Q}_1^{q,0}$ which is anti-isometric to $\mathcal{S}_1^{0,q}$. Their stereographic projection then contains only space dimensions. Their space dimensions correspond to our time dimensions due to anti-isometry.

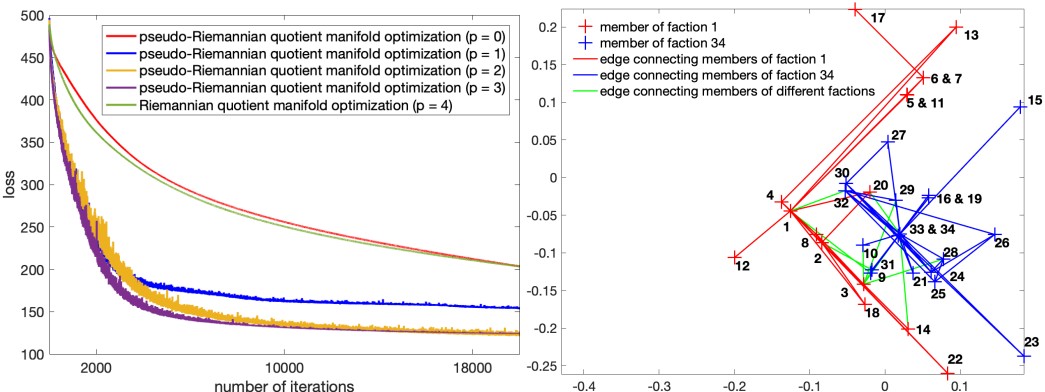

Figure 2: (left) Loss values of Eq. (12) as a function of the number of iterations for different values of $p$ when $\mathcal{P}_r^{p,q}$ is 4-dimensional. (right) Stereographic projection onto $-\mathbf{p}$ of representations lying on $\mathcal{P}_1^{1,1}$ and learned from Zachary's karate club. Node colors define the faction joined by the members.

Table 1: Evaluation scores for the different learned representations (mean $\pm$ standard deviation)

| Evaluation metric | $\mathbb{R}^4$ (Euclidean) | $\mathcal{P}_1^{0,4}$ (Hyperbolic) | $\mathcal{P}_1^{1,3}$ | $\mathcal{P}_1^{2,2}$ | $\mathcal{P}_1^{3,1}$ | $\mathcal{P}_1^{4,0}$ (Elliptic) |
|---|---|---|---|---|---|---|
| Rank of first leader | $4.6 \pm 1.0$ | $2.5 \pm 0.7$ | $\mathbf{1.2 \pm 0.4}$ | $1.3 \pm 0.7$ | $\mathbf{1.2 \pm 0.4}$ | $2.5 \pm 0.8$ |
| Rank of second leader | $6.9 \pm 0.7$ | $3.8 \pm 1.0$ | $\mathbf{2.7 \pm 0.7}$ | $3.1 \pm 1.0$ | $4.4 \pm 3.0$ | $3.6 \pm 0.7$ |
| top 5 Spearman's $\rho$ | $0.06 \pm 0.45$ | $0.36 \pm 0.22$ | $0.62 \pm 0.23$ | $0.61 \pm 0.28$ | $\mathbf{0.63 \pm 0.35}$ | $0.46 \pm 0.29$ |
| top 10 Spearman's $\rho$ | $0.04 \pm 0.19$ | $0.38 \pm 0.18$ | $\mathbf{0.73 \pm 0.12}$ | $0.72 \pm 0.07$ | $0.63 \pm 0.16$ | $0.38 \pm 0.26$ |
| Training time (seconds) | $\mathbf{340 \pm 4}$ | $424 \pm 1$ | $429 \pm 1$ | $430 \pm 2$ | $429 \pm 1$ | $402 \pm 1$ |

# 5 Experiments

We now evaluate our approach on different classification tasks on graphs. We first show that our optimization framework introduced in Section 3.4 learns meaningful representations on a toy hierarchical graph with cycles. We then apply our framework in standard classification tasks.

## 5.1 Last layer optimization on a toy dataset

We evaluate our optimization framework by training a multi-layer perceptron (MLP) $\varphi_\theta : \mathcal{X} \to \mathcal{H}_{\mathbf{p}}$ whose set of parameters is called $\theta$. As in [15], we test our approach on Zachary's karate club dataset [33]. However, instead of learning embeddings, we train a parametric model.

**Zachary's dataset** is a social network graph that represents a karate club split in two factions due to a conflict between two leaders (the instructor and the administrator). It is an undirected graph $G = (V, E)$ which has node-set $V = \{v_i\}_{i=1}^n$ and edge-set $E = \{e_k\}_{k=1}^m$ where $n = 34$ and $m = 78$. Each node $v_i$ represents a karate member and an edge joins two nodes if the two members are friends. The two leaders are $v_1$ and $v_{34}$. We consider that each node $v_i$ is represented as a distinct $n$-dimensional one-hot vector $\boldsymbol{x}_i \in \mathcal{X}$.

**Problem.** Following [15], our goal is to learn representations of nodes such that pairs of nodes joined by an edge (i.e., in $E$) have smaller distance than pairs of nodes that are not joined by an edge (i.e., not in $E$). Our problem is then to find the set of parameters $\theta$ that minimizes the problem:

$$\min_\theta \sum_{(v_i, v_j) \in E} -\log \frac{e^{-\mathsf{d}(\varrho_\theta(\boldsymbol{x}_i), \varrho_\theta(\boldsymbol{x}_j))/\tau}}{\sum_{(v_a, v_b) \in \mathcal{W}_{ij}} e^{-\mathsf{d}(\varrho_\theta(\boldsymbol{x}_a), \varrho_\theta(\boldsymbol{x}_b))/\tau}} \quad \text{where } \varrho_\theta(\boldsymbol{x}_i) := [\overline{\exp}_{\mathbf{p}}(\varphi_\theta(\boldsymbol{x}_i))] \quad (12)$$

and where $\mathcal{W}_{ij} := \{(v_i, v_j)\} \cup \{(v_a, v_b) \notin E\}$, $\tau = 10^{-2}$ is a fixed temperature value, and d denotes the geodesic distance of the manifold (e.g., Eq. (7) for $\mathcal{P}_r^{p,q}$). The geodesic distance satisfies the two properties defined in Section 3.3 with respect to each input and can then be used for optimization.

**Model.** Our MLP $\varphi_\theta : \mathcal{X} \to \mathcal{H}_{\mathbf{p}}$ contains three hidden layers of $10^4$ hidden units each, with standard ReLU as nonlinear activation function. In this toy experiment, our MLP is standard, with the only

exception that its last layer maps to the horizontal space $\mathcal{H}_{\mathbf{p}} = \{0\} \times \mathbb{R}^{p,q}$ of the positive pole $\mathbf{p}$. The output representation is then mapped with the exponential map as explained in Section 3.4.

**Optimizer.** We use the optimizer introduced in Section 3.4 to update $\theta$. By using the descent direction $-\mathbf{G}\boldsymbol{\lambda}_{[\mathbf{x}_i],\mathbf{p}}$ for each sample $[\mathbf{x}_i] = \varrho_\theta(\boldsymbol{x}_i)$, all the parameters of our standard MLP lie in some space equipped with a positive definite metric tensor. Standard backpropagation is then used to optimize the parameters. As an illustration, we consider the 4-dimensional manifold $\mathcal{P}_1^{p,q}$ (i.e., $p + q = 4$) and show in Fig. 2 (left) the loss values of Eq. (12) as a function of the number of iterations for different values of $p \in \{0, \dots, 4\}$. The figure shows that the optimization framework in Section 3.4 decreases the loss value. Moreover, it is worth noting that the algorithm does not converge if $-\boldsymbol{\lambda}_{[\mathbf{x}_i],\mathbf{p}}$ is used as a search direction (instead of $-\mathbf{G}\boldsymbol{\lambda}_{[\mathbf{x}_i],\mathbf{p}}$) when the metric tensor is not positive definite since $-\boldsymbol{\lambda}_{[\mathbf{x}_i],\mathbf{p}}$ is not a descent direction [9]. More details can be found in the supp. material.

**Hierarchy extraction.** We now evaluate the quality of the learned representations in the task of predicting the high-level nodes of the graph. Our evaluation protocol is similar to [15], the only difference is that we train a neural network. We run 10 random initializations for each considered 4-dimensional manifold and report in Table 1 the mean and standard deviation of the different evaluation metrics.

As in [15], following the idea that hyperbolic distances grow exponentially, we take the sum of distances $\delta_i = \sum_{j=1}^n \mathsf{d}([\mathbf{x}_i], [\mathbf{x}_j])$ of a node $v_i$ with all the other nodes as an indicator of importance. We sort the different $\delta_1, \dots, \delta_n$ in ascending order and report the rank of the two leaders (instructor and administrator, in no particular order) in the first two rows of Table 1. The leaders tend to have smaller $\delta_i$ score than low-level nodes with ultrahyperbolic distances, which means that high-level nodes tend to be closer to the rest of the nodes in ultrahyperbolic space.

We also measure the Spearman's rank correlation coefficient [28] between the 5 (or 10) most important nodes in the hierarchy and their corresponding $\delta_i$ score. Once again, the order of the $\delta_i$ scores is more correlated with the hierarchy level in ultrahyperbolic space. Our experimental results are comparable with [15] although our nodes are represented on a quotient manifold and we learn a parametric model. Fig. 2 (right) illustrates our learned representations when the manifold is $\mathcal{P}_1^{1,1}$.

**Products of Riemannian space forms.** In Table 1, we compare the performance of models mapping representations to pseudo-Riemannian space forms (i.e., manifolds of constant curvature [21, 30]). Nonetheless, it was already noticed in the machine learning literature that products of Riemannian space forms (called *mixed-curvature representations*) could outperform Riemannian space forms when the structure of the dataset is not tree-like [3, 11]. It is worth noting that products of space forms are in general not space forms (except if they are all flat). For this reason, we do not compare them to our manifold in the main article as we could similarly consider products of pseudo-spheres $\mathcal{P}_{r_1}^{p_1,q_1} \times \mathcal{P}_{r_2}^{p_2,q_2}$ or even $\mathcal{P}_{r_1}^{p_1,q_1} \times \mathbb{R}^{p_2,q_2}$ for evaluation.

Nonetheless, since our space form $\mathcal{P}_r^{p,q}$ contains hyperbolic and elliptic parts, we provide a detailed comparison with products of hyperbolic and spherical spaces in the supp. material. Such product manifolds perform better than hyperbolic and spherical spaces but slightly worse than the pseudo-Riemannian space form $\mathcal{P}_r^{p,q}$.

**Training times.** We report in Table 1 the training times of our Pytorch [22] implementation to train 25,000 iterations on a machine equipped with a 6-core Intel i7-7800X CPU and NVIDIA GeForce RTX 3090 GPU. All the representations lying on a non-flat manifold have comparable training times. Nonetheless, they are 25% slower than the Euclidean approach because they compute the pseudo-Riemannian gradient (which requires an orthogonal projection) and parallel transport.

## 5.2 Classification with ultrahyperbolic graph convolutional networks

The previous subsection analyzed our framework. We now evaluate it in standard classification tasks.

**Node classification.** We now evaluate the generalization performance of our GCN in the semi-supervised node classification task on three citation network datasets: Citeseer, Cora and Pubmed [26]. They contain sparse bag-of-words feature vectors for each document and a list of citation links between documents. Each document is a node and has a class label. Each citation link is an undirected edge. Dataset statistics are reported in Table 2. During training, all the nodes and edges are preserved, but only 20 nodes per class are labeled, and 500 nodes are used for validation in total, the rest for test. We follow the experimental protocol of Appendix A of [18] and learn a GCN with 2 hidden layers.

Table 2: Statistics of the citation network datasets.

| Name | # Nodes | # Edges | # Classes | # Features | # training nodes per category |
|------|---------|---------|-----------|------------|-------------------------------|
| Citeseer | 3,327 | 4,732 | 6 | 3,703 | 20 |
| Cora | 2,708 | 5,429 | 7 | 1,433 | 20 |
| Pubmed | 19,717 | 44,338 | 3 | 500 | 20 |

Table 3: Test node classification accuracy with 4-dimensional manifolds

| Dataset | $\mathbb{R}^4$ (standard GCN) | $\mathcal{P}_1^{0,4}$ (Hyperbolic) | $\mathcal{P}_1^{1,3}$ | $\mathcal{P}_1^{2,2}$ | $\mathcal{P}_1^{3,1}$ | $\mathcal{P}_1^{4,0}$ (Elliptic) |
|---------|------|------|------|------|------|------|
| Citeseer | $44.5 \pm 5.9$ | $46.7 \pm 1.8$ | $\mathbf{51.8 \pm 2.6}$ | $50.3 \pm 2.1$ | $51.4 \pm 3.2$ | $47.2 \pm 2.6$ |
| Cora | $53.5 \pm 4.3$ | $56.2 \pm 3.1$ | $63.2 \pm 3.3$ | $63.9 \pm 3.1$ | $\mathbf{64.7 \pm 5.3}$ | $61.4 \pm 1.5$ |
| Pubmed | $66.9 \pm 2.3$ | $71.5 \pm 2.9$ | $\mathbf{73.1 \pm 0.6}$ | $72.8 \pm 2.7$ | $71.2 \pm 2.7$ | $71.0 \pm 2.7$ |

Table 4: Statistics of the graph datasets used for the classification task

| Name | # graphs | # classes | Avg. # nodes | Avg. # edges | Type of dataset |
|------|----------|-----------|--------------|--------------|-----------------|
| Collab | 5,000 | 3 | 74.49 | 2457.78 | Scientific collaboration dataset [32] |
| D&D | 1,178 | 2 | 284.32 | 715.66 | Protein dataset [25] |
| Enzymes | 600 | 6 | 32.63 | 62.14 | Protein dataset [25] |
| Proteins | 1,113 | 2 | 39.06 | 72.82 | Protein dataset [25] |
| Reddit-multi-12K | 11,929 | 11 | 391.41 | 456.89 | Social network dataset [32] |

Table 5: Graph classification accuracy in percents. $d$ is the dimensionality of the manifold.

| Method | Collab ($d = 64$) | D&D ($d = 88$) | Enzymes ($d = 256$) | Proteins ($d = 100$) | Reddit ($d = 100$) |
|--------|-------------------|----------------|---------------------|----------------------|--------------------|
| Euclidean (standard GCN) | $81.88 \pm 1.76$ | $76.93 \pm 7.21$ | $43.83 \pm 10.3$ | $75.46 \pm 3.88$ | $45.65 \pm 1.76$ |
| Poincaré (hyperbolic) | $80.92 \pm 1.99$ | $75.89 \pm 8.53$ | $44.15 \pm 8.43$ | $73.64 \pm 4.64$ | $45.84 \pm 1.42$ |
| Lorentz (hyperbolic) | $81.32 \pm 1.21$ | $77.10 \pm 6.65$ | $44.83 \pm 8.14$ | $74.16 \pm 3.25$ | $45.39 \pm 1.53$ |
| Ultrahyperbolic | $\mathbf{82.26 \pm 1.23}$ | $\mathbf{81.97 \pm 3.41}$ | $\mathbf{50.50 \pm 6.71}$ | $\mathbf{76.56 \pm 2.09}$ | $\mathbf{47.08 \pm 1.26}$ |

When the dimensionality of each layer is $d = 600$, all the Euclidean (i.e., standard), Hyperbolic and Ultrahyperbolic GCNs reach the same test accuracy because the model is overparameterized and quickly attains $100\%$ accuracy on the training set. See details and scores in the supp. material.

Due to the problem mentioned above, we trained GCNs whose dimensionality of each layer is $d = 4$ with 100 random initializations. The results reported in Table 3 show the superiority of ultrahyperbolic representations in low-dimensional space for node classification of hierarchical graphs with cycles. We also report results for $d = 10$ in the supp. material. The conclusion is similar.

**Graph classification.** We also evaluate our approach on commonly used graph kernel benchmark datasets [12] whose statistics are reported in Table 4. The evaluation is done via 10-fold cross validation. We use the same protocol evaluation and splits as in Appendix E of [18] and evaluate our approach in the same settings including same number of GNN layers, optimizers, learning rate, and manifold dimensionality $d$ reported in Table 5. The only difference is that the data is represented on $\mathcal{P}_r^{p,q}$ with $p = 1$ in our case. The comparative performances are reported in Table 5 and show that ultrahyperbolic representations significantly improve performance on the D&D and Enzymes datasets, which are protein datasets from [25]. The gain is less significant on the other datasets but our approach is still competitive. It seems that the advantage of our approach over hyperbolic approaches is more visible for protein structures than for social networks, at least in high-dimensional space. More details can be found in the supp. material.

# 6 Conclusion, Limitations and Potential Societal Impacts

We have introduced neural networks that map data to a (quotient) pseudo-Riemannian manifold of constant nonzero curvature. Our considered geometry generalizes both hyperbolic and elliptic geometries. It is the first neural network that maps data to a non-Riemannian manifold to the best of our knowledge. Our framework is general and can be applied to many parametric models and tasks. We demonstrate this via graph convolutional networks and show improved performance compared to Euclidean and hyperbolic approaches to represent hierarchical graphs in different tasks.

Concurrently with this work, Xiong *et al.* [31] proposed an extension of graph convolutional networks to the pseudo-hyperboloid $\mathcal{Q}_r^{q,p}$ which is a pseudo-Riemannian manifold of constant nonzero curvature anti-isometric to the pseudo-sphere $\mathcal{S}_r^{p,q}$. One main difference is that, since there exist pairs of points of $\mathcal{Q}_r^{q,p}$ that cannot be joined by an unbroken geodesic, the optimization framework in [31] does not exploit the intrinsic geometry of the manifold via its Levi-Civita connection. On the other hand, our approach uses pseudo-Riemannian optimization tools that are intrinsic to $\mathcal{P}_r^{p,q}$. The ablation study in [31] also suggests that graphs with more hierarchical structure are better represented when the manifold becomes more hyperbolic, and graphs with cyclic relationships are better represented when the manifold becomes more spherical.

**Limitations.** Our main contribution is a solid optimization framework that is well defined thanks to the use of standard differential geometry tools (e.g., canonical map and horizontal bundle) that we formulate for the quotient manifold $\mathcal{P}_r^{p,q}$. It only requires the properties of the optimized function in Section 3.3 to be satisfied. This is for instance the case if points of $\mathcal{P}_r^{p,q}$ are compared with the geodesic distance in Eq. (7). We applied our framework on nine different datasets with (at least 10) different runs to validate our results. Our work lacks a theoretical analysis similar to Gromov's work [10] in the case of graphs without cycles. However, the optimal geometry for graphs with cycles is still an open problem, and hyperbolic geometry is used heuristically in this case. Our motivation is that ultrahyperbolic manifolds are more general than hyperbolic and elliptic manifolds, they can then combine the strengths of the two induced geometries. We experimentally validate our assumption in different tasks and leave the theoretical analysis for future work.

**Potential societal impacts.** Our contributions are mainly methodological although we apply our approach to hierarchical graphs that could represent social networks. Improving accuracy on these datasets might facilitate the task of discovering leaders in social networks, which could have negative impact if not monitored. Nonetheless, we also show improvement on protein structures, this could have positive impacts on society and healthcare. We did not exploit any personally identifiable information. We used datasets that have been publicly available to the machine learning community for years. Our method to handle and process the data is standard in the graph community.

**Acknowledgments and Funding Transparency Statement.** I thank James Lucas, Rafid Mahmood, Haggai Maron and the anonymous reviewers for helpful feedback on early versions of this manuscript. This project was entirely funded by NVIDIA corporation while I was working from home during the COVID-19 pandemic. I am also grateful to the NeurIPS 2021 program committee for giving me free registration to the conference thanks to an Outstanding Reviewer Award.

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
