# A   Supplementary Material

The supplementary material is structured as follows:

• In Section B, we give the formulations of the differential geometry tools to work on the pseudo-sphere $\mathcal{S}_r^{p,q}$ (Section B.1), the indefinite elliptic space $\mathcal{P}_r^{p,q}$ (Section B.2) and the pseudo-hyperboloid $\mathcal{Q}_r^{q,p}$ (Section B.3). The tools include the formulation of a geodesic, exponential map, logarithm map and geodesic *distance*. In Section B.4, we explain the anti-isometry between the pseudo-sphere and the pseudo-hyperboloid. In Section B.5, we give more details about Figure 1. In Section B.6, we explain how the ultrahyperbolic manifold $\mathcal{P}_r^{p,q}$ contains hyperbolic and spherical/elliptic parts.

• In Section C, we explain how we optimize our neural networks. In particular, the pseudo-Riemannian gradient is not always a descent direction so we exploit results in [9] to find a descent direction in an efficient way.

• In Section D, we provide experimental details and additional results.

# B   Differential Geometry Tools

We provide here the necessary differential geometry tools to work on the pseudo-sphere $\mathcal{S}_r^{p,q}$ and the quotient manifold $\mathcal{P}_r^{p,q}$. Most of them are explained in [15] for the case of the pseudo-hyperboloid that is anti-isometric to the pseudo-sphere (see Section B.4 for details). We recall that the radius $r$ of the pseudo-sphere is positive, and we consider that $r = 1$ in our experiments.

## B.1   Pseudo-sphere $\mathcal{S}_r^{p,q}$

We give here the differential geometry tools specific to the pseudo-sphere which is defined as the following set: $\mathcal{S}_r^{p,q} := \left\{ \mathbf{x} \in \mathbb{R}^{p+1,q} : \langle \mathbf{x}, \mathbf{x} \rangle_q = r^2 \right\}$.

### B.1.1   Geodesic, exponential map and distance

**Geodesic.** The geodesic $\overline{\gamma}_{\mathbf{x} \to \overline{\boldsymbol{\xi}}_{\mathbf{x}}} : \mathbb{R} \to \mathcal{S}_r^{p,q}$ satisfying $\overline{\gamma}_{\mathbf{x} \to \overline{\boldsymbol{\xi}}_{\mathbf{x}}}(0) = \mathbf{x}$ and $\overline{\gamma}'_{\mathbf{x} \to \overline{\boldsymbol{\xi}}_{\mathbf{x}}}(0) = \overline{\boldsymbol{\xi}}_{\mathbf{x}} \in T_{\mathbf{x}}\mathcal{S}_r^{p,q}$ is formulated for all $t \in \mathbb{R}$:

$$
\overline{\gamma}_{\mathbf{x} \to \overline{\boldsymbol{\xi}}_{\mathbf{x}}}(t) = \begin{cases} \cos\left( \frac{t\sqrt{|\langle \overline{\boldsymbol{\xi}}_{\mathbf{x}}, \overline{\boldsymbol{\xi}}_{\mathbf{x}} \rangle_q|}}{r} \right) \mathbf{x} \ + \frac{r}{\sqrt{|\langle \overline{\boldsymbol{\xi}}_{\mathbf{x}}, \overline{\boldsymbol{\xi}}_{\mathbf{x}} \rangle_q|}} \sin\left( \frac{t\sqrt{|\langle \overline{\boldsymbol{\xi}}_{\mathbf{x}}, \overline{\boldsymbol{\xi}}_{\mathbf{x}} \rangle_q|}}{r} \right) \overline{\boldsymbol{\xi}}_{\mathbf{x}} & \text{if } \langle \overline{\boldsymbol{\xi}}_{\mathbf{x}}, \overline{\boldsymbol{\xi}}_{\mathbf{x}} \rangle_q > 0 \\ \mathbf{x} + t\overline{\boldsymbol{\xi}}_{\mathbf{x}} & \text{if } \langle \overline{\boldsymbol{\xi}}_{\mathbf{x}}, \overline{\boldsymbol{\xi}}_{\mathbf{x}} \rangle_q = 0 \\ \cosh\left( \frac{t\sqrt{|\langle \overline{\boldsymbol{\xi}}_{\mathbf{x}}, \overline{\boldsymbol{\xi}}_{\mathbf{x}} \rangle_q|}}{r} \right) \mathbf{x} + \frac{r}{\sqrt{|\langle \overline{\boldsymbol{\xi}}_{\mathbf{x}}, \overline{\boldsymbol{\xi}}_{\mathbf{x}} \rangle_q|}} \sinh\left( \frac{t\sqrt{|\langle \overline{\boldsymbol{\xi}}_{\mathbf{x}}, \overline{\boldsymbol{\xi}}_{\mathbf{x}} \rangle_q|}}{r} \right) \overline{\boldsymbol{\xi}}_{\mathbf{x}} & \text{if } \langle \overline{\boldsymbol{\xi}}_{\mathbf{x}}, \overline{\boldsymbol{\xi}}_{\mathbf{x}} \rangle_q < 0 \end{cases}
\tag{13}
$$

The nonconstant geodesic $\overline{\gamma}_{\mathbf{x} \to \overline{\boldsymbol{\xi}}_{\mathbf{x}}}$ (i.e., $\overline{\boldsymbol{\xi}}_{\mathbf{x}} \neq \mathbf{0}$) is called:

• **space-like** if $\langle \overline{\boldsymbol{\xi}}_{\mathbf{x}}, \overline{\boldsymbol{\xi}}_{\mathbf{x}} \rangle_q > 0$.

• **null** if $\langle \overline{\boldsymbol{\xi}}_{\mathbf{x}}, \overline{\boldsymbol{\xi}}_{\mathbf{x}} \rangle_q = 0$.

• **time-like** if $\langle \overline{\boldsymbol{\xi}}_{\mathbf{x}}, \overline{\boldsymbol{\xi}}_{\mathbf{x}} \rangle_q < 0$.

**Exponential map.** The exponential map $\overline{\exp}_{\mathbf{x}} : T_{\mathbf{x}}\mathcal{S}_r^{p,q} \to \mathcal{S}_r^{p,q}$ is defined such that $\forall \overline{\boldsymbol{\xi}}_{\mathbf{x}} \in T_{\mathbf{x}}\mathcal{S}_r^{p,q}, \overline{\exp}_{\mathbf{x}}(\overline{\boldsymbol{\xi}}_{\mathbf{x}}) = \overline{\gamma}_{\mathbf{x} \to \overline{\boldsymbol{\xi}}_{\mathbf{x}}}(1)$. We then have:

$$
\overline{\exp}_{\mathbf{x}}(\overline{\boldsymbol{\xi}}_{\mathbf{x}}) = \begin{cases} \cos\left( \frac{\sqrt{|\langle \overline{\boldsymbol{\xi}}_{\mathbf{x}}, \overline{\boldsymbol{\xi}}_{\mathbf{x}} \rangle_q|}}{r} \right) \mathbf{x} \ + \frac{r}{\sqrt{|\langle \overline{\boldsymbol{\xi}}_{\mathbf{x}}, \overline{\boldsymbol{\xi}}_{\mathbf{x}} \rangle_q|}} \sin\left( \frac{\sqrt{|\langle \overline{\boldsymbol{\xi}}_{\mathbf{x}}, \overline{\boldsymbol{\xi}}_{\mathbf{x}} \rangle_q|}}{r} \right) \overline{\boldsymbol{\xi}}_{\mathbf{x}} & \text{if } \langle \overline{\boldsymbol{\xi}}_{\mathbf{x}}, \overline{\boldsymbol{\xi}}_{\mathbf{x}} \rangle_q > 0 \\ \mathbf{x} + \overline{\boldsymbol{\xi}}_{\mathbf{x}} & \text{if } \langle \overline{\boldsymbol{\xi}}_{\mathbf{x}}, \overline{\boldsymbol{\xi}}_{\mathbf{x}} \rangle_q = 0 \\ \cosh\left( \frac{\sqrt{|\langle \overline{\boldsymbol{\xi}}_{\mathbf{x}}, \overline{\boldsymbol{\xi}}_{\mathbf{x}} \rangle_q|}}{r} \right) \mathbf{x} + \frac{r}{\sqrt{|\langle \overline{\boldsymbol{\xi}}_{\mathbf{x}}, \overline{\boldsymbol{\xi}}_{\mathbf{x}} \rangle_q|}} \sinh\left( \frac{\sqrt{|\langle \overline{\boldsymbol{\xi}}_{\mathbf{x}}, \overline{\boldsymbol{\xi}}_{\mathbf{x}} \rangle_q|}}{r} \right) \overline{\boldsymbol{\xi}}_{\mathbf{x}} & \text{if } \langle \overline{\boldsymbol{\xi}}_{\mathbf{x}}, \overline{\boldsymbol{\xi}}_{\mathbf{x}} \rangle_q < 0 \end{cases}
\tag{14}
$$

**Logarithm map.** The logarithm map $\overline{\log}_{\mathbf{x}}$ is defined as the inverse of the exponential map $\overline{\exp}_{\mathbf{x}}$ on a normal neighborhood of $\mathbf{x} \in \mathcal{S}_r^{p,q}$ denoted by $\mathcal{U}_{\mathbf{x}} = \{\mathbf{y} \in \mathcal{S}_r^{p,q} : \frac{\langle \mathbf{x}, \mathbf{y} \rangle_q}{r^2} > -1\}$. It is then formulated:

$$\forall \mathbf{y} \in \mathcal{U}_{\mathbf{x}}, \overline{\log}_{\mathbf{x}}(\mathbf{y}) = \begin{cases} \dfrac{\cos^{-1}(\frac{\langle \mathbf{x},\mathbf{y}\rangle_q}{r^2})}{\sqrt{1-(\frac{\langle \mathbf{x},\mathbf{y}\rangle_q}{r^2})^2}}\left(\mathbf{y}-\dfrac{\langle \mathbf{x},\mathbf{y}\rangle_q}{r^2}\mathbf{x}\right) & \text{if } \frac{\langle \mathbf{x},\mathbf{y}\rangle_q}{r^2} \in (-1,1) \\[3ex] \mathbf{y}-\mathbf{x} & \text{if } \frac{\langle \mathbf{x},\mathbf{y}\rangle_q}{r^2}=1 \\[3ex] \dfrac{\cosh^{-1}(\frac{\langle \mathbf{x},\mathbf{y}\rangle_q}{r^2})}{\sqrt{(\frac{\langle \mathbf{x},\mathbf{y}\rangle_q}{r^2})^2-1}}\left(\mathbf{y}-\dfrac{\langle \mathbf{x},\mathbf{y}\rangle_q}{r^2}\mathbf{x}\right) & \text{if } \frac{\langle \mathbf{x},\mathbf{y}\rangle_q}{r^2}>1 \end{cases} \tag{15}$$

**Geodesic "distance".** As explained in [15] and Chapter 5 of [21], when the logarithm map $\overline{\log}$ exists for some pseudo-Riemannian manifold $\mathcal{M}$, the arc length of the tangent vector joining $\mathbf{x} \in \mathcal{M}$ and $\mathbf{y} \in \mathcal{M}$ corresponds to the radius function: $\sqrt{|g_{\mathbf{x}}(\overline{\log}_{\mathbf{x}}(\mathbf{y}), \overline{\log}_{\mathbf{x}}(\mathbf{y}))|}$ where $g_{\mathbf{x}} : T_{\mathbf{x}}\mathcal{M} \times T_{\mathbf{x}}\mathcal{M} \to \mathbb{R}$ is the metric tensor at $\mathbf{x}$ and $\overline{\log}_{\mathbf{x}}$ is the logarithm map. In the case of the pseudo-sphere, we have $g_{\mathbf{x}}(\cdot,\cdot) = \langle \cdot,\cdot\rangle_q$. The geodesic distance $\overline{d}_{\overline{\gamma}} : \mathcal{S}_r^{p,q} \times \mathcal{S}_r^{p,q} \to \mathbb{R}$ is then:

$$\overline{d}_{\overline{\gamma}}(\mathbf{x},\mathbf{y}) = \sqrt{|\langle \overline{\log}_{\mathbf{x}}(\mathbf{y}), \overline{\log}_{\mathbf{x}}(\mathbf{y})\rangle_q|} = \begin{cases} r\cosh^{-1}(\frac{\langle \mathbf{x},\mathbf{y}\rangle_q}{r^2}) & \text{if } \frac{\langle \mathbf{x},\mathbf{y}\rangle_q}{r^2} \geq 1 \\[2ex] r\cos^{-1}(\frac{\langle \mathbf{x},\mathbf{y}\rangle_q}{r^2}) & \text{if } \frac{\langle \mathbf{x},\mathbf{y}\rangle_q}{r^2} \in (-1,1) \end{cases} \tag{16}$$

$\overline{d}_{\overline{\gamma}}$ is not a "distance metric" but a symmetric premetric: it satisfies (i) $\overline{d}_{\overline{\gamma}}(\mathbf{x},\mathbf{y}) = \overline{d}_{\overline{\gamma}}(\mathbf{y},\mathbf{x}) \geq 0$ and (ii) $\overline{d}_{\overline{\gamma}}(\mathbf{x},\mathbf{x}) = 0$.

In [21], the "minimizing geodesic" is defined by its arc length and then also corresponds to our geodesic distance.

### B.1.2  Parallel transport on $\mathcal{S}_r^{p,q}$

The parallel transport formula is given in Eq. (6) of the main paper. For completeness, we write it here again. We also provide the proof that is inspired by [9] wherein the parallel transport on $\mathcal{S}_1^{p,q}$ along any geodesic is provided. We assume that $\mathbf{x}$ and $\mathbf{y}$ can be joined by an unbroken geodesic, the minimizing geodesic can then be formulated as a function of the logarithm map.

Given the minimizing geodesic $\overline{\gamma}$ connecting $\mathbf{x}$ to $\mathbf{y}$, the parallel transport $P_{\mathbf{x}\curvearrowright\mathbf{y}}^{\overline{\gamma}} : T_{\mathbf{x}}\mathcal{S}_r^{p,q} \to T_{\mathbf{y}}\mathcal{S}_r^{p,q}$ is a linear isometry such that $\forall \overline{\boldsymbol{\xi}}_{\mathbf{x}}, \overline{\boldsymbol{\zeta}}_{\mathbf{x}}, \langle \overline{\boldsymbol{\xi}}_{\mathbf{x}}, \overline{\boldsymbol{\zeta}}_{\mathbf{x}}\rangle_q = \langle P_{\mathbf{x}\curvearrowright\mathbf{y}}^{\overline{\gamma}}(\overline{\boldsymbol{\xi}}_{\mathbf{x}}), P_{\mathbf{x}\curvearrowright\mathbf{y}}^{\overline{\gamma}}(\overline{\boldsymbol{\zeta}}_{\mathbf{x}})\rangle_q$. The parallel transport along $\overline{\gamma}$ from $\mathbf{x} = \overline{\gamma}(0)$ to $\mathbf{y} = \overline{\gamma}(1)$ (where $\mathbf{x}$ and $\mathbf{y}$ satisfy $\langle \mathbf{x},\mathbf{y}\rangle_q > -r^2$) is:

$$P_{\mathbf{x}\curvearrowright\mathbf{y}}^{\overline{\gamma}}(\overline{\boldsymbol{\xi}}_{\mathbf{x}}) := \overline{\boldsymbol{\xi}}_{\mathbf{x}} - \frac{\langle \mathbf{y},\overline{\boldsymbol{\xi}}_{\mathbf{x}}\rangle_q}{\langle \mathbf{x},\mathbf{y}\rangle_q + r^2}(\mathbf{y}+\mathbf{x}) \tag{17}$$

**Proof.** To prove the correctness of the above formula, we follow the general properties of parallel transport mentioned in [9]. We briefly recall them here. We refer the reader to [9] for details.

We denote the *semi-normal space* of $\mathcal{S}_r^{p,q}$ in $\mathbb{R}^{p+1,q}$ at $\mathbf{x}$ by $\mathrm{SN}_{\mathbf{x}}(\mathcal{S}_r^{p,q}, \mathbb{R}^{p+1,q})$. It is defined as:

$$\mathrm{SN}_{\mathbf{x}}(\mathcal{S}_r^{p,q}, \mathbb{R}^{p+1,q}) := \{\mathbf{y} \in \mathbb{R}^{p+1,q} : \forall \overline{\boldsymbol{\zeta}}_{\mathbf{x}} \in T_{\mathbf{x}}\mathcal{S}_r^{p,q}, \langle \mathbf{y},\overline{\boldsymbol{\zeta}}_{\mathbf{x}}\rangle_q = 0\} = \{\lambda \mathbf{x} : \lambda \in \mathbb{R}\} \tag{18}$$

A parallel translation of $\overline{\boldsymbol{\xi}}_{\mathbf{x}} \in T_{\mathbf{x}}\mathcal{S}_r^{p,q}$ along some geodesic $\overline{\gamma}_{\mathbf{x}\to\overline{\boldsymbol{\zeta}}_{\mathbf{x}}} : \mathbb{R} \to \mathcal{S}_r^{p,q}$ is a vector field. For the purpose of notation, we write $\overline{\gamma}$ instead $\overline{\gamma}_{\mathbf{x}\to\overline{\boldsymbol{\zeta}}_{\mathbf{x}}}$ when the indices are not necessary and this vector field satisfies $\overline{\boldsymbol{\xi}}_{\mathbf{x}}(0) = \overline{\boldsymbol{\xi}}_{\mathbf{x}}$ and $\forall t \in \mathbb{R}, \dot{\overline{\boldsymbol{\xi}}}_{\mathbf{x}}(t) := \frac{\mathrm{D}}{dt}\left(\overline{\boldsymbol{\xi}}_{\mathbf{x}}(t)\right) \in \mathrm{SN}_{\overline{\gamma}(t)}(\mathcal{S}_r^{p,q}, \mathbb{R}^{p+1,q})$ where $\frac{\mathrm{D}}{dt}\left(\overline{\boldsymbol{\xi}}_{\mathbf{x}}(t)\right)$ is the covariant derivative of $\overline{\boldsymbol{\xi}}_{\mathbf{x}}(t)$ along $\overline{\gamma}(t)$ in the ambient space $\mathbb{R}^{p+1,q}$.

By definition of the parallel transport, we have $\forall t, \overline{\boldsymbol{\xi}}_{\mathbf{x}}(t) \in T_{\overline{\gamma}(t)}\mathcal{S}_r^{p,q}$, which implies:

$$\forall t, \langle \overline{\boldsymbol{\xi}}_{\mathbf{x}}(t),\overline{\gamma}(t)\rangle_q = 0 \implies \langle \dot{\overline{\boldsymbol{\xi}}}_{\mathbf{x}}(t),\overline{\gamma}(t)\rangle_q = -\langle \overline{\boldsymbol{\xi}}_{\mathbf{x}}(t),\overline{\gamma}'(t)\rangle_q \quad \text{(obtained by differentiating)} \tag{19}$$

By definition, we have $\forall t, \dot{\overline{\boldsymbol{\xi}}}_{\mathbf{x}}(t) \in \mathrm{SN}_{\overline{\gamma}(t)}(\mathcal{S}_r^{p,q}, \mathbb{R}^{p+1,q})$ and $\forall t, \frac{1}{r^2}\langle \overline{\gamma}(t),\overline{\gamma}(t)\rangle_q = 1$, which implies $\forall t, \langle \dot{\overline{\boldsymbol{\xi}}}_{\mathbf{x}}(t),\overline{\gamma}(t)\rangle_q = \frac{1}{r^2}\langle \dot{\overline{\boldsymbol{\xi}}}_{\mathbf{x}}(t),\overline{\gamma}(t)\rangle_q\langle \overline{\gamma}(t),\overline{\gamma}(t)\rangle_q$ and we have:

$$\forall t, \dot{\overline{\boldsymbol{\xi}}}_{\mathbf{x}}(t) = \frac{1}{r^2}\langle \dot{\overline{\boldsymbol{\xi}}}_{\mathbf{x}}(t),\overline{\gamma}(t)\rangle_q\overline{\gamma}(t) = -\frac{1}{r^2}\langle \overline{\boldsymbol{\xi}}_{\mathbf{x}}(t),\overline{\gamma}'(t)\rangle_q\overline{\gamma}(t) \tag{20}$$

Since parallel translation preserves the metric, we have $\forall t, \langle \bar{\xi}_{\mathbf{x}}(t), \overline{\gamma}'(t) \rangle_q = \langle \bar{\xi}_{\mathbf{x}}(0), \overline{\gamma}'(0) \rangle_q = \langle \bar{\xi}_{\mathbf{x}}, \overline{\zeta}_{\mathbf{x}} \rangle_q$. By using the initial condition $\bar{\xi}_{\mathbf{x}}(0) = \bar{\xi}_{\mathbf{x}}$ and integrating $\dot{\bar{\xi}}_{\mathbf{x}}(t) = -\frac{1}{r^2}\langle \bar{\xi}_{\mathbf{x}}, \overline{\zeta}_{\mathbf{x}} \rangle_q \overline{\gamma}(t)$, the parallel transport of $\bar{\xi}_{\mathbf{x}}$ along $\overline{\gamma}_{\mathbf{x}\to\overline{\zeta}_{\mathbf{x}}}(t)$ is:

$$\bar{\xi}_{\mathbf{x}}(t) := \bar{\xi}_{\mathbf{x}} - \frac{1}{r^2}\langle \bar{\xi}_{\mathbf{x}}, \overline{\zeta}_{\mathbf{x}} \rangle_q \int_0^t \overline{\gamma}_{\mathbf{x}\to\overline{\zeta}_{\mathbf{x}}}(\tau)d\tau \tag{21}$$

We have three cases to consider:

(1) If $\langle \overline{\zeta}_{\mathbf{x}}, \overline{\zeta}_{\mathbf{x}} \rangle_q = 0$, we find (see Eq. (13)):

$$\bar{\xi}_{\mathbf{x}}(t) = \bar{\xi}_{\mathbf{x}} - \frac{1}{r^2}\langle \bar{\xi}_{\mathbf{x}}, \overline{\zeta}_{\mathbf{x}} \rangle_q (t\mathbf{x} + \frac{1}{2}t^2\overline{\zeta}_{\mathbf{x}}) \tag{22}$$

By setting $t = 1$ and $\overline{\zeta}_{\mathbf{x}} = \overline{\log}_{\mathbf{x}}(\mathbf{y})$ (i.e., , $\langle \mathbf{x}, \mathbf{y} \rangle_q = r^2$ since $\langle \overline{\zeta}_{\mathbf{x}}, \overline{\zeta}_{\mathbf{x}} \rangle_q = 0$), we have:

$$\bar{\xi}_{\mathbf{x}}(1) = \bar{\xi}_{\mathbf{x}} - \frac{1}{r^2}\langle \bar{\xi}_{\mathbf{x}}, \mathbf{y} - \mathbf{x} \rangle_q \left( \mathbf{x} + \frac{1}{2}(\mathbf{y} - \mathbf{x}) \right) = \bar{\xi}_{\mathbf{x}} - \frac{1}{2r^2}\langle \bar{\xi}_{\mathbf{x}}, \mathbf{y} \rangle_q (\mathbf{x} + \mathbf{y}) \tag{23}$$

$$= \bar{\xi}_{\mathbf{x}} - \frac{\langle \mathbf{y}, \bar{\xi}_{\mathbf{x}} \rangle_q}{\langle \mathbf{x}, \mathbf{y} \rangle_q + r^2}(\mathbf{y} + \mathbf{x}) \tag{24}$$

(2) If $\langle \overline{\zeta}_{\mathbf{x}}, \overline{\zeta}_{\mathbf{x}} \rangle_q > 0$, we find:

$$\bar{\xi}_{\mathbf{x}}(t) = \bar{\xi}_{\mathbf{x}} - \frac{\langle \bar{\xi}_{\mathbf{x}}, \overline{\zeta}_{\mathbf{x}} \rangle_q}{r\sqrt{|\langle \overline{\zeta}_{\mathbf{x}}, \overline{\zeta}_{\mathbf{x}} \rangle_q|}}\left( \sin\left( \frac{t\sqrt{|\langle \overline{\zeta}_{\mathbf{x}}, \overline{\zeta}_{\mathbf{x}} \rangle_q|}}{r} \right)\mathbf{x} + \frac{r}{\sqrt{|\langle \overline{\zeta}_{\mathbf{x}}, \overline{\zeta}_{\mathbf{x}} \rangle_q|}}\left( 1 - \cos\left( \frac{t\sqrt{|\langle \overline{\zeta}_{\mathbf{x}}, \overline{\zeta}_{\mathbf{x}} \rangle_q|}}{r} \right) \right)\overline{\zeta}_{\mathbf{x}} \right)$$

By setting $t = 1$ and $\overline{\zeta}_{\mathbf{x}} = \overline{\log}_{\mathbf{x}}(\mathbf{y})$ (i.e., , $\langle \mathbf{x}, \mathbf{y} \rangle_q \in (-r^2, r^2)$), and using the fact that $\sin(\cos^{-1}(x)) = \sqrt{1 - x^2}$, we find:

$$\bar{\xi}_{\mathbf{x}}(1) = \bar{\xi}_{\mathbf{x}} - \frac{\langle \bar{\xi}_{\mathbf{x}}, \mathbf{y} \rangle_q}{r^2\sqrt{1 - (\frac{\langle \mathbf{x}, \mathbf{y} \rangle_q}{r^2})^2}}\left( \left( \sqrt{1 - (\frac{\langle \mathbf{x}, \mathbf{y} \rangle_q}{r^2})^2} \right)\mathbf{x} + \frac{1 - \frac{\langle \mathbf{x}, \mathbf{y} \rangle_q}{r^2}}{\sqrt{1 - (\frac{\langle \mathbf{x}, \mathbf{y} \rangle_q}{r^2})^2}}\left( \mathbf{y} - \frac{\langle \mathbf{x}, \mathbf{y} \rangle_q}{r^2}\mathbf{x} \right) \right) \tag{25}$$

$$= \bar{\xi}_{\mathbf{x}} - \frac{\langle \bar{\xi}_{\mathbf{x}}, \mathbf{y} \rangle_q}{r^2(1 - (\frac{\langle \mathbf{x}, \mathbf{y} \rangle_q}{r^2})^2)}\left( (1 - (\frac{\langle \mathbf{x}, \mathbf{y} \rangle_q}{r^2})^2)\mathbf{x} + (1 - \frac{\langle \mathbf{x}, \mathbf{y} \rangle_q}{r^2})\left( \mathbf{y} - \frac{\langle \mathbf{x}, \mathbf{y} \rangle_q}{r^2}\mathbf{x} \right) \right) \tag{26}$$

$$= \bar{\xi}_{\mathbf{x}} - \frac{\langle \bar{\xi}_{\mathbf{x}}, \mathbf{y} \rangle_q}{r^2(1 + \frac{\langle \mathbf{x}, \mathbf{y} \rangle_q}{r^2})}\left( (1 + \frac{\langle \mathbf{x}, \mathbf{y} \rangle_q}{r^2})\mathbf{x} + \left( \mathbf{y} - \frac{\langle \mathbf{x}, \mathbf{y} \rangle_q}{r^2}\mathbf{x} \right) \right) \tag{27}$$

$$= \bar{\xi}_{\mathbf{x}} - \frac{\langle \mathbf{y}, \bar{\xi}_{\mathbf{x}} \rangle_q}{\langle \mathbf{x}, \mathbf{y} \rangle_q + r^2}(\mathbf{y} + \mathbf{x}) \tag{28}$$

(3) If $\langle \overline{\zeta}_{\mathbf{x}}, \overline{\zeta}_{\mathbf{x}} \rangle_q < 0$, we find:

$$\bar{\xi}_{\mathbf{x}}(t) = \bar{\xi}_{\mathbf{x}} - \frac{\langle \bar{\xi}_{\mathbf{x}}, \overline{\zeta}_{\mathbf{x}} \rangle_q}{r\sqrt{|\langle \overline{\zeta}_{\mathbf{x}}, \overline{\zeta}_{\mathbf{x}} \rangle_q|}}\left( \sinh\left( \frac{t\sqrt{|\langle \overline{\zeta}_{\mathbf{x}}, \overline{\zeta}_{\mathbf{x}} \rangle_q|}}{r} \right)\mathbf{x} + \frac{r}{\sqrt{|\langle \overline{\zeta}_{\mathbf{x}}, \overline{\zeta}_{\mathbf{x}} \rangle_q|}}\left( \cosh\left( \frac{t\sqrt{|\langle \overline{\zeta}_{\mathbf{x}}, \overline{\zeta}_{\mathbf{x}} \rangle_q|}}{r} \right) - 1 \right)\overline{\zeta}_{\mathbf{x}} \right)$$

By setting $t = 1$ and $\overline{\zeta}_{\mathbf{x}} = \overline{\log}_{\mathbf{x}}(\mathbf{y})$ (i.e., , $\langle \mathbf{x}, \mathbf{y} \rangle_q > r^2$), and using the fact that $\sinh(\cosh^{-1}(x)) = \sqrt{x^2 - 1}$, we find:

$$\overline{\boldsymbol{\xi}}_{\mathbf{x}}(1) = \overline{\boldsymbol{\xi}}_{\mathbf{x}} - \frac{\langle \overline{\boldsymbol{\xi}}_{\mathbf{x}}, \mathbf{y} \rangle_q}{r^2 \sqrt{(\frac{\langle \mathbf{x}, \mathbf{y} \rangle_q}{r^2})^2 - 1}} \left( \left( \sqrt{(\frac{\langle \mathbf{x}, \mathbf{y} \rangle_q}{r^2})^2 - 1} \right) \mathbf{x} + \frac{\frac{\langle \mathbf{x}, \mathbf{y} \rangle_q}{r^2} - 1}{\sqrt{(\frac{\langle \mathbf{x}, \mathbf{y} \rangle_q}{r^2})^2 - 1}} \left( \mathbf{y} - \frac{\langle \mathbf{x}, \mathbf{y} \rangle_q}{r^2} \mathbf{x} \right) \right)$$
(29)

$$= \overline{\boldsymbol{\xi}}_{\mathbf{x}} - \frac{\langle \overline{\boldsymbol{\xi}}_{\mathbf{x}}, \mathbf{y} \rangle_q}{r^2 ((\frac{\langle \mathbf{x}, \mathbf{y} \rangle_q}{r^2})^2 - 1)} \left( ((\frac{\langle \mathbf{x}, \mathbf{y} \rangle_q}{r^2})^2 - 1) \mathbf{x} + (\frac{\langle \mathbf{x}, \mathbf{y} \rangle_q}{r^2} - 1) \left( \mathbf{y} - \frac{\langle \mathbf{x}, \mathbf{y} \rangle_q}{r^2} \mathbf{x} \right) \right)$$
(30)

$$= \overline{\boldsymbol{\xi}}_{\mathbf{x}} - \frac{\langle \overline{\boldsymbol{\xi}}_{\mathbf{x}}, \mathbf{y} \rangle_q}{r^2 (1 + \frac{\langle \mathbf{x}, \mathbf{y} \rangle_q}{r^2})} \left( (1 + \frac{\langle \mathbf{x}, \mathbf{y} \rangle_q}{r^2}) \mathbf{x} + \left( \mathbf{y} - \frac{\langle \mathbf{x}, \mathbf{y} \rangle_q}{r^2} \mathbf{x} \right) \right)$$
(31)

$$= \overline{\boldsymbol{\xi}}_{\mathbf{x}} - \frac{\langle \mathbf{y}, \overline{\boldsymbol{\xi}}_{\mathbf{x}} \rangle_q}{\langle \mathbf{x}, \mathbf{y} \rangle_q + r^2} (\mathbf{y} + \mathbf{x})$$
(32)

In all cases, we define $P_{\mathbf{x} \curvearrowright \mathbf{y}}^{\overline{\gamma}}(\overline{\boldsymbol{\xi}}_{\mathbf{x}}) := \overline{\boldsymbol{\xi}}_{\mathbf{x}}(1)$ as formulated in Eq. (6). The parallel translation $P_{\mathbf{x} \curvearrowright \mathbf{y}}^{\overline{\gamma}}(\overline{\boldsymbol{\xi}}_{\mathbf{x}})$ is then performed along the **minimizing** geodesic $\overline{\gamma}_{\mathbf{x} \to \overline{\zeta}_{\mathbf{x}}}$ defined such that $\overline{\gamma}_{\mathbf{x} \to \overline{\zeta}_{\mathbf{x}}}(1) = \mathbf{y}$ (i.e., $\overline{\exp}_{\mathbf{x}}^{-1}(\mathbf{y}) = \overline{\zeta}_{\mathbf{x}}$) and $\overline{\gamma}'_{\mathbf{x} \to \overline{\zeta}_{\mathbf{x}}}(1) = P_{\mathbf{x} \curvearrowright \mathbf{y}}^{\overline{\gamma}}(\overline{\boldsymbol{\xi}}_{\mathbf{x}})$.

One can also verify that we have:

$$\forall \overline{\boldsymbol{\xi}}_{\mathbf{x}} \in T_{\mathbf{x}} \mathcal{S}_r^{p,q}, \ P_{\mathbf{y} \curvearrowright \mathbf{x}}^{\overline{\gamma}} \left( P_{\mathbf{x} \curvearrowright \mathbf{y}}^{\overline{\gamma}}(\overline{\boldsymbol{\xi}}_{\mathbf{x}}) \right) = \overline{\boldsymbol{\xi}}_{\mathbf{x}} \text{ if } \langle \mathbf{x}, \mathbf{y} \rangle_q > -r^2.$$
(33)

**Nonexistence of (unbroken) geodesic joining pairs of points.** $\mathbf{x} \in \mathcal{S}_r^{p,q}$ and $\mathbf{y} \in \mathcal{S}_r^{p,q}$ are joined by a geodesic iff $\langle \mathbf{x}, \mathbf{y} \rangle_q > -r^2$ or $\mathbf{y} = -\mathbf{x}$. A proof can be found in Appendix C.2 of [15] for the pseudo-hyperboloid $\mathcal{Q}_r^{q,p}$ that is anti-isometric to $\mathcal{S}_r^{p,q}$ as explained in Section B.4.

## B.2 Indefinite elliptic space $\mathcal{P}_r^{p,q}$

The differential geometry tools of $\mathcal{P}_r^{p,q}$ depend on those of the pseudo-sphere described above. We recall that the canonical map $\pi : \mathcal{S}_r^{p,q} \to \mathcal{P}_r^{p,q}$ is defined as $\forall \mathbf{x} \in \mathcal{S}_r^{p,q}, \pi(\mathbf{x}) := [\mathbf{x}] = \{\mathbf{x}, -\mathbf{x}\}$.

### B.2.1 Geodesic, exponential map and distance

**Geodesic.** By using the notation of the main paper, we recall that $\gamma = \pi \circ \overline{\gamma}$ and:

$$\forall \mathbf{x} \in \mathcal{S}_r^{p,q}, \boldsymbol{\xi} \in T_{[\mathbf{x}]} \mathcal{P}_r^{p,q}, \ \overline{\boldsymbol{\xi}}_{\mathbf{x}} = \text{lift}_{\mathbf{x}}(\boldsymbol{\xi}) = -\text{lift}_{-\mathbf{x}}(\boldsymbol{\xi}) = -\overline{\boldsymbol{\xi}}_{-\mathbf{x}}$$
(34)

and we have for all $t \in \mathbb{R}$, $\gamma_{[\mathbf{x}] \to \boldsymbol{\xi}}(t) = \{\overline{\gamma}_{\mathbf{x} \to \overline{\boldsymbol{\xi}}_{\mathbf{x}}}(t), \overline{\gamma}_{-\mathbf{x} \to \overline{\boldsymbol{\xi}}_{-\mathbf{x}}}(t)\}$.

**Exponential map.** The exponential map $\exp_{[\mathbf{x}]} : T_{[\mathbf{x}]} \mathcal{P}_r^{p,q} \to \mathcal{P}_r^{p,q}$ is defined such that

$$\exp_{[\mathbf{x}]}(\boldsymbol{\xi}) := \gamma_{[\mathbf{x}] \to \boldsymbol{\xi}}(1) = \{\overline{\gamma}_{\mathbf{x} \to \overline{\boldsymbol{\xi}}_{\mathbf{x}}}(1), \overline{\gamma}_{-\mathbf{x} \to \overline{\boldsymbol{\xi}}_{-\mathbf{x}}}(1)\} = [\overline{\exp}_{\mathbf{x}}(\overline{\boldsymbol{\xi}}_{\mathbf{x}})].$$
(35)

**Logarithm map.** $\log_{[\mathbf{x}]} := \exp_{[\mathbf{x}]}^{-1}$ is the inverse function of the exponential map. We can write:

$$\text{lift}_{\mathbf{x}} \left( \log_{[\mathbf{x}]}([\mathbf{y}]) \right) = \begin{cases} \overline{\log}_{\mathbf{x}}(\mathbf{y}) & \text{if } \langle \mathbf{x}, \mathbf{y} \rangle_q > 0 \\ \overline{\log}_{\mathbf{x}}(-\mathbf{y}) & \text{if } \langle \mathbf{x}, \mathbf{y} \rangle_q < 0 \end{cases}$$
(36)

where $\overline{\log}_{\mathbf{x}}$ is defined in Eq. (15). In theory, $\log_{[\mathbf{x}]}$ is not defined if $\langle \mathbf{x}, \mathbf{y} \rangle_q = 0$ because there exist two minimizing geodesics. In practice, we consider that its lift equals $\overline{\log}_{\mathbf{x}}(\mathbf{y})$ if $\langle \mathbf{x}, \mathbf{y} \rangle_q = 0$.

**Geodesic distance.** As stated in the paper, the geodesic distance $\text{d}_\gamma(\cdot, \cdot)$ is then formulated:

$$\forall [\mathbf{x}] \in \mathcal{P}_r^{p,q}, [\mathbf{y}] \in \mathcal{P}_r^{p,q}, \ \text{d}_\gamma([\mathbf{x}], [\mathbf{y}]) = \begin{cases} r \cosh^{-1}(|\frac{\langle \mathbf{x}, \mathbf{y} \rangle_q}{r^2}|) & \text{if } |\frac{\langle \mathbf{x}, \mathbf{y} \rangle_q}{r^2}| \geq 1 \\ r \cos^{-1}(|\frac{\langle \mathbf{x}, \mathbf{y} \rangle_q}{r^2}|) & \text{otherwise.} \end{cases}$$
(37)

It satisfies $\text{d}_\gamma([\mathbf{x}], [\mathbf{y}]) = \min\{\overline{\text{d}}_{\overline{\gamma}}(\mathbf{x}, \mathbf{y}), \overline{\text{d}}_{\overline{\gamma}}(-\mathbf{x}, \mathbf{y})\}$.

### B.2.2 $\mathcal{P}_r^{p,q}$ **is a quotient manifold**

We now explain why $\mathcal{P}_r^{p,q} := \mathcal{S}_r^{p,q}/\pm 1 = \mathcal{S}_r^{p,q}/\pm \mathbf{I}$ is a quotient manifold. The explanation is based on the "Orbit manifolds" section of Chapter 7 of [21] (see also page 192 of [21]). We first recall its Definition 6 and Proposition 7.

**Definition B.2.1** (Definition 6 of Chapter 7 of [21]). A group $\Gamma$ of diffeomorphisms of a manifold $\mathcal{M}$ is properly discontinuous (and acts freely) provided:

(PD1) Each point $\mathbf{x} \in \mathcal{M}$ has a neighborhood $\mathcal{A}$ such that if $\phi(\mathcal{A})$ meets $\mathcal{A}$ for $\phi \in \Gamma$ then $\phi =$ id.

(PD2) Points $\mathbf{x}, \mathbf{z} \in \mathcal{M}$ not in the same orbit have neighborhoods $\mathcal{A}$ and $\mathcal{B}$ such that for every $\phi \in \Gamma$, $\phi(\mathcal{A})$ and $\mathcal{B}$ are disjoint.

**Proposition B.2.1** (Proposition 7 of Chapter 7 of [21]). Let $\Gamma$ be a properly discontinuous group of diffeomorphisms of a manifold $\mathcal{M}$. There is a unique way to make $\mathcal{M}/\Gamma$ a manifold so that the natural map $\pi : \mathcal{M} \to \mathcal{M}/\Gamma$ is a covering map.

In our case, we have $\mathcal{M} = \mathcal{S}_r^{p,q}$, and $\Gamma = \pm 1 = \pm \mathbf{I}$ is a group of diffeomorphisms of $\mathcal{S}_r^{p,q}$. To be more precise, $\Gamma$ is composed of the identity map $\mathbf{x} \mapsto \mathbf{x}$ and the antipodal map $\mathbf{x} \mapsto -\mathbf{x}$. For all $\mathbf{x} \in \mathcal{S}_r^{p,q}$, the set $\{\phi(\mathbf{x}) : \phi \in \Gamma\} = [\mathbf{x}] = \{-\mathbf{x}, \mathbf{x}\}$ is called the orbit of $\mathbf{x}$ under $\Gamma$. The collection of all such orbits is our set $\mathcal{P}_r^{p,q} := \mathcal{S}_r^{p,q}/\pm 1$.

(PD1) is satisfied when the neighborhood $\mathcal{A}$ of $\mathbf{x} \in \mathcal{S}_r^{p,q}$ is defined as $\mathcal{A} = \{\mathbf{y} \in \mathcal{S}_r^{p,q} : \langle \mathbf{x}, \mathbf{y} \rangle_q > 0\}$.

(PD2) is satisfied when $\mathbf{z} \neq \pm \mathbf{x}$ by determining some neighborhood small enough for both $\mathbf{z}$ and $\pm \mathbf{x}$ so that they are disjoint.

By definition, each point $\mathbf{x} \in \mathcal{S}_r^{p,q}$ has a connected neighborhood $\mathcal{A} = \{\mathbf{y} \in \mathcal{S}_r^{p,q} : \langle \mathbf{x}, \mathbf{y} \rangle_q > 0\}$ that is evenly covered by $\pi$ since it maps each component of $\pi^{-1}(\mathcal{A})$ diffeomorphically onto $\mathcal{V}$ (see Definition 7 of Chapter A of [21]). It is then a covering map and $\mathcal{P}_r^{p,q}$ is a quotient manifold.

It is worth noting that $\mathcal{P}_r^{p,q}$ is briefly mentioned in page 214 of [21]. It is also called an *indefinite elliptic space* and defined in Equation (12.2.2a) of [30], and the Riemannian case of elliptic geometry is briefly explained in page 74 of [30].

### B.3 Pseudo-hyperboloid $\mathcal{Q}_r^{q,p}$

We recall here the differential geometry tools (from [15]) specific to the pseudo-hyperboloid which is defined as the following set: $\mathcal{Q}_r^{q,p} := \{\mathbf{x} \in \mathbb{R}^{q,p+1} : \langle \mathbf{x}, \mathbf{x} \rangle_{p+1} = -r^2\}$.

**Geodesic.** The geodesic $\overline{\gamma}_{\mathbf{x} \to \overline{\boldsymbol{\xi}}_{\mathbf{x}}} : \mathbb{R} \to \mathcal{Q}_r^{q,p}$ satisfying $\overline{\gamma}_{\mathbf{x} \to \overline{\boldsymbol{\xi}}_{\mathbf{x}}}(0) = \mathbf{x}$ and $\overline{\gamma}'_{\mathbf{x} \to \overline{\boldsymbol{\xi}}_{\mathbf{x}}}(0) = \overline{\boldsymbol{\xi}}_{\mathbf{x}} \in T_{\mathbf{x}}\mathcal{Q}_r^{q,p}$ is formulated for all $t \in \mathbb{R}$:

$$
\overline{\gamma}_{\mathbf{x} \to \overline{\boldsymbol{\xi}}_{\mathbf{x}}}(t) = \begin{cases} \cos\left(\frac{t\sqrt{|\langle \overline{\boldsymbol{\xi}}_{\mathbf{x}}, \overline{\boldsymbol{\xi}}_{\mathbf{x}} \rangle_{p+1}|}}{r}\right) \mathbf{x} + \frac{r}{\sqrt{|\langle \overline{\boldsymbol{\xi}}_{\mathbf{x}}, \overline{\boldsymbol{\xi}}_{\mathbf{x}} \rangle_{p+1}|}} \sin\left(\frac{t\sqrt{|\langle \overline{\boldsymbol{\xi}}_{\mathbf{x}}, \overline{\boldsymbol{\xi}}_{\mathbf{x}} \rangle_{p+1}|}}{r}\right) \overline{\boldsymbol{\xi}}_{\mathbf{x}} & \text{if } \langle \overline{\boldsymbol{\xi}}_{\mathbf{x}}, \overline{\boldsymbol{\xi}}_{\mathbf{x}} \rangle_{p+1} < 0 \\ \mathbf{x} + t\overline{\boldsymbol{\xi}}_{\mathbf{x}} & \text{if } \langle \overline{\boldsymbol{\xi}}_{\mathbf{x}}, \overline{\boldsymbol{\xi}}_{\mathbf{x}} \rangle_{p+1} = 0 \\ \cosh\left(\frac{t\sqrt{|\langle \overline{\boldsymbol{\xi}}_{\mathbf{x}}, \overline{\boldsymbol{\xi}}_{\mathbf{x}} \rangle_{p+1}|}}{r}\right) \mathbf{x} + \frac{r}{\sqrt{|\langle \overline{\boldsymbol{\xi}}_{\mathbf{x}}, \overline{\boldsymbol{\xi}}_{\mathbf{x}} \rangle_{p+1}|}} \sinh\left(\frac{t\sqrt{|\langle \overline{\boldsymbol{\xi}}_{\mathbf{x}}, \overline{\boldsymbol{\xi}}_{\mathbf{x}} \rangle_{p+1}|}}{r}\right) \overline{\boldsymbol{\xi}}_{\mathbf{x}} & \text{if } \langle \overline{\boldsymbol{\xi}}_{\mathbf{x}}, \overline{\boldsymbol{\xi}}_{\mathbf{x}} \rangle_{p+1} > 0 \end{cases}
$$
$$(38)$$

**Exponential map.** The exponential map $\overline{\exp}_{\mathbf{x}} : T_{\mathbf{x}}\mathcal{Q}_r^{q,p} \to \mathcal{Q}_r^{q,p}$ is defined such that $\forall \overline{\boldsymbol{\xi}}_{\mathbf{x}} \in T_{\mathbf{x}}\mathcal{Q}_r^{q,p}, \overline{\exp}_{\mathbf{x}}(\overline{\boldsymbol{\xi}}_{\mathbf{x}}) = \overline{\gamma}_{\mathbf{x} \to \overline{\boldsymbol{\xi}}_{\mathbf{x}}}(1)$. We then have:

$$
\overline{\exp}_{\mathbf{x}}(\overline{\boldsymbol{\xi}}_{\mathbf{x}}) = \begin{cases} \cos\left(\frac{\sqrt{|\langle \overline{\boldsymbol{\xi}}_{\mathbf{x}}, \overline{\boldsymbol{\xi}}_{\mathbf{x}} \rangle_{p+1}|}}{r}\right) \mathbf{x} + \frac{r}{\sqrt{|\langle \overline{\boldsymbol{\xi}}_{\mathbf{x}}, \overline{\boldsymbol{\xi}}_{\mathbf{x}} \rangle_{p+1}|}} \sin\left(\frac{\sqrt{|\langle \overline{\boldsymbol{\xi}}_{\mathbf{x}}, \overline{\boldsymbol{\xi}}_{\mathbf{x}} \rangle_{p+1}|}}{r}\right) \overline{\boldsymbol{\xi}}_{\mathbf{x}} & \text{if } \langle \overline{\boldsymbol{\xi}}_{\mathbf{x}}, \overline{\boldsymbol{\xi}}_{\mathbf{x}} \rangle_{p+1} < 0 \\ \mathbf{x} + \overline{\boldsymbol{\xi}}_{\mathbf{x}} & \text{if } \langle \overline{\boldsymbol{\xi}}_{\mathbf{x}}, \overline{\boldsymbol{\xi}}_{\mathbf{x}} \rangle_{p+1} = 0 \\ \cosh\left(\frac{\sqrt{|\langle \overline{\boldsymbol{\xi}}_{\mathbf{x}}, \overline{\boldsymbol{\xi}}_{\mathbf{x}} \rangle_{p+1}|}}{r}\right) \mathbf{x} + \frac{r}{\sqrt{|\langle \overline{\boldsymbol{\xi}}_{\mathbf{x}}, \overline{\boldsymbol{\xi}}_{\mathbf{x}} \rangle_{p+1}|}} \sinh\left(\frac{\sqrt{|\langle \overline{\boldsymbol{\xi}}_{\mathbf{x}}, \overline{\boldsymbol{\xi}}_{\mathbf{x}} \rangle_{p+1}|}}{r}\right) \overline{\boldsymbol{\xi}}_{\mathbf{x}} & \text{if } \langle \overline{\boldsymbol{\xi}}_{\mathbf{x}}, \overline{\boldsymbol{\xi}}_{\mathbf{x}} \rangle_{p+1} > 0 \end{cases}
$$
$$(39)$$

**Logarithm map.** The logarithm map $\overline{\log}_{\mathbf{x}}$ is defined as the inverse of the exponential map $\overline{\exp}_{\mathbf{x}}$ on a normal neighborhood of $\mathbf{x} \in \mathcal{Q}_r^{q,p}$ denoted by $\mathcal{U}_{\mathbf{x}} = \{\mathbf{y} \in \mathcal{Q}_r^{q,p} : \frac{\langle \mathbf{x}, \mathbf{y} \rangle_{p+1}}{r^2} < 1\}$. It is then

formulated:

$$\forall \mathbf{y} \in \mathcal{U}_{\mathbf{x}}, \; \overline{\log}_{\mathbf{x}}(\mathbf{y}) = \begin{cases} \dfrac{\cosh^{-1}(-\frac{\langle \mathbf{x}, \mathbf{y} \rangle_{p+1}}{r^2})}{\sqrt{(\frac{\langle \mathbf{x}, \mathbf{y} \rangle_{p+1}}{r^2})^2 - 1}} \left( \mathbf{y} + \dfrac{\langle \mathbf{x}, \mathbf{y} \rangle_{p+1}}{r^2} \mathbf{x} \right) & \text{if } \frac{\langle \mathbf{x}, \mathbf{y} \rangle_{p+1}}{r^2} < -1 \\[1.2em] \mathbf{y} - \mathbf{x} & \text{if } \frac{\langle \mathbf{x}, \mathbf{y} \rangle_{p+1}}{r^2} = -1 \\[1.2em] \dfrac{\cos^{-1}(-\frac{\langle \mathbf{x}, \mathbf{y} \rangle_{p+1}}{r^2})}{\sqrt{1 - (\frac{\langle \mathbf{x}, \mathbf{y} \rangle_{p+1}}{r^2})^2}} \left( \mathbf{y} + \dfrac{\langle \mathbf{x}, \mathbf{y} \rangle_{p+1}}{r^2} \mathbf{x} \right) & \text{if } \frac{\langle \mathbf{x}, \mathbf{y} \rangle_{p+1}}{r^2} \in (-1, 1) \end{cases} \tag{40}$$

**Geodesic "distance".** The geodesic distance $\overline{\mathsf{d}}_{\overline{\gamma}} : \mathcal{Q}_r^{q,p} \times \mathcal{Q}_r^{q,p} \to \mathbb{R}$ is then:

$$\overline{\mathsf{d}}_{\overline{\gamma}}(\mathbf{x}, \mathbf{y}) = \sqrt{|\langle \overline{\log}_{\mathbf{x}}(\mathbf{y}), \overline{\log}_{\mathbf{x}}(\mathbf{y}) \rangle_{p+1}|} = \begin{cases} r \cosh^{-1}(-\frac{\langle \mathbf{x}, \mathbf{y} \rangle_{p+1}}{r^2}) & \text{if } \frac{\langle \mathbf{x}, \mathbf{y} \rangle_{p+1}}{r^2} \leq -1 \\ r \cos^{-1}(-\frac{\langle \mathbf{x}, \mathbf{y} \rangle_{p+1}}{r^2}) & \text{if } \frac{\langle \mathbf{x}, \mathbf{y} \rangle_{p+1}}{r^2} \in (-1, 1) \end{cases} \tag{41}$$

**Parallel transport on $\mathcal{Q}_r^{q,p}$.** The parallel transport connecting $\mathbf{x} \in \mathcal{Q}_r^{q,p}$ to $\mathbf{y} \in \mathcal{Q}_r^{q,p}$ is formulated:

$$P_{\mathbf{x} \curvearrowright \mathbf{y}}^{\overline{\gamma}}(\overline{\boldsymbol{\xi}}_{\mathbf{x}}) := \overline{\boldsymbol{\xi}}_{\mathbf{x}} - \frac{\langle \mathbf{y}, \overline{\boldsymbol{\xi}}_{\mathbf{x}} \rangle_{p+1}}{\langle \mathbf{x}, \mathbf{y} \rangle_{p+1} - r^2} (\mathbf{y} + \mathbf{x}) \quad \text{where} \quad \langle \mathbf{x}, \mathbf{y} \rangle_{p+1} < r^2 \tag{42}$$

### B.4 Anti-isometry between the pseudo-sphere and the pseudo-hyperboloid

We now explain why the pseudo-sphere $\mathcal{S}_r^{p,q} := \{ \mathbf{x} \in \mathbb{R}^{p+1,q} : \langle \mathbf{x}, \mathbf{x} \rangle_q = r^2 \}$ is anti-isometric to the pseudo-hyperboloid $\mathcal{Q}_r^{q,p} := \{ \mathbf{x} \in \mathbb{R}^{q,p+1} : \langle \mathbf{x}, \mathbf{x} \rangle_{p+1} = -r^2 \}$. This can actually be generalized to the anti-isometry between $\mathbb{R}^{p+1,q}$ and $\mathbb{R}^{q,p+1}$ that we describe below.

Let us note the vectors $\mathbf{x} = (x_0, x_1, \ldots, x_{d-1}, x_d)^\top \in \mathbb{R}^{p+1,q}$ and $\mathbf{y} = (y_0, y_1, \ldots, y_{d-1}, y_d)^\top \in \mathbb{R}^{p+1,q}$. We can construct vectors in $\mathbb{R}^{q,p+1}$ that reverse the order of the elements of $\mathbf{x}$ and $\mathbf{y}$. We obtain the following vectors $\mathbf{a} = (x_d, x_{d-1}, \ldots, x_1, x_0)^\top \in \mathbb{R}^{q,p+1}$ and $\mathbf{b} = (y_d, y_{d-1}, \ldots, y_1, y_0)^\top \in \mathbb{R}^{q,p+1}$. By definition of our scalar product in Eq. (1), the anti-isometry between $\mathbb{R}^{p+1,q}$ and $\mathbb{R}^{q,p+1}$ corresponds to:

$$\langle \mathbf{x}, \mathbf{y} \rangle_q = -\langle \mathbf{a}, \mathbf{b} \rangle_{p+1}. \tag{43}$$

For instance, for the hyperboloid, let us assume that $\mathbf{x} = (x_0, x_1, \ldots, x_{d-1}, x_d)^\top \in \mathcal{S}_1^{0,q}$ and we note $\mathbf{a} = (x_d, x_{d-1}, \ldots, x_1, x_0)^\top \in \mathcal{Q}_1^{q,0}$. We find:

$$\langle \mathbf{a}, \mathbf{a} \rangle_1 = -\langle \mathbf{x}, \mathbf{x} \rangle_q = -x_0^2 + \sum_{j=1}^d x_j^2 = -1. \tag{44}$$

### B.5 Explanation of Figure 1

We give the definition of space-like and time-like geodesics in Appendix B.1. We recall that $r = 1$ in the figure.

**Space-like geodesic.** In Figure 1, $\mathbf{x}$ and $\mathbf{y}$ are connected by a space-like geodesic. Therefore, according to Eq. (16), the geodesic distance between $\mathbf{x}$ and $\mathbf{y}$ is $\overline{\mathsf{d}}_{\overline{\gamma}}(\mathbf{x}, \mathbf{y}) = r \cos^{-1}(\frac{\langle \mathbf{x}, \mathbf{y} \rangle_q}{r^2})$ and the geodesic distance between $[\mathbf{x}]$ and $[\mathbf{y}]$ is $\mathsf{d}_{\gamma}([\mathbf{x}], [\mathbf{y}]) = r \cos^{-1}(|\frac{\langle \mathbf{x}, \mathbf{y} \rangle_q}{r^2}|) = \overline{\mathsf{d}}_{\overline{\gamma}}(\mathbf{x}, -\mathbf{y})$.

**Time-like geodesic.** In Figure 1, $\mathbf{x}$ and $\mathbf{z}$ are connected by a time-like geodesic. Therefore, the geodesic distance between $\mathbf{x}$ and $\mathbf{z}$ is $\overline{\mathsf{d}}_{\overline{\gamma}}(\mathbf{x}, \mathbf{z}) = r \cosh^{-1}(\frac{\langle \mathbf{x}, \mathbf{z} \rangle_q}{r^2})$ and the geodesic distance between $[\mathbf{x}]$ and $[\mathbf{z}]$ is $\mathsf{d}_{\gamma}([\mathbf{x}], [\mathbf{z}]) = r \cos^{-1}(|\frac{\langle \mathbf{x}, \mathbf{z} \rangle_q}{r^2}|) = \overline{\mathsf{d}}_{\overline{\gamma}}(\mathbf{x}, \mathbf{z})$.

**Null geodesic.** For completeness, the geodesic "distance" between two points joined by a null geodesic is 0 even if the two points are distinct.

### B.6 Hyperbolic and elliptic parts of the ultrahyperbolic manifold

In the main paper, we state that $\mathcal{P}_r^{p,q}$ contains hyperbolic and elliptic parts. Our explanation is similar to the one in [15].

● **Elliptic parts.** We first recall that if all the time dimensions of $\mathcal{P}_r^{p,q}$ are set to $0$, then the considered manifold can be written $\mathcal{P}_r^{p,0} \times \{\mathbf{0}\}$ which corresponds to elliptic geometry.

Moreover, in spherical geometry, geodesics are all written in the following way:

$$\overline{\gamma}_{\mathbf{x} \to \overline{\boldsymbol{\xi}}_{\mathbf{x}}}(t) = \cos\left(\frac{t\sqrt{|\langle \overline{\boldsymbol{\xi}}_{\mathbf{x}}, \overline{\boldsymbol{\xi}}_{\mathbf{x}}\rangle|}}{r}\right)\mathbf{x} \; + \frac{r}{\sqrt{|\langle \overline{\boldsymbol{\xi}}_{\mathbf{x}}, \overline{\boldsymbol{\xi}}_{\mathbf{x}}\rangle|}} \sin\left(\frac{t\sqrt{|\langle \overline{\boldsymbol{\xi}}_{\mathbf{x}}, \overline{\boldsymbol{\xi}}_{\mathbf{x}}\rangle|}}{r}\right)\overline{\boldsymbol{\xi}}_{\mathbf{x}} \qquad (45)$$

Their formulation is then very similar to the formulation of our space-like geodesics of Eq. (13) except that a different scalar product is used. In fact, it corresponds to a special case of our scalar product when the number of time dimensions is zero.

● **Hyperbolic parts.** We also recall that if all the space dimensions except one of $\mathcal{P}_r^{p,q}$ are set to $0$, then the considered manifold is diffeomorphic to $\{\mathbf{0}\} \times \mathcal{P}_r^{0,q}$ which corresponds to the hyperboloid model of hyperbolic geometry.

Moreover, in the hyperboloid model of hyperbolic geometry, geodesics are all written:

$$\overline{\gamma}_{\mathbf{x} \to \overline{\boldsymbol{\xi}}_{\mathbf{x}}}(t) = \cosh\left(\frac{t\sqrt{|\langle \overline{\boldsymbol{\xi}}_{\mathbf{x}}, \overline{\boldsymbol{\xi}}_{\mathbf{x}}\rangle_q|}}{r}\right)\mathbf{x} \; + \frac{r}{\sqrt{|\langle \overline{\boldsymbol{\xi}}_{\mathbf{x}}, \overline{\boldsymbol{\xi}}_{\mathbf{x}}\rangle_q|}} \sinh\left(\frac{t\sqrt{|\langle \overline{\boldsymbol{\xi}}_{\mathbf{x}}, \overline{\boldsymbol{\xi}}_{\mathbf{x}}\rangle_q|}}{r}\right)\overline{\boldsymbol{\xi}}_{\mathbf{x}} \qquad (46)$$

Their formulation is then similar to the formulation of our time-like geodesics of Eq. (13) except that a larger number of time dimensions is used in our case.

In conclusion, our proposed geometry is more general and manages to describe relationships considered in elliptic and hyperbolic geometries.

## C  Descent direction and optimization

### C.1  Descent direction of Section 3.4

**Proof.** We provide here the detailed proof that the negative of $\mathbf{G}\boldsymbol{\lambda}_{[\mathbf{x}],\mathbf{p}} \in \mathcal{H}_{\mathbf{p}}$ is a descent direction. We recall that $\mathbf{x} := \overline{\exp}_{\mathbf{p}}(\varphi_\theta(\boldsymbol{x})) \in \mathcal{S}_r^{p,q}$ and $\boldsymbol{\lambda}_{[\mathbf{x}],\mathbf{p}} := \text{lift}_{\mathbf{p}}\left(P_{[\mathbf{x}]\frown[\mathbf{p}]}^{\gamma}(Df([\mathbf{x}]))\right) \in T_{\mathbf{p}}\mathcal{S}_r^{p,q}$.

● We first consider the case where $\langle \mathbf{x}, \mathbf{p}\rangle_q \geq 0$.

Let us consider some tangent vector $\overline{\boldsymbol{\zeta}}_{\mathbf{x}} \in T_{\mathbf{x}}\mathcal{S}_r^{p,q}$ and some point $\mathbf{y} \in \mathcal{S}_r^{p,q}$ defined such that $\overline{f}(\mathbf{y}) = \overline{f} \circ \overline{\gamma}_{\mathbf{x}\to\overline{\zeta}_{\mathbf{x}}}(1)$. By exploiting Taylor's first-order approximation, the function $\overline{f} \circ \overline{\gamma}_{\mathbf{x}\to\overline{\zeta}_{\mathbf{x}}}$ can be approximated at $t = 1$ by:

$$\overline{f}(\mathbf{y}) = \overline{f} \circ \overline{\gamma}_{\mathbf{x}\to\overline{\zeta}_{\mathbf{x}}}(1) \simeq \overline{f} \circ \overline{\gamma}_{\mathbf{x}\to\overline{\zeta}_{\mathbf{x}}}(0) + (\overline{f} \circ \overline{\gamma}_{\mathbf{x}\to\overline{\zeta}_{\mathbf{x}}})'(0) = \overline{f}(\mathbf{x}) + \langle D\overline{f}(\mathbf{x}), \overline{\zeta}_{\mathbf{x}}\rangle_q \qquad (47)$$

where $D\overline{f}(\mathbf{x}) \in T_{\mathbf{x}}\mathcal{S}_r^{p,q}$ is the pseudo-Riemannian gradient of $\overline{f}$ at $\mathbf{x}$ (see Section 4.2 of [15] for details).

Our goal is to determine some tangent vector $\overline{\boldsymbol{\zeta}}_{\mathbf{x}} \in T_{\mathbf{x}}\mathcal{S}_r^{p,q}$ such that it is a descent direction. In other words, we want $\overline{\boldsymbol{\zeta}}_{\mathbf{x}} \in T_{\mathbf{x}}\mathcal{S}_r^{p,q}$ to satisfy $\overline{f}(\mathbf{y}) < \overline{f}(\mathbf{x})$ (i.e., $\langle D\overline{f}(\mathbf{x}), \overline{\zeta}_{\mathbf{x}}\rangle_q < 0$).

We also recall that our neural network $\varphi_\theta$ maps to $T_{\mathbf{p}}\mathcal{S}_r^{p,q}$ but $\overline{\boldsymbol{\zeta}}_{\mathbf{x}}$ lies in $T_{\mathbf{x}}\mathcal{S}_r^{p,q}$, which is a different tangent space if $\mathbf{p} \neq \mathbf{x}$. The parallel transport allows us to work with both tangent spaces. To simplify the notation, we define the following tangent vector:

$$\overline{\boldsymbol{\chi}}_{\mathbf{p}} := \boldsymbol{\lambda}_{[\mathbf{x}],\mathbf{p}} = \text{lift}_{\mathbf{p}}\left(P_{[\mathbf{x}]\frown[\mathbf{p}]}^{\gamma}(Df([\mathbf{x}]))\right) = P_{\mathbf{x}\frown\mathbf{p}}^{\gamma}(D\overline{f}(\mathbf{x})) \in T_{\mathbf{p}}\mathcal{S}_r^{p,q}. \qquad (48)$$

As explained in Section B.1.2, the geodesic $\overline{\gamma}_{\mathbf{p}\to\overline{\chi}_{\mathbf{p}}}$ satisfies the properties $\overline{\gamma}_{\mathbf{p}\to\overline{\chi}_{\mathbf{p}}}(0) = \mathbf{p}$, $\overline{\gamma}'_{\mathbf{p}\to\overline{\chi}_{\mathbf{p}}}(0) = \overline{\chi}_{\mathbf{p}}$ and $\overline{\gamma}'_{\mathbf{p}\to\overline{\chi}_{\mathbf{p}}}(1) = D\overline{f}(\mathbf{x})$. We then have $P_{\mathbf{p}\frown\mathbf{x}}^{\gamma}(\overline{\chi}_{\mathbf{p}}) = D\overline{f}(\mathbf{x})$.

It is worth noting that we also have the following property: $\forall \overline{\boldsymbol{\xi}}_{\mathbf{p}} \in T_{\mathbf{p}}\mathcal{S}_r^{p,q}, \mathbf{G}\overline{\boldsymbol{\xi}}_{\mathbf{p}} \in T_{\mathbf{p}}\mathcal{S}_r^{p,q}$. Let us then define $\overline{\boldsymbol{\zeta}}_{\mathbf{x}}$ such that $\overline{\boldsymbol{\zeta}}_{\mathbf{x}} := P_{\mathbf{p}\frown\mathbf{x}}^{\gamma}(-\mathbf{G}\boldsymbol{\lambda}_{[\mathbf{x}],\mathbf{p}}) = P_{\mathbf{p}\frown\mathbf{x}}^{\gamma}(-\mathbf{G}\overline{\chi}_{\mathbf{p}})$. From Eq. (33), we know that $-\mathbf{G}\overline{\chi}_{\mathbf{p}} = P_{\mathbf{x}\frown\mathbf{p}}^{\gamma}(\overline{\zeta}_{\mathbf{x}})$ and $\overline{\chi}_{\mathbf{p}} = P_{\mathbf{x}\frown\mathbf{p}}^{\gamma}(D\overline{f}(\mathbf{x}))$.

Due to the linear isometry property of the parallel transport (see page 66 of [21]), we have:

$$\langle D\overline{f}(\mathbf{x}), \overline{\boldsymbol{\zeta}}_{\mathbf{x}}\rangle_q = \langle P^{\overline{\gamma}}_{\mathbf{x}\frown\mathbf{p}}(D\overline{f}(\mathbf{x})), P^{\overline{\gamma}}_{\mathbf{x}\frown\mathbf{p}}(-\mathbf{G}\overline{\boldsymbol{\chi}}_{\mathbf{p}})\rangle_q = \langle \overline{\boldsymbol{\chi}}_{\mathbf{p}}, -\mathbf{G}\overline{\boldsymbol{\chi}}_{\mathbf{p}}\rangle_q = -\|\overline{\boldsymbol{\chi}}_{\mathbf{p}}\|^2 \leq 0 \quad (49)$$

where $\|\cdot\|$ denotes the standard Euclidean norm defined as $\forall\mathbf{x}, \|\mathbf{x}\| := \sqrt{\langle\mathbf{x},\mathbf{x}\rangle}$. Eq. (49) is zero iff $\overline{\boldsymbol{\chi}}_{\mathbf{p}} = \mathbf{0}$, and negative otherwise. It is also worth noting that $\overline{\boldsymbol{\chi}}_{\mathbf{p}} = \mathbf{0}$ iff $D\overline{f}(\mathbf{x}) = \mathbf{0}$ (i.e., $\mathbf{x}$ is a stationary point). This shows that the negative of $\mathbf{G}\boldsymbol{\lambda}_{[\mathbf{x}],\mathbf{p}}$ is a descent direction.

Due to the properties of the exponential map, the differential of the exponential map $\mathrm{d}(\overline{\exp}_{\mathbf{p}})_{\mathbf{0}}$ at the origin $\mathbf{0}$ satisfies the following property:

$$\langle \mathrm{d}(\overline{\exp}_{\mathbf{p}})_{\mathbf{0}}(\overline{\boldsymbol{\chi}}_{\mathbf{p}}), -\mathbf{G}\overline{\boldsymbol{\chi}}_{\mathbf{p}}\rangle_q = \langle \overline{\boldsymbol{\chi}}_{\mathbf{p}}, -\mathbf{G}\overline{\boldsymbol{\chi}}_{\mathbf{p}}\rangle_q = -\|\overline{\boldsymbol{\chi}}_{\mathbf{p}}\|^2 \leq 0 \quad (50)$$

Eq. (50) implies that $-\mathbf{G}\boldsymbol{\lambda}_{[\mathbf{x}],\mathbf{p}}$ is a descent direction of the neural network $\varphi_\theta : \mathcal{X} \to T_{\mathbf{p}}\mathcal{S}^{p,q}_r$.

- The case where $\langle\mathbf{x},\mathbf{p}\rangle_q < 0$ is similar to the case above except that we now have:

$$\overline{f}(\mathbf{y}) = \overline{f} \circ \overline{\gamma}_{-\mathbf{x}\to\overline{\boldsymbol{\zeta}}_{-\mathbf{x}}}(1) \simeq \overline{f}(-\mathbf{x}) + (\overline{f}\circ\overline{\gamma}_{-\mathbf{x}\to\overline{\boldsymbol{\zeta}}_{-\mathbf{x}}})'(0) = \overline{f}(-\mathbf{x}) + \langle D\overline{f}(-\mathbf{x}), \overline{\boldsymbol{\zeta}}_{-\mathbf{x}}\rangle_q \quad (51)$$

$$= \overline{f}(\mathbf{x}) + \langle -D\overline{f}(\mathbf{x}), \overline{\boldsymbol{\zeta}}_{-\mathbf{x}}\rangle_q \quad (52)$$

where $\overline{\boldsymbol{\zeta}}_{-\mathbf{x}} := P^{\overline{\gamma}}_{\mathbf{p}\frown-\mathbf{x}}(-\mathbf{G}\boldsymbol{\lambda}_{[\mathbf{x}],\mathbf{p}}) \in T_{-\mathbf{x}}\mathcal{S}^{p,q}_r$ and $D\overline{f}(-\mathbf{x}) = -D\overline{f}(\mathbf{x}) \in T_{-\mathbf{x}}\mathcal{S}^{p,q}_r$.

We also have $\overline{\boldsymbol{\chi}}_{\mathbf{p}} := \boldsymbol{\lambda}_{[\mathbf{x}],\mathbf{p}} = \mathrm{lift}_{\mathbf{p}}\left(P^{\gamma}_{[\mathbf{x}]\frown[\mathbf{p}]}(Df([\mathbf{x}]))\right) = P^{\overline{\gamma}}_{-\mathbf{x}\frown\mathbf{p}}(-D\overline{f}(\mathbf{x})) \in T_{\mathbf{p}}\mathcal{S}^{p,q}_r$, which implies $\langle -D\overline{f}(\mathbf{x}), \overline{\boldsymbol{\zeta}}_{-\mathbf{x}}\rangle_q = \langle \overline{\boldsymbol{\chi}}_{\mathbf{p}}, -\mathbf{G}\boldsymbol{\lambda}_{[\mathbf{x}],\mathbf{p}}\rangle_q = \langle \overline{\boldsymbol{\chi}}_{\mathbf{p}}, -\mathbf{G}\overline{\boldsymbol{\chi}}_{\mathbf{p}}\rangle_q = -\|\overline{\boldsymbol{\chi}}_{\mathbf{p}}\|^2 \leq 0$.

This completes the proof. $\qquad\square$

### C.2 Optimizing the MLP in the toy experiment

In the toy experiment, we define a new PyTorch autograd function to define the exponential map $\overline{\exp}_{\mathbf{p}}$ as explained in Section 3.4. The custom gradient of our autograd function is $\mathbf{G}\boldsymbol{\lambda}_{[\mathbf{x}],\mathbf{p}}$.

Naïvely using (the negative of) $\mathbf{G}\boldsymbol{\lambda}_{[\mathbf{x}],\mathbf{p}}$ as descent direction and exploiting standard backpropagation decreases the optimized function because all the hidden layers already lie in some space equipped with a positive definite metric tensor [9].

### C.3 Optimizing the Graph Convolutional Network

For the GCN introduced in Section 4, the optimized parameters are the matrices $\mathbf{W}^k$. To be fair with the baselines, we modified the code of Liu *et al.* [18] that is available at the following address: `https://github.com/facebookresearch/hgnn`

We added a Python class for our ultrahyperbolic manifold, it is very similar to the Lorentz manifold Python class. Our code replaces the standard Lorentz inner product (that corresponds to our scalar product in the special case where $p = 0$) used in [18] with our scalar product, its induced exponential/logarithm map and geodesic distance. We also modified the activation function as explained in Section 4. Standard backpropagation is used to train the parameters $\mathbf{W}^k$ that exploit operations over the horizontal space of the positive pole $\mathbf{p}$ as explained in the paper. To have a fair comparison, we used the optimizer of [18] and did not use the optimizer introduced in Section 3.4.

## D Experiments

### D.1 Type of resources used and amount of compute

We ran all our experiments on Zachary's karate club dataset and the node classification task on a machine equipped with a 6-core Intel i7-7800X CPU and NVIDIA GeForce RTX 3090 GPU. The machine was also used to run most of our graph classification experiments.

Since the Reddit-multi-12K dataset requires more than 24GB of VRAM, we ran each experiment of the Reddit dataset on a single 32 GB NVIDIA Tesla V100 GPU of an NVIDIA DGX-1 server. Most experiments take several minutes. Some graph classification experiments take several hours and the longest experiment (one split of Reddit-multi-12k) takes one day.

## D.2 Zachary's karate club dataset

**Parameters and hyperparameters.** We train our framework as explained in the main paper and Section C.2. In practice, we define a new PyTorch autograd function to define the exponential map $\overline{\exp}_{\mathbf{p}}$. The custom gradient of our autograd function is $\mathbf{G}\lambda_{[\mathbf{x}],\mathbf{p}}$. Concerning the choice of hyperparameters (e.g., temperature $\tau$, optimizer and learning rate), we chose the same hyperparameter values as Law & Stam [15]. During training, we use a standard Stochastic Gradient Descent (SGD) optimizer without momentum, with learning rate of $10^{-7}$. We run our experiments for 25,000 iterations (which are also epochs since the dataset is small). We chose a standard MLP with 3 hidden layers to show that our optimizer can be used with neural networks. Other architectures can be used.

**Figure 2 (right) and other illustrative two-dimensional plots.** As a qualitative way to understand the method, we plot two-dimensional representations that were learned for different runs. For all the illustrative figures, we replace the geodesic distance used in Eq. (12) by the squared geodesic distance. It tends to give nicer illustrations.

• We plot two-dimensional projections of learned points lying on the non-Riemannian manifold $\mathcal{P}_1^{1,1}$ in Fig. 2 (right) and Fig 3 (see caption of the figure for details). These projections lie in non-Euclidean space so they should not be interpreted by using standard Euclidean distances. Instead, the figures on the right correspond to spacetime diagrams. As explained in the caption of Fig. 3, points lying on an oblique line have very small distance. Nonetheless, we can see a clear separation between nodes of different factions.

• We plot the same kind of two-dimensional hyperbolic and elliptic representations in Fig. 5 and Fig 4, respectively. Although the separation between factions is clear, the learned node representations do not satisfy the standard structure of a tree or a cycle graph. For instance, high-level nodes of the hierarchy (i.e., nodes $v_1$ and $v_{34}$) do not lie closer to the origin than low-level nodes although this is generally the case when hyperbolic representations are used to learn trees [19, 20].

**Evaluation metrics.** Following the evaluation protocol of [15], we take the capacity matrix $\mathbf{C} \in \mathbb{R}^{n \times n}$ of [33] which defines the level of friendship between the different members. We then consider instead its symmetrized version $\mathbf{S} = \mathbf{C} + \mathbf{C}^\top$. The score $s_i = \sum_{j=1}^{n} \mathbf{S}_{ij}$ defines the importance of the node $v_i$ in the hierarchy. The higher the score, the more important the node is in the hierarchy.

These $s_i$ scores are then used to calculate the Spearman's rank correlation coefficient between the selected $s_i$ scores (top 5 or top 10) and corresponding $\delta_i$ scores. As reported in Table 1, ultrahyperbolic representations are more correlated with the node importance in the hierarchy.

**Other proxy to quantify importance in hyperbolic space.** In machine learning, when hyperbolic embeddings are used to represent hierarchies or trees, a standard way to determine the importance of nodes is to compare the Euclidean norm of the embeddings in the Poincaré ball (or equivalently on the hyperboloid) [19, 20]. High-level nodes tend to have smaller Euclidean norm in hyperbolic geometry. In the first column of Table 6, we report the different scores when the Euclidean norm of the learned hyperbolic representations is used as a proxy of the importance. The second column corresponds to the scores reported in Table 1 of the main paper (i.e., sum of the $\delta_i$ scores).

According to the results in Table 6, the $\ell_2$-norm is a worse indicator of importance than $\delta_i$ scores for this dataset due to the presence of cycles in the graph. This observation is also in accordance with the qualitative two-dimensional results of Fig. 4 where nodes $v_1$ and $v_{34}$ do not lie closer to the origin than other nodes.

Table 6: Evaluation scores for the different learned representations (mean $\pm$ standard deviation)

| Evaluation metric | Hyperbolic with $\ell_2$ norm as proxy | Hyperbolic with $\delta_i$ score as proxy |
|---|---|---|
| Rank of first leader | **2.2 $\pm$ 1.0** | 2.5 $\pm$ 0.7 |
| Rank of second leader | 7.3 $\pm$ 2.4 | **3.8 $\pm$ 1.0** |
| top 5 Spearman's $\rho$ | 0.30 $\pm$ 0.44 | **0.36 $\pm$ 0.22** |
| top 10 Spearman's $\rho$ | 0.22 $\pm$ 0.21 | **0.38 $\pm$ 0.18** |

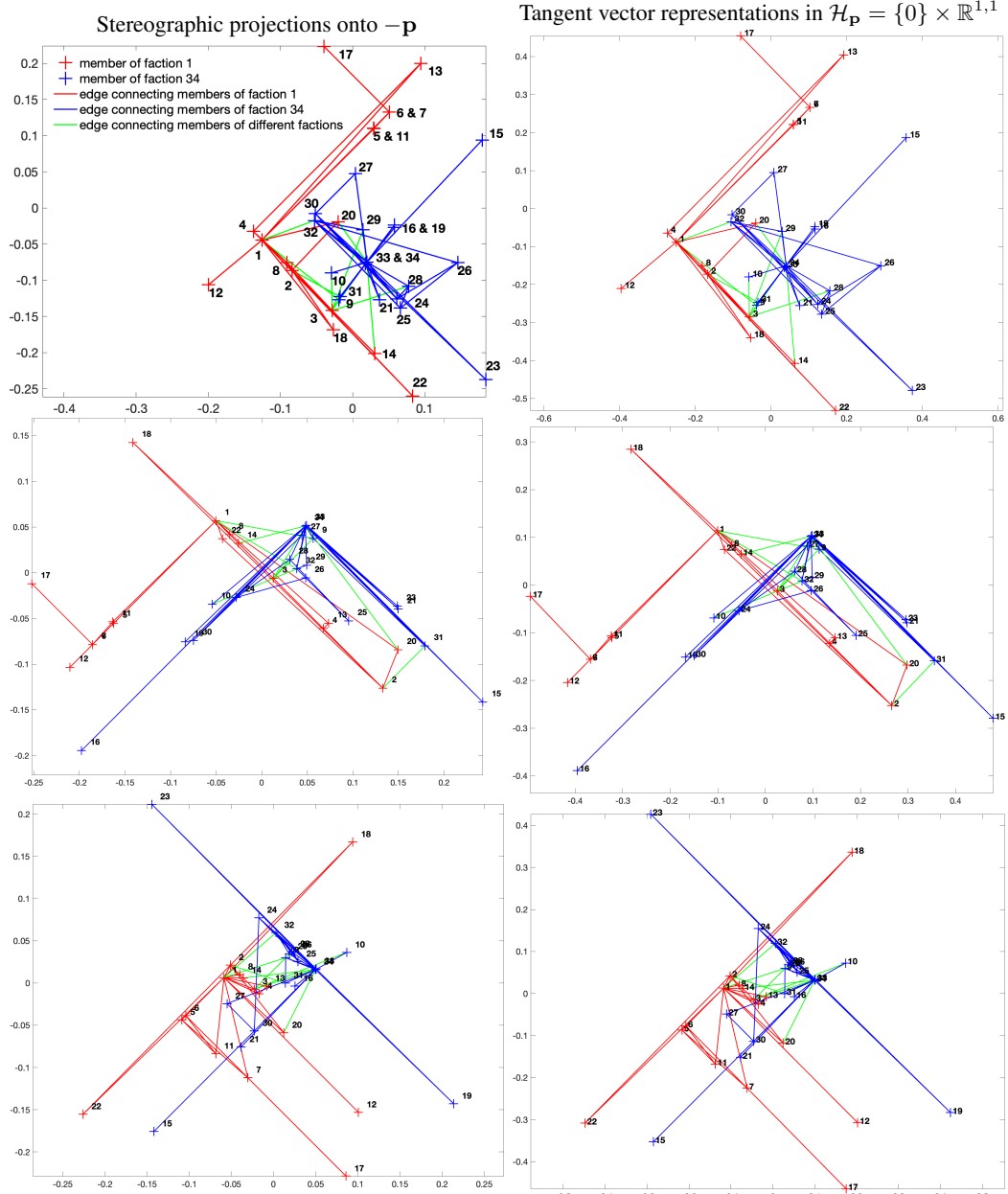

Figure 3: (left) Stereographic projection of learned node representations in $\mathcal{P}_1^{1,1}$ for three different initializations. (right) Tangent vector representations of node representations. For every node representation $[\mathbf{x}_i] \in \mathcal{P}_1^{1,1}$, we plot the last two elements of its tangent vector representation: $\boldsymbol{\xi}_i = \text{lift}_{\mathbf{p}}\left(\log_{[\mathbf{p}]}([\mathbf{x}_i])\right) \in \mathcal{H}_{\mathbf{p}} = \{0\} \times \mathbb{R}^{1,1}$. Tangent vector representations are easier to interpret since they lie in some space diffeomorphic to $\mathbb{R}^{1,1}$. Let us consider two vectors $\mathbf{a} = (a_1, a_2) \in \mathbb{R}^{1,1}$ and $\mathbf{b} = (b_1, b_2) \in \mathbb{R}^{1,1}$. Their distance in $\mathbb{R}^{1,1}$ is $\sqrt{|\langle \mathbf{a} - \mathbf{b}, \mathbf{a} - \mathbf{b}\rangle_q|} = \sqrt{|(a_1 - b_1)^2 - (a_2 - b_2)^2|}$, which explains why similar examples (i.e., connected by an edge) are joined by an oblique line. Their distance in that space is very small and does not follow the intuition of the standard Euclidean distance.

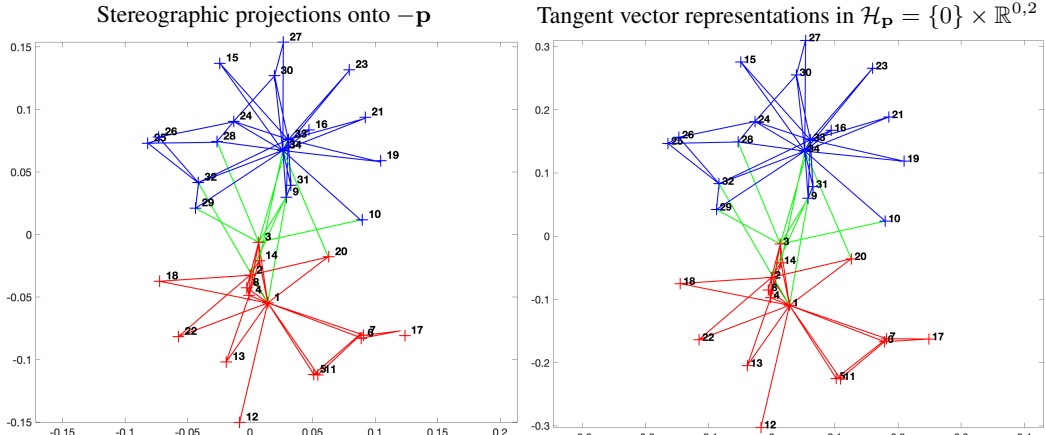

Figure 4: (left) Stereographic projection of learned **hyperbolic** node representations in $\mathcal{P}_1^{0,2}$. In the machine learning literature, they are also called **Poincaré representations**. (right) Tangent vector representations of node representations. For every node representation $[\mathbf{x}_i] \in \mathcal{P}_1^{1,1}$, we plot the last two elements of its tangent vector representation: $\boldsymbol{\xi}_i = \mathrm{lift}_{\mathbf{p}}\left(\log_{[\mathbf{p}]}([\mathbf{x}_i])\right) \in \mathcal{H}_{\mathbf{p}} = \{0\} \times \mathbb{R}^{0,2}$. It is worth noting that, since the represented graph is not a tree, the high-level nodes (i.e., nodes $v_1$ and $v_{34}$) do not have smaller Euclidean norm than other nodes in the hierarchy.

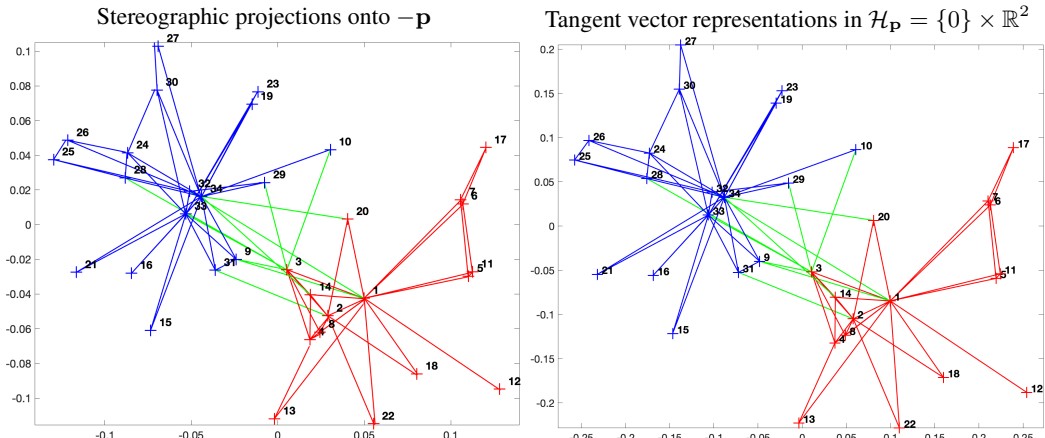

Figure 5: (left) Stereographic projection of learned **elliptic** node representations in $\mathcal{P}_1^{2,0}$. (right) Tangent vector representations of node representations. For every node representation $[\mathbf{x}_i] \in \mathcal{P}_1^{2,0}$, we plot the last two elements of its tangent vector representation: $\boldsymbol{\xi}_i = \mathrm{lift}_{\mathbf{p}}\left(\log_{[\mathbf{p}]}([\mathbf{x}_i])\right) \in \mathcal{H}_{\mathbf{p}} = \{0\} \times \mathbb{R}^2$.

Table 7: Test node classification accuracy with 10-dimensional manifolds

| Dataset | $\mathbb{R}^{10}$ (Euclidean) | $\mathcal{P}_1^{0,10}$ (Hyperbolic) | $\mathcal{P}_1^{1,9}$ | $\mathcal{P}_1^{2,8}$ | $\mathcal{P}_1^{9,1}$ | $\mathcal{P}_1^{10,0}$ (Elliptic) |
|---|---|---|---|---|---|---|
| Citeseer | $58.4 \pm 2.1$ | $56.4 \pm 2.9$ | $\mathbf{62.2 \pm 2.1}$ | $60.9 \pm 2.8$ | $60.4 \pm 3.4$ | $61.3 \pm 2.7$ |
| Cora | $67.8 \pm 4.8$ | $72.6 \pm 2.1$ | $\mathbf{75.1 \pm 1.6}$ | $73.7 \pm 2.3$ | $73.3 \pm 2.7$ | $71.9 \pm 1.9$ |
| Pubmed | $73.1 \pm 2.5$ | $\mathbf{75.3 \pm 1.6}$ | $74.9 \pm 1.9$ | $75.0 \pm 1.0$ | $75.1 \pm 1.3$ | $\mathbf{75.3 \pm 0.8}$ |

Table 8: Test node classification accuracy with 600-dimensional manifolds

| Dataset | $\mathbb{R}^{600}$ | $\mathcal{P}_1^{0,600}$ | $\mathcal{P}_1^{1,599}$ |
|---|---|---|---|
| Citeseer | $70.9 \pm 0.4$ | $70.8 \pm 0.4$ | $70.6 \pm 0.5$ |
| Cora | $81.6 \pm 0.4$ | $81.9 \pm 0.3$ | $82.0 \pm 0.4$ |
| Pubmed | $79.0 \pm 0.5$ | $79.0 \pm 0.8$ | $78.9 \pm 0.8$ |

## D.3 Node and graph classification

We now give details about the experiments of Section 5.2. As explained in Section C.3, for the node and graph classification tasks, we simply adapted the code of Liu *et al.* [18] to the ultrahyperbolic case. We refer the reader to [18] for more details since our experimental protocol is the same.

**Data preprocessing and choice of splits.** To download the datasets, we used the splits extracted from Liu's project page ( `https://github.com/facebookresearch/hgnn` ). The node classification extraction script is `download_node.sh` and the graph classification extraction script is `data_preprocess.py` which provides 10 fixed splits per dataset to perform 10-fold cross validation.

**Prototype-based classification.** Following Section 3 of [18], the output of an ultrahyperbolic neural network with $K$ steps is a set of node representations in ultrahyperbolic space: $\{\mathbf{h}_1^K, \ldots, \mathbf{h}_{|V|}^K\}$ where each $\mathbf{h}_i^K$ lies on the manifold. A list of prototypes (called "centroids" in [18]) is created $\mathcal{C} = \{\mathbf{c}_1, \ldots, \mathbf{c}_{|\mathcal{C}|}\}$ where each $\mathbf{c}_j$ lies on the same manifold as $\mathbf{h}_i^K$. All the prototypes are points, they are learned jointly with the GNN using backpropagation.

A distance matrix $\mathbf{D} \in \mathbb{R}^{|V| \times |\mathcal{C}|}$ defined such that $\mathbf{D}_{ij} = \mathsf{d}(\mathbf{h}_i^K, \mathbf{c}_j)$ is created. In practice, $\mathsf{d}$ is the geodesic distance. It satisfies the properties in Section 3.3 and our optimization framework can be used.

• **Node classification.** Let us note $C$ the number of node classes and $\mathbf{W} \in \mathbb{R}^{|\mathcal{C}| \times C}$ some matrix to be learned. The posterior probability distribution to determine the category of each node is calculated as follows: $\mathbf{Y} = \mathrm{softmax}(\mathbf{DW})$ where the $j$-th element of the $i$-th row of $\mathbf{Y}$ corresponds to the probability that the $i$-th node belongs to the $j$-th category. Cross-entropy is used for learning.

• **Graph classification.** For graph-level predictions, average pooling is first used to combine the distances of different nodes into a single score per node. As done in [18], a fully connected layer is then used with standard cross-entropy to perform graph classification.

**Choice of parameters and hyperparameters.** In the same way as Section D.2 for Zachary's karate club dataset, our code is based on the code of Liu *et al.* [18] as explained in Section C.3. To be fair with the baselines, we take the parameters available at `https://github.com/facebookresearch/hgnn/tree/master/params` that were used for the hyperbolic manifold. We only replace the hyperboloid by $\mathcal{P}_r^{p,q}$, and we adapt the activation function as explained in Section 4.

For instance, for node classification, we use the following parameters: `https://github.com/facebookresearch/hgnn/blob/master/params/NodeClassificationHyperbolicParams.py` (i.e., same optimizer, learning rate, number of prototypes, number of layers etc). We do the same thing for the graph classification task.

**Reported results.** In the tables of results, the baselines "Euclidean", "Poincaré " and "Lorentz" correspond to the implementations of [18]. Liu *et al.* show that their implementation matches the scores of the standard GCN. We did not manage to reproduce their results for the collab and reddit datasets even when we tried different optimizers, learning rates, activation functions, number of centroids. We then reran their code and reported the obtained results in the main paper. We report results when the manifold is 10-dimensional (resp. 600-dimensional) in Table 7 (resp. Table 8).

Table 9: Evaluation scores for the different learned representations on Zachary's karate club dataset (mean ± standard deviation)

| Manifold | Distance | Rank of first leader | Rank of second leader | top 5 Spearman's $\rho$ | top 10 Spearman's $\rho$ |
|---|---|---|---|---|---|
| $\mathbb{S}^1_{r_1} \times \mathbb{H}^3_{r_2}$ | $\mathsf{d}_{\ell_1}$ | $1.8 \pm 0.5$ | $3.4 \pm 0.7$ | $0.47 \pm 0.25$ | $0.52 \pm 0.13$ |
| $\mathbb{S}^1_{r_1} \times \mathbb{H}^3_{r_2}$ | $\mathsf{d}_{\ell_2}$ | $1.9 \pm 0.8$ | $3.4 \pm 0.9$ | $0.47 \pm 0.20$ | $0.51 \pm 0.18$ |
| $\mathbb{S}^1_{r_1} \times \mathbb{H}^3_{r_2}$ | $\mathsf{d}_{\min}$ | $3.0 \pm 2.3$ | $7.2 \pm 3.4$ | $0.23 \pm 0.23$ | $0.39 \pm 0.15$ |
| $\mathbb{S}^2_{r_1} \times \mathbb{H}^2_{r_2}$ | $\mathsf{d}_{\ell_1}$ | $2.2 \pm 0.7$ | $3.8 \pm 0.7$ | $0.24 \pm 0.29$ | $0.48 \pm 0.17$ |
| $\mathbb{S}^2_{r_1} \times \mathbb{H}^2_{r_2}$ | $\mathsf{d}_{\ell_2}$ | $2.0 \pm 0.7$ | $3.6 \pm 1.5$ | $0.48 \pm 0.24$ | $0.50 \pm 0.23$ |
| $\mathbb{S}^2_{r_1} \times \mathbb{H}^2_{r_2}$ | $\mathsf{d}_{\min}$ | $3.6 \pm 2.5$ | $8.0 \pm 3.6$ | $0.16 \pm 0.30$ | $0.48 \pm 0.24$ |
| $\mathbb{S}^3_{r_1} \times \mathbb{H}^1_{r_2}$ | $\mathsf{d}_{\ell_1}$ | $1.8 \pm 0.7$ | $3.4 \pm 0.8$ | $0.48 \pm 0.19$ | $0.51 \pm 0.17$ |
| $\mathbb{S}^3_{r_1} \times \mathbb{H}^1_{r_2}$ | $\mathsf{d}_{\ell_2}$ | $1.8 \pm 0.7$ | $3.6 \pm 0.9$ | $0.31 \pm 0.21$ | $0.52 \pm 0.16$ |
| $\mathbb{S}^3_{r_1} \times \mathbb{H}^1_{r_2}$ | $\mathsf{d}_{\min}$ | $3.0 \pm 2.3$ | $7.8 \pm 3.2$ | $0.13 \pm 0.42$ | $0.46 \pm 0.22$ |

### D.4 Comparison with products of Riemannian space forms

In the main paper, we do not compare $\mathcal{P}^{p,q}_r$ to products of spherical and hyperbolic manifolds [3, 11] because these product manifolds do not have constant curvature and we could similarly consider products of pseudo-sphere of same dimension to add more complexity, which would have made the paper hard to read. In this subsection, we report these comparisons.

**Notation.** $\mathbb{S}^p_{r_1} := \mathcal{S}^{p,0}_{r_1}$ denotes the $p$-sphere of radius $r_1$ (embedded in a $(p+1)$-dimensional Euclidean space). Similarly, $\mathbb{H}^q_{r_2}$ denotes the $q$-dimensional hyperboloid of "radius" $r_2$ and embedded in a $(q+1)$-dimensional space. Following [3, 11], the radii $r_1 > 0$ and $r_2 > 0$ are trained parameters (both initialized at 1) and we define the following distance metrics for the product manifold $\mathbb{S}^p_{r_1} \times \mathbb{H}^q_{r_2}$ (see [11] for details):

- The geodesic $\ell_2$ distance: $\mathsf{d}_{\ell_2}((\mathbf{x}_1, \mathbf{y}_1), (\mathbf{x}_2, \mathbf{y}_2)) := \sqrt{\mathsf{d}_1^2(\mathbf{x}_1, \mathbf{x}_2) + \mathsf{d}_2^2(\mathbf{y}_1, \mathbf{y}_2)}$

- The $\ell_1$ distance: $\mathsf{d}_{\ell_1}((\mathbf{x}_1, \mathbf{y}_1), (\mathbf{x}_2, \mathbf{y}_2)) := \mathsf{d}_1(\mathbf{x}_1, \mathbf{x}_2) + \mathsf{d}_2(\mathbf{y}_1, \mathbf{y}_2)$

- The min distance: $\mathsf{d}_{\min}((\mathbf{x}_1, \mathbf{y}_1), (\mathbf{x}_2, \mathbf{y}_2)) := \min(\mathsf{d}_1(\mathbf{x}_1, \mathbf{x}_2), \mathsf{d}_2(\mathbf{y}_1, \mathbf{y}_2))$

where $\mathsf{d}_1(\mathbf{x}_1, \mathbf{x}_2) := r_1 \cos^{-1}(\langle \mathbf{x}_1, \mathbf{x}_2 \rangle / r_1^2)$ and $\mathsf{d}_2(\mathbf{y}_1, \mathbf{y}_2) := r_2 \cosh^{-1}(|\langle \mathbf{y}_1, \mathbf{y}_2 \rangle_q / r_2^2|)$ are the geodesic distances of the $p$-sphere of radius $r_1$ and $q-$hyperboloid of radius $r_2$, respectively.

#### D.4.1 Zachary's karate club dataset

We report the scores on Zachary's karate club dataset in Table 9 by using the following evaluation metrics: Rank of first leader, Rank of second leader, Spearman's $\rho$ for the top 5 nodes, Spearman's $\rho$ for the top 10 nodes. These evaluation metrics quantify how much the chosen distance extracts the hierarchy information in the graph.

All these product manifolds perform better than Riemannian space forms (see Table 1) but worse than the quotient manifold $\mathcal{P}^{p,q}_1$. It is worth noting that the best performing distance metrics are $\mathsf{d}_{\ell_1}$ and $\mathsf{d}_{\ell_2}$. They both add the spherical and hyperbolic distances and then explicitly enforce both a spherical and hyperbolic structure when comparing pairs of samples. The fact that they perform worse than the geodesic distance of $\mathcal{P}^{p,q}_r$ indicates that explicitly constructing hyperbolic and spherical parts to the manifold by using products of Riemannian manifolds may not be optimal depending on the selected pairs.

Interestingly, the distance metric $\mathsf{d}_{\min}$ that selects some hyperbolic or spherical distance depending on the pair of samples performs much worse. This is in contrast with our approach that also intrinsically selects a elliptic or hyperbolic type of distance depending on the pair of compared samples (see Eq. (7)). However, the selection in Eq. (7) is based on the (intrinsic) geodesic "distance" of the manifold $\mathcal{P}^{p,q}_r$. Experimental results suggest that the fact that $\mathcal{P}^{p,q}_r$ intrinsically contains hyperbolic and elliptic parts due to the indefiniteness of the metric tensor allows us to better describe hierarchical relationships between samples when the hierarchical graph contains cycles.

Table 10: Evaluation scores for the learned 4-dimensional representations in the node classification task (mean $\pm$ standard deviation)

| Manifold | Distance | Citeseer | Cora | Pubmed |
|---|---|---|---|---|
| $\mathbb{S}^1_{r_1} \times \mathbb{H}^3_{r_2}$ | $\mathsf{d}_{\ell_1}$ | $43.4 \pm 2.6$ | $56.6 \pm 2.9$ | $68.5 \pm 4.8$ |
| $\mathbb{S}^1_{r_1} \times \mathbb{H}^3_{r_2}$ | $\mathsf{d}_{\ell_2}$ | $46.8 \pm 2.1$ | $57.6 \pm 2.4$ | $71.5 \pm 2.1$ |
| $\mathbb{S}^1_{r_1} \times \mathbb{H}^3_{r_2}$ | $\mathsf{d}_{\min}$ | $40.7 \pm 3.9$ | $47.5 \pm 2.5$ | $63.0 \pm 1.4$ |
| $\mathbb{S}^2_{r_1} \times \mathbb{H}^2_{r_2}$ | $\mathsf{d}_{\ell_1}$ | $45.9 \pm 1.9$ | $60.4 \pm 2.8$ | $70.5 \pm 2.6$ |
| $\mathbb{S}^2_{r_1} \times \mathbb{H}^2_{r_2}$ | $\mathsf{d}_{\ell_2}$ | $47.2 \pm 2.1$ | $60.5 \pm 3.2$ | $71.1 \pm 2.5$ |
| $\mathbb{S}^2_{r_1} \times \mathbb{H}^2_{r_2}$ | $\mathsf{d}_{\min}$ | $44.4 \pm 2.3$ | $55.2 \pm 4.9$ | $70.1 \pm 2.1$ |
| $\mathbb{S}^3_{r_1} \times \mathbb{H}^1_{r_2}$ | $\mathsf{d}_{\ell_1}$ | $47.3 \pm 2.0$ | $56.5 \pm 2.4$ | $71.9 \pm 2.1$ |
| $\mathbb{S}^3_{r_1} \times \mathbb{H}^1_{r_2}$ | $\mathsf{d}_{\ell_2}$ | $48.1 \pm 2.1$ | $60.8 \pm 2.8$ | $72.5 \pm 1.8$ |
| $\mathbb{S}^3_{r_1} \times \mathbb{H}^1_{r_2}$ | $\mathsf{d}_{\min}$ | $43.6 \pm 3.2$ | $55.2 \pm 2.9$ | $68.9 \pm 2.6$ |

### D.4.2 Results in node classification

We ran the same kind of experiment as above in the node classification task described in Section 5.2. We report in Table 10 the results obtained with 4-dimensional manifolds and the same distance metrics (see Table 3 for comparison).

Once again, $\mathsf{d}_{\min}$ performs worse than the other distance metrics that perform slight better than hyperbolic and elliptic distances but are still outperformed by our proposed distances on the Cora and Citeseer datasets.

We ran similar experiments for 10-dimensional manifolds. The conclusion is similar.