# OpenReview forum: "Ultrahyperbolic Neural Networks"
_NeurIPS.cc/2021/Conference — NeurIPS 2021 Spotlight_

### Official Review · Reviewer_J8on · 2021-07-15

**Rating:** 5
**Confidence:** 4

**Summary:**

This paper presents a (graph) neural network in ultrahyperbolic (semi-Riemannian) space that can be used to model hierarchical graphs with cycles. The motivation is that ultrahyperbolic manifold generalizes hyperbolic and spherical manifolds, thus providing inductive bias to those geometries. In order to avoid broken cases, the paper considers ultrahyperbolic quotient manifold such that every two points in the manifold can be connected by a geodesic. The method is evaluated in graph and node classification tasks.

**Main Review:**

Originality:
1)	In general, extending neural networks to semi-Riemannian space is a new and valuable idea. However, the method used in this paper to solve the brokenness of geodesic is straightforward, even a little bit trivial and lack theoretical analysis. The idea is simply that, if points x and y cannot be geodesically connected, they connect the point x with the antipodal point -y. This method would potentially hurt the expressiveness of the ultrahyperbolic space in paper [1].

2)	In section 3.2, the author claimed that they first formulated parallel transport of the ultrahyperbolic space. However, the parallel transport of ultrahyperbolic space was first introduced in the paper [2] (see Section 4.2.1 and 4.2.2). The authors should clarify the contributions and cite the related works properly.

Quality:
1)	The authors try to solve the broken issues of ultrahyperbolic manifold by identifying each pair of antipodal points to one point, I was mainly concerned with the expressiveness of the induced space as it is no longer a standard pseudo-sphere. It is clear that the pseudo-sphere can generalize spherical and hyperbolic spaces well [1], but simply identifying each pair of antipodal points to one point would hurt the expressiveness of the space. The author used the quotient space, which does not satisfy a lot of properties of the pseudo-hyperboloid used in the paper [1]. Taking sphere as an example, why a point x and its antipodal point -x leads to duplications? these two points could be used to describe different nodes in a cycle of graph. It seems like the “quotient space” even cannot model a cycle graph well. Note that in a spherical space, point x and point -x can describe two antipodal nodes in a graph.  Instead, the quotient space can only model a half-cycle relationship, as it treats points x and -x equally.
2)	The author showed some intuitions of the quotient manifold in Fig 1, where the space-like geodesic could be used to describe cycle, and the time-like geodesic could be used to describe trees, but if we treat each pair of antipodal points to one point, the geodesic cannot fully describe some relationships. (e.g. starting with point x, there is only a half-cycle passing x, but not a full cycle).
3)	In line 83, the author also claimed that the quotient manifold is only a submanifold of pseudo-sphere, but it is not clear which submanifold it is. A submanifold can only express some sub-information of the original manifold. The author might need to clarify what information loss it led to.
4)	I would advocate the authors to provide more theoretical analysis w.r.t the method, and try to understand more on the expressiveness of the ultrahyperbolic quotient manifold. So far, it is unclear how the method works to model tree and cycles as the space used in this paper is not an ultrahyperbolic manifold anymore.

Clarity: the paper is well-written and well-organized, there are some minor points that need to be clarified.

1)	In line 78, does the radius r only play a role of scaling factor? Assuming infinite precision, I think it is true, but in practice, I would imagine that changing the radius r would also influence the experiment results. In paper [3], the authors showed some improvements by learning the right curvatures. The author might also want to do some ablation study on this issue.
2)	The toy experiments in section 5.1 follows mainly the experimental setting of paper [1], but show different results (the best performance is achieved by different time dimension), it would be better if the author can add more details to explain the results.

Significance:
      The experiments lack some comparisons to some existing works.
1)	Since the proposed method is a little bit straightforward, and the theoretical contribution is not significant, I would suggest doing more experiments to compare the method with some more recent baselines, e.g. produce space [4], to showcase the advantages of the method.

2)	How to determine the time dimension? Could you provide some insights on this? it would also be interesting to do some ablation study on the time dimensions if possible. Consider adding them into the appendix.

[1] Law, M.T. and Stam, J., 2020. Ultrahyperbolic representation learning. NeurIPS 2020.
[2] Gao, T., Lim, L.H. and Ye, K., 2018. Semi-Riemannian manifold optimization. arXiv preprint arXiv:1812.07643.
[3] Chami, I., Ying, Z., Ré, C. and Leskovec, J., 2019. Hyperbolic graph convolutional neural networks. Advances in neural information processing systems, 32, pp.4868-4879.
[4] Bachmann, G., Bécigneul, G. and Ganea, O., 2020, November. Constant curvature graph convolutional networks. In International Conference on Machine Learning (pp. 486-496). PMLR.

Post Rebuttal:
 The authors give more intuitions to the expressiveness of the manifold (e.g.. to describe cyclic relationships), so I raised my score. But I still suggest the authors provide more theoretical analysis w.r.t the method.


**Time Spent Reviewing:**

5

---

> ### Author Response · Authors · 2021-08-04
> **Comments about originality**
>
> Thank you for your review. We focus here on the "originality part".
>
> ### 2. Contribution of the parallel transport of $\mathcal{S}_{r}^{p,q}$
>
> Following the notation of [B], the formulation of the parallel transport of $\Delta$ along some geodesic $\gamma(t)$ is indeed provided only for the unit pseudo-sphere (i.e., $r=1$) in Proposition 4.20 of [B] in the following form: $\Delta(t) = - \langle \Delta, X \rangle \int_0^t \gamma(\tau) d\tau + \Delta$. Here, $\langle \cdot, \cdot \rangle$ corresponds to the scalar product $\langle \cdot, \cdot \rangle_q$ in our paper.
>
> For any radius $r > 0$, it can be shown that this parallel transport is formulated $\Delta(t) = - \frac{1}{r^2} \langle \Delta, X \rangle \int_0^t \gamma(\tau) d\tau + \Delta$.
>
> Eq. (6) corresponds to setting $t=1$, considering that $\gamma(0) = \textbf{x}$ and setting $X$ to the logarithm map of $\textbf{y}$ at $\textbf{x}$ (i.e., $X = \bar{\log}_{\textbf{x}} (\textbf{y})$). In this case, $X$ is obtained via the minimizing geodesic (i.e., the inverse of the exponential map), which is not defined in [B].
> We will clarify this in the next version. We can write the complete proof in the discussion if requested but the proof simply follows the definition.
>
> Moreover, we also provide the parallel transport formulation for $\mathcal{P}_{r}^{p,q}$, which is not discussed in ref [B].
>
> ### 1. Loss of expressiveness compared to the manifold in [C]?
>
>  We would like to remind that we do not consider the same manifold $\mathcal{S}_r^{p,q}$ as [C]. Our considered manifold is instead the projective space $\mathcal{P}_r^{p,q} = \mathcal{S}_r^{p,q} / \pm 1$.
>
> $\mathcal{S}_r^{p,q}$ generalizes spherical and hyperbolic geometries into a single pseudo-Riemannian manifold of constant curvature $1/r^2$. On the other hand, $\mathcal{P}_r^{p,q}$ generalizes elliptic and hyperbolic geometries into another single pseudo-Riemannian manifold of constant curvature $1/r^2$ as explained in Section 2.
>
> Considering $\mathcal{P}_r^{p,q}$ does not "hurt" the expressiveness of $\mathcal{S}_r^{p,q}$. For instance, the $d$-sphere does not hurt the expressiveness of the Euclidean space and the hyperboloid does not hurt the expressiveness of the ambient Minkowski space. However, due to the nature of the discrete group that we chose, $\mathcal{P}_r^{p,q}$ behaves locally like $\mathcal{S}_r^{p,q}$. Indeed, if a set of $n$ points all satisfy $\forall i,j, \langle \textbf{x}_i, \textbf{x}_j \rangle_q \geq 0$ (this condition for a given pair is valid on a given half space), their geodesic distance of $\mathcal{P}_r^{p,q}$ is exactly  the geodesic distance of $\mathcal{S}_r^{p,q}$.
>
> The fact that we work in a projective space and that pairs of antipodal points of $\mathcal{S}_r^{p,q}$ are considered as a single point of $\mathcal{P}_r^{p,q}$ does not prevent considering cycles (hence cyclic relationships) in our projective space (e.g., ref [D] and our message "Quality and other comments"). Cyclic relationships do not have to be represented only as a standard cycle. It is true that a cycle in $\mathcal{P}_r^{p,q}$ is half as long as a cycle in  $\mathcal{S}_r^{p,q}$ in terms of arc length (i.e., the minimum length of the arc to pass through the same point) but a cycle can also be described in $\mathcal{P}_r^{p,q}$ as explained in our message "Quality and other comments".
>
> We do not agree that our choice of manifold hurts expressiveness. Nonetheless, we would like to remind that although $\mathcal{S}_r^{p,q}$ is geodesically complete, there is no way to define a minimizing (unbroken) geodesic for every pair of points when $p$ and $q$ are both positive since $\mathcal{S}_r^{p,q}$ is non-Riemannian. Law and Stam [C] discuss this in their paper (end of Section 3) and say that defining a distance for pairs of points that are not joined by an unbroken geodesic is still an open problem.
>
> By considering instead the projective space $\mathcal{P}_r^{p,q}$, we are able to define a well-founded geodesic distance (see Eq. (7)) between every pair of points. This "geodesic distance" is the arc length of the geodesic as defined for pseudo-Riemannian manifolds (that generalize the Riemannian case). We can then also define the logarithm map for pairs of points for which the logarithm map is not defined on $\mathcal{S}_r^{p,q}$.
>
> ### Simplicity and proof of descent direction
>
> We are glad that the reviewer thinks our method to optimize a neural network defined on a non-Riemannian manifold (using pseudo-Riemannian optimization tools) is simple. However, we do not agree that it is straightforward and we would like to emphasize that this has never been done for parametric models. We formulate our framework based on the differential geometry notion of vertical and horizontal bundles (see Section 3.1), which allows us to have a bijection between the tangent spaces of $\mathcal{P}_r^{p,q}$ and $\mathcal{S}_r^{p,q}$. We do not know other work in deep learning that uses this kind of bijection.
>
> For simplicity of notation, we note $\boldsymbol{\lambda} := \boldsymbol{\lambda}_{[\textbf{x}],\textbf{p}}$ the parallel translate of the pseudo-Riemannian gradient defined in Eq. (9). We assume $\langle \textbf{x}, \textbf{p} \rangle_q > 0$ (the case $\langle -\textbf{x}, \textbf{p} \rangle_q > 0$ follows the same idea by considering geodesics whose initial point is $-\textbf{x}$ instead of its antipodal point).
>
> By using Taylor's first order approximation and following the proof of Section 4.2 of [C], decreasing the objective function $\bar{f}$ around $\textbf{x}$ corresponds to finding a tangent vector $\boldsymbol{\zeta} \in T_{\textbf{x}} \mathcal{S}_r^{p,q}$ such that: $\langle  D\bar{f}(\textbf{x}), \boldsymbol{\zeta} \rangle_q < 0$.
>
> However, the neural network $\varphi_{\theta} : \mathcal{X} \to \mathcal{H}_{\textbf{p}}$ maps data onto the horizontal space of $\textbf{p}$ which is different from the tangent space of $\textbf{x}$  if $\textbf{x} \neq \textbf{p}$.
>
> Therefore, by using the linear isometry property of the parallel transport that we mention in lines 160-162 (see page 66 of [A]), we find:
>
> $$
> \langle  D\bar{f}(\textbf{x}), \boldsymbol{\zeta} \rangle_q = \langle \boldsymbol{\lambda}, - \textbf{G} \boldsymbol{\lambda} \rangle_q = - ||\boldsymbol{\lambda}||^2_2 \leq 0$$
>
> Due to the properties of the differential of the exponential map at the origin, the search direction $- \textbf{G} \boldsymbol{\lambda} \in \mathcal{H}_{\textbf{p}}$ in Section 3.4 is then a descent direction that can optimize the neural network via standard backpropagation (by applying the chain rule).
>
> [A] O'Neill, "semi-Riemannian Geometry with Applications to Relativity", 1983
>
> [B] Gao et al., "Semi-Riemannian optimization", 2018
>
> [C] Law and Stam, "Ultrahyperbolic representation learning", 2020.
>
> [D] Aceves et al., "Cycles in projective spaces", Journal of Geometry, 2014

---

> ### Author Response · Authors · 2021-08-10
> **Quality and other comments**
>
> ## Questions about quality:
>
> We partially answered some of these questions in our previous message, we give more details in this message.
>
> ### 1. Cyclic relationships:  ###
>
> It seems that most of the confusion of Reviewer J8on comes from the fact that they try to interpret the manifold $\mathcal{P}_r^{p,q}$ only by considering the pseudo-sphere $\mathcal{S}_r^{p,q}$ in the ambient space. However, $\mathcal{P}_r^{p,q}$ is a projective space so every point $[\textbf{x}] = (-\textbf{x}, \textbf{x})$ can be interpreted as the intersection of the pseudo-sphere with a line passing through the origin of $\mathbb{R}^{p+1,q}$. In some cases, it might be easier to interpret points of $\mathcal{P}_r^{p,q}$ as lines passing through the origin with projective geometry and then think about how they behave when they intersect with the pseudo-sphere.
>
> In the Riemannian case of the hyperboloid (i.e., $p=0$), the considered manifold corresponds exactly to the standard (nonprojective) geometry of the hyperboloid since a line passing through the origin intersects one sheet of the hyperboloid at only one point.
>
> The other Riemannian case (i.e., $q=0$) is a well studied projective geometry called "(projective) elliptic geometry" and each point corresponds to  the intersection of the standard sphere with a line passing through the origin. In elliptic geometry, a great circle is called "elliptic straight line", and other circles in $\mathcal{S}^2$ are called "elliptic cycles", they are all in fact "generalized circles" (also called "clines") in the Möbius subgeometry model where straight lines and circles have very similar properties. An elliptic straight line or elliptic cycle will reach multiple times the initial point (which is also equivalent to its antipodal point in that geometry) and follow the same path if it is extended indefinitely, it then forms a cycle.
> By its nature in that geometry, a straight line in elliptic geometry can then describe a cyclic relationship between points. This can be extended to the indefinite case $\mathcal{P}_r^{p,q}$ (e.g., see Fig. 1 of our submission for an illustration of space-like geodesic  $\bar{\gamma}$ of $\mathcal{S}_r^{p,q}$ that can be written as a geodesic $\gamma := \pi \circ \bar{\gamma}$ of $\mathcal{P}_r^{p,q}$ as explained in Section 3.2). We hope this clarifies your concern.
>
>
> In the paper, we also mentioned the argument of duplication of geometric information which was directly quoted from ref [G].
>
> 2. We answered this concern above by mentioning how "elliptic straight lines" and "elliptic cycles" can describe cyclic relationships by their nature in elliptic geometry. Cycles in $\mathcal{P}_r^{p,q}$ are slightly different from cycles in $\mathcal{S}_r^{p,q}$ but the idea of passing through the same point along a geodesic and using the same path indefinitely is also satisfied for $\mathcal{P}_r^{p,q}$. The representations learned in $\mathcal{S}_r^{p,q}$ or $\mathcal{P}_r^{p,q}$ lie in different spaces but both spaces can describe cyclic relationships. See also ref [D].
> It is also worth noting that if we use the intrinsic geometry view of $\mathcal{P}_r^{p,q}$, the (extrinsic) geometry of $\mathcal{S}_r^{p,q}$ does not exist for the inhabitants of $\mathcal{P}_r^{p,q}$ that cannot see pairs of antipodal points for instance.
>
> 3. In line 83, we say that the pair $[\textbf{x}] = (-\textbf{x}, \textbf{x})$ is a submanifold of $\mathcal{S}_r^{p,q}$ and a discrete space, as explained in Appendix A of [H] for instance. We only use this property to define the horizontal space of $\mathcal{S}_r^{p,q}$ at any point in lines 131-132. As explained in [G],  the antipodal duplication information in spherical geometry is lost, but we can still use properties of projective/elliptic geometry to describe cyclic relationships [D].
>
> The manifold $\mathcal{P}_r^{p,q}$ is a quotient manifold as we explain in our message "Major concern about Section 2" to Reviewer xWxM.
> The notion of "antipodal point" is lost but cycles and tree-like structures can still be represented in $\mathcal{P}_r^{p,q}$ as we explained in 1.. Considering $\mathcal{P}_r^{p,q}$ instead of $\mathcal{S}_r^{p,q}$ allows us to be sure that every pair of points can be joined by an unbroken geodesic.
>
> 4. We already answered this question in 1. Our geometry generalizes both hyperbolic geometry (that can represent trees) and elliptic geometry (that can represent cycles), and has to be interpreted with projective geometry. We will clarify that our manifold is an "indefinite elliptic space" [I] and is not exactly the same manifold as [C]
>
>
> ## Clarity
>
> ### 1. Role of the curvature 1/r^2
>
> In theory, the manifold $\mathcal{P}_r^{p,q}$ is homothetic to $\mathcal{P}_1^{p,q}$ for any $r > 0$ that then only plays a role of scaling factor. Nonetheless, we provided all the tools in Section 3 to work with any value of $r > 0$.
>
> In practice, and even in standard Euclidean models, the scale of neural network weights, learning rate, temperature hyperparameter, used optimizer etc. can play a huge role on performance. Therefore, it is not surprising that learning $r$ in a neural network might improve accuracy. For simplicity, we fixed $r=1$ in our experiments but this can easily be learned.
>
> ### 2. Difference of results with Law and Stam [C]
>
> This is a good point. We recall that the manifold considered in [C] generalizes hyperbolic and spherical geometries whereas our manifold generalizes hyperbolic and elliptic geometries. When $p$ is small (i.e., when our manifolds are very similar to a hyperboloid), our scores are similar to those reported in [C]. As $p$ gets larger, our manifold becomes more elliptic whereas the manifold in [C] becomes more spherical and we get better scores than them in some cases. It seems that elliptic geometry performs slightly better than spherical geometry in the considered task.
>
> ## Significance
>
> ### 1. More baselines
>
> Please see our message to Reviewer KvL5 for a comparison and discussion about mixed curvature models. We will add the discussion in the next version.
>
> ### 2. More insight
>
> The optimal index (or number of time dimensions $q$) is still an open problem that could be determined based on a validation set. We believe that, as a graph contains more and more cycles (in particular more communities of nodes), the number of time dimensions has to be decreased (i.e., the manifold becomes more elliptic and less hyperbolic), we will perform such an ablation and include it in the appendix. Thank you for this suggestion.
>
> ### References
>
> [C] Law and Stam, "Ultrahyperbolic representation learning", 2020.
>
> [D] Aceves et al., "Cycles in projective spaces", Journal of Geometry, 2014
>
> [G] Ratcliffe, Foundations of Hyperbolic Manifolds, Springer 2006
>
> [H] Lee, Introduction of Smooth manifolds, second edition
>
> [I] Wolf, "Spaces of constant curvature", sixth edition, 2010 (first published in 1967)

---

> > ### Author Response · Authors · 2021-08-28
> > **Message to Reviewer J8on**
> >
> > Dear Reviewer J8on,
> >
> > Thanks for your initial detailed review. We hope that we have addressed your concerns and adequately answered your questions about the difference between our projective space and the manifold considered in Law & Stam.
> > In particular, we have explained how our projective space is able to describe cycles (hence cyclic relationships) despite its lack of antipodal information, and also generalize hyperbolic geometry.
> > We would greatly appreciate if you could update your review with any further comments or new questions, we would be happy to answer them and we will update the draft accordingly. Thanks again for your time!

---

### Official Review · Reviewer_v58v · 2021-07-16

**Rating:** 6
**Confidence:** 3

**Summary:**

This paper proposes graph neural networks for ultrahyperbolic space along with various ways to optimize them.

**Ethical Concerns:**

There are no ethical concerns.

**Limitations And Societal Impact:**

The others adequately addressed the limitations and potential negative societal impact of their work.

**Main Review:**

Strengths

* The construction is technically sound.
* The experiments show modest improvement across the board.

Weaknesses

* None of the constructions are particularly novel. In particular, they seem to be natural extensions of the same operations in regular hyperbolic space.
* The main motivation for ultrahyperbolic manifolds seems to be to represent different parts of a graph that have different geometries. This is also the goal of the mixed curvature literature [1, 2, 3], but no discussion and experimental comparison is made with this prior work.
* The experiments heavily rely on 3 canonical datasets (Citeseer cora and pubmed) but doesn't delve into the geometric properties of said datasets. Is there any reason why one would use ultrahyperbolic GCNs on these datasets?
* While there is a general increase in performance, SOTA Euclidean methods are not compared against. This is especially important since the three datasets were shown to perform worse on hyperbolic graph models when compared against SOTA Euclidean methods.

Verdict

While the neural networks are well-constructed, nothing is particularly surprising or novel. Furthermore, the experiments do not compare against SOTA methods for both Euclidean and manifold-based graph methods and rely on 3 canonical datasets with no apparent geometric structure. As such, I lean reject.

[1] Learning Mixed-Curvature Representations in Product Spaces, ICLR 2018
[2] Mixed-curvature Variational Autoencoders, ICLR 2020
[3] Constant Curvature Graph Convolutional Networks, ICML 2020


**Time Spent Reviewing:**

3

---

> ### Author Response · Authors · 2021-08-10
> **Response to Reviewer v58v**
>
> ### Novelty ###
>
> We are glad that you think our framework is technically sound since the goal of our paper was to propose an optimization framework to learn any neural network that maps to some specific kind of non-Riemannian manifold of constant nonzero curvature. We do not know any other deep learning model (or parametric model) that maps data to a non-Riemannian manifold and uses pseudo-Riemannian optimization tools to optimize it. This is the main novelty of our paper.
>
> This is not an easy task since even if a non-Riemannian manifold is geodesically complete, there may exist pairs of points on the manifold that cannot be joined by an unbroken geodesic. Therefore, standard tools to optimize Riemannian neural networks such as parallel transport and the logarithm map are not straightforward to extend.
>
> As explained in Section 2, we propose to work on a non-Riemannian quotient manifold $\mathcal{P}_r^{p,q}$ that generalizes both the elliptic geometry when $q = 0$ and hyperbolic geometry when $p=0$. Elliptic and hyperbolic geometries are actually nice special cases since they correspond to working on Riemannian manifolds so standard Riemannian geometry tools can be used to optimize the neural network. In particular, the negative of the Riemannian gradient is a descent direction.
>
> Our framework has two major challenges when the considered manifold is non-Riemannian (e.g., when both $p$ and $q$ are positive):
>
> $\bullet$ To be sure that any pair of points can be joined by an unbroken geodesic, we consider the quotient manifold $\mathcal{P}_r^{p,q}$. We then have to use principal bundle tools such as the horizontal space and horizontal lift operator to be able to equivalently work on the quotient space $\mathcal{P}_r^{p,q}$ and its total space (in our case, the total space is the pseudo-sphere $\mathcal{S}_r^{p,q}$ defined in Eq. (2)). In this way, we are able to have a bijection between any tangent vector in the quotient manifold $\mathcal{P}_r^{p,q}$ and the total space $\mathcal{S}_r^{p,q}$. We can then write in closed form differential geometry tools such as the geodesics of $\mathcal{P}_r^{p,q}$, its exponential map, logarithm map, geodesic distance and parallel transport (see Sections B.1 and B.2 in the appendix for details).
> Since we have all these tools in closed form, we can directly represent the weights of our neural network in some specific horizontal space and extend standard neural networks.
>
> $\bullet$ The negative of the pseudo-Riemannian gradient is not a descent direction if the manifold is non-Riemannian. We propose a simple and efficient descent direction in Section 3.4. We give the detailed proof in our message "Comments about originality" to Reviewer J8on that it is a descent direction and can be used to optimize our neural network via backpropagation.
>
> We believe that proposing a framework to optimize a difficult problem in a way that sounds natural to the reader is a nice contribution, particularly since it includes elliptic and hyperbolic geometries as simple special cases. We recall that this is the first neural network that optimizes over a non-Riemannian manifold so many properties satisfied in Riemannian geometry are not valid anymore. Therefore, we propose an optimization framework using the tools available for general smooth manifolds.
>
> ### Mixed curvature literature
>
> Please see our message to Reviewer KvL5 for a comparison and discussion about mixed curvature models. We will add the discussion in the next version.
>
> ### Geometry of the datasets
>
> We will add a discussion on the datasets in the next version. In the paper, we described Zachary's karate club dataset to understand our model since it is a simple dataset and the main motivation was to work on hierarchical graphs with cycles.
>
> Cora, Citeseer and Pubmed are bibliographic/citation network data sets that are also hierarchical graphs with cycles. For instance, in a similar way as Zachary's karate club dataset, articles that have lots of citations are higher in the hierarchy than articles that are less cited. This is apparently why our considered manifold works better than other manifolds on these datasets.
>
> ### Comparison against Euclidean SOTA
>
> Although comparing or adapting Euclidean SOTA to ultrahyperbolic space would be nice, this is not the focus of this work. The main motivation of this paper was a general optimization framework to work on pseudo-Riemannian manifolds of constant nonzero curvature, which is not trivial. As described in Section 3, our framework is general and can be applied to different kinds of models and tasks. In Section 4, we adapted it to some specific but classic graph neural network and we compared the effect of a given model (GCN) when it represents intermediate levels on different manifolds (Euclidean, hyperbolic, elliptic etc.). For instance, in Section 5.1, we did not use GCNs, we trained a standard multilayer perceptron. This shows the great flexibility of our approach that could definitely be adapted to other SOTA models.

---

> > ### Comment · Reviewer_v58v · 2021-08-15
> > **Updated Review**
> >
> > Thank you for responding to my concerns. I have raised my score from 5 -> 6.

---

### Official Review · Reviewer_xWxM · 2021-07-16

**Rating:** 7
**Confidence:** 1

**Summary:**

The paper focuses on data embedded in ultrahyperbolic spaces and proposes a method to learn this representation via neural networks. By imposing a quotient structure over the ultrahyperbolic space defined in Low and Stam 2020, the authors are able to define geodesics and thus to estimate the parameters of the learning model selected.

**Limitations And Societal Impact:**

Yes. The conclusions and the limitations are well written.

**Main Review:**

The paper is overall well written. Section 2 is clear and well organized.

Major Comments:
- Section 2: why is the quotient space a manifold? Are the authors excluding the axes? Because with the axes, the equivalence classes (orbits) do not have the same dimension, thus the action is not free and stating that the quotient space is a manifold because the total space is a manifold it is not straightforward.

- Section 3: in quotient spaces, the horizontal lift is a good solution to work on the tangent space. However, the curvature of the space can create problem by using only a one step horizontal lift (see e.g. the procrustes algorithm for shapes or the Align All and compute algorithm for graphs). If I am not misunderstanding, some discussion about the single lift procedure should be added.

- Section 5: if I have properly understood, the usage of the hyperbolic space to estimate the parameters allows a proper embedding of the network. If so, I would rather like to see the algorithm applied to a prediction of a graph task more than a classification task, where the output should be embedded into a suitable geoemtrical framework.

Additional comments:
- To introduce complex geometrical framework to perform classification, a well written justification is necessary. It is not clear to me how much of all the introduced framework it is acctually needed to obtain a better performance.

**Time Spent Reviewing:**

2

---

> ### Author Response · Authors · 2021-08-04
> **Major concern about Section 2**
>
> We thank you for your positive review. We address the major concern about Section 2 in this message and will post separate responses for other concerns.
>
> ## Quotient manifold ##
>
> The quotient space $\mathcal{P}_r^{p,q} := \mathcal{S}_r^{p,q} / \pm 1 =  \mathcal{S}_r^{p,q} / \pm \textbf{I}$ is a quotient manifold. In the submission, we refer the reader to Chapter 7 of [A] for general definitions but we will provide the proof here. It is mostly based on the "Orbit manifolds" section of Chapter 7 of ref [A]. We then first copy paste its Definition 6 and Proposition 7.
>
> ### Definition 6 of Chapter 7 of [A].
>
> A group $\Gamma$ of diffeomorphisms of a manifold $\mathcal{M}$ is properly discontinuous and acts freely provided:
>
> (PD1) Each point $\textbf{x} \in \mathcal{M}$ has a neighborhood $\mathscr{U}$ such that if $\phi(\mathscr{U})$ meets $\mathscr{U}$ for $\phi \in \Gamma$ then $\phi = $id.
>
> (PD2) Points $\textbf{x}, \textbf{z} \in \mathcal{M}$ not in the same orbit have neighborhoods $\mathscr{U}$ and $\mathscr{V}$ such that for every $\phi \in \Gamma$, $\phi(\mathscr{U})$ and $\mathscr{V}$ are disjoint.
>
> ### Proposition 7 of Chapter 7 of [A].
>
> Let $\Gamma$ be a properly discontinuous group of diffeomorphisms of a manifold $\mathcal{M}$. There is a unique way to make a $\mathcal{M} / \Gamma$ a manifold so that the natural map $\pi : \mathcal{M} \to \mathcal{M} / \Gamma$ is a covering map.
>
>
> ### Proof. ###
>
> In our case, we have $\mathcal{M} = \mathcal{S}_r^{p,q}$, and $\Gamma = \{ \pm 1 \} = \{ \pm \textbf{I} \}$ is a group of diffeomorphisms of $\mathcal{S}_r^{p,q}$.
> More exactly, $\Gamma$ is composed of the identity map $\textbf{x} \mapsto \textbf{x}$ and the antipodal map $\textbf{x} \mapsto -\textbf{x}$.
> For all $\textbf{x} \in \mathcal{S}_r^{p,q}$, the set $\{ \phi(\textbf{x}) : \phi \in \Gamma \} = [ \textbf{x} ]$ (i.e., the pair composed of $\textbf{x}$ and $-\textbf{x}$) is the orbit of $\textbf{x}$ under $\Gamma$. The collection of all such orbits is our set $\mathcal{P}_r^{p,q} := \mathcal{S}_r^{p,q} / \pm 1$.
>
> (PD1) is satisfied when the neighborhood $\mathscr{U}$ is defined as $\mathscr{U} = \{ \textbf{y} \in \mathcal{S}_r^{p,q} : \langle \textbf{x}, \textbf{y} \rangle_q > 0 \}$.
>
> (PD2) is satisfied if $\textbf{z} \neq \pm \textbf{x}$ by determining some neighborhood small enough for both $\textbf{z}$ and $\pm \textbf{x}$ so that they are disjoint.
>
> By definition, each point $\textbf{x} \in \mathcal{M}$ has a connected neighborhood $\mathscr{U}  = \{ \textbf{y} \in \mathcal{S}_r^{p,q} : \langle \textbf{x}, \textbf{y} \rangle_q > 0 \}$ that is evenly covered by $\pi$, and it maps each component of $\pi^{-1} (\mathscr{U})$ diffeomorphically onto $\mathscr{U}$. It is then a covering map.
>
> $\mathcal{P}_r^{p,q}$ is then a quotient manifold.
>
> It is worth noting that $\mathcal{P}_r^{p,q}$  is briefly mentioned in page 214 (Chapter 7) of ref [A].
>
> For instance, in the Riemannian case, the group $\{ \pm \textbf{I} \} \subset O(d+1)$ acts freely and properly discontinuously on the sphere $\mathcal{S}^d$, where $O(d+1)$ is the orthogonal group in dimension $d+1$. The projective lines, projective planes etc. in  $\mathcal{P}^d$ are the images of the sub-spheres obtained by intersecting $\mathcal{S}^d$ with linear subspaces of appropriate dimension in $\mathbb{R}^{d+1}$ (see page 74 of [I]).
>
> This is easily generalized to the pseudo-sphere as we showed above, and $\mathcal{P}_r^{p,q}$  is an "indefinite elliptic space" and defined in Equation (12.2.2a) of ref [I].
>
> In an ambient space of dimensionality $d+1$, both the pseudo-sphere $\mathcal{S}_r^{p,q}$ and the projective space $\mathcal{P}_r^{p,q}$ are d-dimensional (so their dimensionality is one less than the ambient space $\mathbb{R}^{p+1,q}$). We hope this answers your concern.
>
> [A] O'Neill, "semi-Riemannian Geometry with Applications to Relativity", 1983
>
> [I] Wolf, "Spaces of constant curvature", sixth edition, 2010 (first published in 1967)

---

> ### Author Response · Authors · 2021-08-04
> **Other important important comments**
>
> ## A single horizontal lift step is sufficient due to the nature of our quotient manifold. ##
>
> $\bullet$ Section 3: We are not familiar with the mentioned algorithms. However, we would like to emphasize two nice properties of our quotient manifold $\mathcal{P}_r^{p,q}$ that make our one step horizontal lift sufficient for our problem.
>
> 1) $\mathcal{P}_r^{p,q}$ has constant curvature $1/r^2$, which is not the case for most shapes in the literature.
>
> 2) Our manifold only considers pairs (-$\textbf{x}, \textbf{x}$) = $[ \textbf{x} ]$ as single points of $\mathcal{P}_r^{p,q}$.
>
> Thanks to these two properties, many differential geometry tools such as the geodesic, exponential map and its inverse (logarithm map), parallel transport and geodesic distance can be written in closed form for $\mathcal{P}_r^{p,q}$. We also chose this manifold so that any pair of points can be joined by at least one geodesic.
> As we explain in our message "Comments about originality" to Reviewer J8on, parallel transport can be defined along any smooth curve as long there exists a parallel vector field.
>
> We chose to define our parallel transport over the minimizing geodesic that is obtained via the logarith map in closed form. Finding the minimizing geodesic is difficult in general for other kinds of shapes and can lead to some issues in terms of parallel translation. In our case, choosing the logarithm map leads to a canonical way of performing parallel transport (see Eq. (8)).
> Since our goal is to find the parameters of the neural network $\varphi_{\theta} : \mathcal{X} \to \mathcal{H}_{\textbf{p}}$ that minimize some objective function. We show in our message "Comments about originality" to Reviewer J8on that the selected search direction in Section 3.4 is a descent direction that decreases the value of our function. It can then be used to optimize our neural network by applying the chain rule. For our specific optimization framework, using a single horizontal lift step is sufficient by exploiting the linear isometry property of the parallel transport (see our message "Comments about originality" to Reviewer J8on).
>
> ## Our optimization framework is general and can optimize any function as described in Section 3. ##
>
> $\bullet$  Section 5: We agree that our optimized function in Section 3 is general and need not describe a classification task. It can for instance also be applied for other graph prediction tasks. In our toy experiment, we do not perform classication and we show that our algorithm framework is able to extract the hierarchy of a social network graph dataset (see Table 1 of our paper).
>
> The nodes of the graph are mapped to the manifold and (sums of) distances between nodes are compared to predict which nodes are the most important in the (social network) hierarchy. With our manifold, important nodes tend to have a smaller geodesic distance with other nodes than nodes lower in the hierarchy compared to Riemannian manifolds (see Table 1).
> We believe that this is a suitable geometrical framework to validate the relevance of our proposed representations.
>
> If you have another application in mind, we would be happy to test it.
>
> ## Many approaches have tried to combine spherical and hyperbolic approaches in the machine learning literature to represent graphs. ##
>
> $\bullet$  Additional comment: Our proposed framework is a generalization of hyperbolic and elliptic approaches. Some of these approaches were shown to improve performance on some types of graphs. Our manifold can describe any relationship of hyperbolic and elliptic geometries but is more flexible and can also describe other kinds of relationships since the geodesic distance does not satisfy all the properties of a distance metric. More details can be found in Appendix, Section B of [C]. Satisfying all the properties of a distance metric has never been an issue in many machine learning subfields such as the metric learning literature where the learned metric is a pseudo-metric (that is sometimes not even positive semi-definite).
>
> Some recent approaches have proposed to use products of manifolds of constant curvature (i.e., sphere or hyperbolic space). However, these constructed manifolds are in general not space forms themselves (i.e., they do not have constant curvature). On the other hand, $\mathcal{P}_r^{p,q}$ is a space form and we could equivalently consider products of pseudo-Riemannian space forms, which would also generalize the products of Riemannian space forms proposed as baselines by other reviewers. We briefly justify our choice of baseline in lines 284-285 of our submission. We have added a detailed comparison baselines using products of manifolds in our message to Reviewer KvL5.
>
> [C] Law and Stam, "Ultrahyperbolic representation learning", 2020.

---

### Official Review · Reviewer_KvL5 · 2021-07-18

**Rating:** 7
**Confidence:** 4

**Summary:**

The paper proposes to learn ultrahyperbolic representations with neural networks. Unlike prior work in ultrahyperbolic representation learning, the paper proposes dealing with an ultrahyperbolic quotient manifold to attain geodesic completeness and enable optimization of parametric models that use this space as an intermediate representation. The optimization of ultrahyperbolic neural networks is then developed in the paper, and an extension to Graph Neural Networks (GNNs) is introduced. Afterwards, the approach is evaluated in various different graph classification tasks.

**Ethical Concerns:**

This paper is reasonably theoretical in nature; I have no ethical concerns about its impact.

**Limitations And Societal Impact:**

The paper has two subsections of the conclusion that address limitations and societal impacts. In particular, the paper admits the lack of a theoretical analysis (similar to what can be found in [7]) for graphs without cycles.

**Main Review:**

Strengths

1. Overall, I think the theory presented in this paper was fairly original and significant, as it enables learning neural networks with intermediate representations that are ultrahyperbolic; notably, this is a generalization of prior work, which limits intermediate representations to spaces of constant sectional curvature. I think that using a quotient manifold construction to attain geodesic completeness and enable optimization was a clever idea, and further, the other theory necessary for optimization (namely parallel transport on $\mathcal{S}^{(p,q)}_r$ and the use of horizontal lifts) was clearly introduced and well-developed. Once these constructs were in place, the generalization of GCNs to ultrahyperbolic space was naturally given in Section 4. I think all throughout, the writing was clear and the construction was clean.

2. The results in Table 3 on node classification in graph datasets show the benefit of ultrahyperbolic space in attaining considerably better results than both the standard GCN and the hyperbolic GCN; the margin of improvement exhibited by $\mathcal{P}^{(0,4)}_1$ seems fairly significant.

Weaknesses

1. I find the results of Table 5 on graph classification to be unconvincing. The standard deviations on most numbers in the table are rather high, high enough that I'm not sure the claimed performance increase of an ultrahyperbolic space intermediate representation is statistically significant.

2. On a more fundamental level, I am unconvinced that ultrahyperbolic space is necessary to glean the benefit of an intermediate representation that allows for both positive and negative curvature. For example, I would like to see a comparison against Constant Curvature Graph Convolutional Networks (CC-GCN) [a], which allow for an intermediate representation that is a product of spaces of constant curvature. A fair comparison would be for example, a mixed signature ultrahyperbolic model ($\\mathcal{P}^{(p,q)}_r$) against a CC-GCN model with a product-like intermediate representation, e.g. $H^{p+1} \times S^q$. I would also like to see a discussion or analysis of the benefits of a product model (like CC-GCN) vs. a model that incorporates positive and negative curvatures into a single space, like the ultrahyperbolic GCN. I think this aspect is completely unaddressed as of right now.

Verdict

The paper developed ultrahyperbolic neural networks, together with clearly presented original theory. I think the way the authors handled the manifold, i.e. turning it into a quotient to enable optimization, was clever, and the necessary operations for optimization (e.g. horizontal lift) were introduced in a principled way. Empirically, the benefit of the method is not entirely convincing. Although ultrahyperbolic space seems to give a considerable improvement in the setting of node classification (Table 3), this was not the case for graph classification (Table 5) where standard deviations are high enough that it is unclear to me if the margin of improvement is statistically significant. Moreover, the paper fails to address prior work [a] on GCNs that utilize a product of constant curvature spaces to extract additional structure, as opposed to using a single space with both positive and negative curvature (i.e. ultrahyperbolic space); in particular, an empirical comparison is missing, and a discussion/analysis of differences (in particular with respect to applying these models to graph tasks). Given that this is the case, I must recommend rejection (for the time being) and that the authors improve their paper by addressing this directly; however, I do emphasize that overall, the paper is very promising.

References

[a] Bachmann, G., B'ecigneul, G., & Ganea, O. (2020). Constant Curvature Graph Convolutional Networks. ICML.

### Post-rebuttal Update

Given the thoroughness of the rebuttal and the pertinent strong results demonstrated, I've decided to update my score from a 5 to a 7.

**Time Spent Reviewing:**

6 hours

---

> ### Author Response · Authors · 2021-08-09
> **Comparisons with products of manifold baselines (+ weakness of Table 5)**
>
> Thank your for your positive review. We are glad that you appreciate the elegance of our framework based on tools from principal bundles (i.e. horizontal lift etc.) to learn neural networks that map data to quotient non-Riemannian (pseudo-Riemannian) manifolds of constant nonzero curvature.
>
> In the paper, we proposed to represent data in a space form (i.e. a manifold of constant curvature). We did not compare to products of spherical and hyperbolic manifolds because these constructed product manifolds do not have constant curvature and we could similarly consider products of pseudo-spheres of same dimension to add more complexity (see lines 284-285), which would have made the paper hard to read.
>
> Nonetheless, we understand the concern of the reviewers saying that, since our manifold contains hyperbolic and elliptic parts, it should be compared to products of spherical and hyperbolic manifolds of same topological dimension.
>
> ## We ran the suggested experiments during the rebuttal period: ##
>
> We define $\mathbb{S}^p_{r_1}$ as the $p$-sphere with radius $r_1$ (embedded in a $(p+1)$-dimensional Euclidean space) and $\mathbb{H}^q_{r_2}$ as the $q$-dimensional hyperboloid of "radius" $r_2$ and embedded in a $(q+1)$-dimensional space.
>
> Following the suggested references [E,F], the radii $r_1 > 0$ and $r_2 > 0$ are trained parameters (both initialized at 1) and we define the following distance metrics for the product manifold $\mathbb{S}^p_{r_1} \times \mathbb{H}^q_{r_2}$ (see [E] for details):
>
>
> - The geodesic $\ell_2$ distance: $d((x1, y1), (x2, y2)) = \sqrt{d_1^2 (x1, x2) + d_2^2 (y1, y2)}$
>
> - The $\ell_1$ distance: $d((x1, y1), (x2, y2)) = d_1 (x1, x2) + d_2 (y1, y2)$
>
> - The min distance: $d((x1, y1), (x2, y2)) = \min( d_1 (x1, x2) , d_2 (y1, y2))$
>
> where $d_1 = r_1 \cos^{-1}( \langle x1, x2 \rangle /r_1^2)$ and $d_2 = r_2 \cosh^{-1}(| \langle y1, y2 \rangle_{L}/r_2^2|)$ are the geodesic distances of the $p$-sphere of radius $r_1$ and $q-$hyperboloid of radius $r_2$, respectively.
>
> ### Zachary's karate club dataset ###
>
> We report the scores for Zachary's karate club dataset by using the following evaluation metrics in the same order as Table 1: Rank of first leader (lower is better), Rank of second leader (lower is better), Spearman's $\rho$ for the top 5 nodes (higher is better), Spearman's $\rho$ for the top 10 nodes (higher is better).
>
> These evaluation metrics allows us to quantify how much the chosen distance extracts the hierarchy information in the graph since they quantify how much higher level nodes are closer to the rest of the nodes.
>
> $\bullet$ Product of manifolds $\mathbb{S}^1_{r_1} \times \mathbb{H}^3_{r_2}$:
>
> Scores for $\ell_2$ distance: $1.9 \pm 0.8$, $3.4 \pm 0.9$, $0.47 \pm 0.20$, $0.51 \pm 0.18$
>
> Scores for $\ell_1$ distance: $1.8 \pm 0.5$,
> $3.4 \pm 0.7$, $0.47 \pm 0.25$,  $0.52 \pm 0.13$
>
> Scores for min distance: $3.0 \pm 2.3$, $7.2 \pm 3.4$, $0.23 \pm 0.23$, $0.39 \pm 0.15$
>
>
> $\bullet$ Product of manifolds $\mathbb{S}^2_{r_1} \times \mathbb{H}^2_{r_2}$:
>
>
> Scores for $\ell_2$ distance: $2.0 \pm 0.7$, $3.6 \pm 1.5$, $0.48 \pm 0.24$, $0.50 \pm 0.23$.
>
> Scores for $\ell_1$ distance: $2.2 \pm 0.7$, $3.8 \pm 0.7$, $0.24 \pm 0.29$, $0.48 \pm 0.17$
>
>
> Scores for min distance: $3.6 \pm 2.5$, $8.0 \pm 3.6$, $0.16 \pm 0.30$, $0.48 \pm 0.24$
>
>
> $\bullet$ Product of manifolds $\mathbb{S}^3_{r_1} \times \mathbb{H}^1_{r_2}$:
>
> Scores for $\ell_2$ distance: $1.8 \pm 0.7$, $3.6 \pm 0.9$,$0.31 \pm 0.21$, $0.52 \pm 0.16$.
>
> Scores for $\ell_1$ distance: $1.8 \pm 0.7$, $3.4 \pm 0.8$,$0.48 \pm 0.19$, $0.51 \pm 0.17$.
>
> Scores for min distance: $3.0 \pm 2.3$, $7.8 \pm 3.2$, $0.13 \pm 0.42$, $0.46 \pm 0.22$
>
> All these product manifolds perform better than Riemannian space forms but worse than the quotient manifold $\mathcal{P}^{p,q}_1$.
> It is worth noting that the best performing distance metrics are those that add the spherical and hyperbolic distances and then explicitly enforce both a spherical and hyperbolic structure at the same time when comparing pairs of samples. The fact that they perform worse than the geodesic distance of $\mathcal{P}^{p,q}_r$ indicates that explicitly constructing separate hyperbolic and spherical parts to the manifold by using products of Riemannian manifolds may not be optimal depending on the compared pairs.
>
> Surprisingly, the (min) distance metric that selects some hyperbolic or spherical distance depending on the pair of samples performs much worse. This might be explained by the fact that this kind of distance focuses only on the spherical or hyperbolic part at a time and ignores the other aspect of the product manifold. This is in contrast with our approach that also intrinsically selects an elliptic or hyperbolic type of distance depending on the pair of compared samples (see Equation (7)). However, this selection depends on the (intrinsic) geodesic "distance" of the whole manifold $\mathcal{P}^{p,q}_r$ based on the notion of arc length of pseudo-Riemannian manifolds (see lines 512-517 and 537 in the supplementary material).
>
> This suggests that our parametric model learns where to represent samples on the manifold so that pairs are compared either with hyperbolic or elliptic distance depending on the kind of relationship they have (e.g. higher in the hierarchy or not).
> Experimental results suggest that the fact that $\mathcal{P}^{p,q}_r$ intrinsically contains hyperbolic and elliptic parts in a single space form due to the indefiniteness of the metric tensor allows us to better describe hierarchical relationships between samples when the hierarchical graph contains cycles.
>
>
>
> ### Results in node classification ###
>
> We ran the same kind of experiments as above in the node classification task using the same experimental protocol as in the submission (see Table 2 for comparison). We report here the results for $4$-dimensional manifolds. We report the scores obtained by exploiting the geodesic ($\ell_2$) distance, the $\ell_1$ distance and the min distance in that order:
>
>
>
> $\bullet$ Citeseer:
>
> $\mathbb{S}^1_{r_1} \times \mathbb{H}^3_{r_2}$:
> $46.8 \pm 2.1$,  $43.4 \pm 2.6$,  $40.7 \pm 3.9$
>
> $\mathbb{S}^2_{r_1} \times \mathbb{H}^2_{r_2}$:
> $47.2 \pm 2.1$, $45.9 \pm 1.9$, $44.4  \pm 2.3$.
>
> $\mathbb{S}^3_{r_1} \times \mathbb{H}^1_{r_2}$: $48.1  \pm  2.1$, $47.3  \pm 2.0$, $43.6  \pm 3.2$
>
>
>
> $\bullet$ Cora:
>
> $\mathbb{S}^1_{r_1} \times \mathbb{H}^3_{r_2}$: $57.6  \pm   2.4$, $56.6  \pm 2.9$, $47.5 \pm 2.5$
>
> $\mathbb{S}^2_{r_1} \times \mathbb{H}^2_{r_2}$: $60.5 \pm 3.2$, $60.4 \pm 2.8$,  $55.2 \pm 4.9$
>
> $\mathbb{S}^3_{r_1} \times \mathbb{H}^1_{r_2}$: $60.8 \pm 2.8$, $56.5  \pm 2.4$, $55.2  \pm 2.9$
>
> $\bullet$ Pubmed:
>
> $\mathbb{S}^1_{r_1} \times \mathbb{H}^3_{r_2}$: $71.5 \pm    2.1$, $68.5 \pm 4.8$, $63.0  \pm 1.4$
>
> $\mathbb{S}^2_{r_1} \times \mathbb{H}^2_{r_2}$:
> $71.1  \pm 2.5$,  $70.5  \pm    2.6$, $70.1  \pm 2.1$
>
> $\mathbb{S}^3_{r_1} \times \mathbb{H}^1_{r_2}$:
>  $72.5 \pm 1.8$, $71.9  \pm 2.1$, $68.9\pm2.6$.
>
> Once again, the min distance performs worse than the other distances. The other distances perform slight better than hyperbolic and elliptic distances but are still outperformed by our proposed distances on the Cora and Citeseer datasets.
>
> We also ran similar experiments for 10-dimensional manifolds. The conclusion is similar. We can report the whole list of scores for $10$-dimensional manifolds if requested.
>
> In conclusion, using non-Riemannian manifolds of constant nonzero curvature that intrinsically contain hyperbolic and spherical/elliptic parts allows us to better represent hierarchical graphs with cycles.
>
> ## Comments on weakness of results of Table 5 ##
>
> We agree that our results in Table 5 are not "strong" but they are not weak either. The goal of this experiments was to show that our framework is flexible and can be adapted to most tasks and most models (i.e., standard neural networks or graph convolutional networks).
> We would also like to emphasize that we did not tune the hyperparameters. We only took the publicly available code of Liu et al., replaced their Lorentz manifold class with our quotient manifold and reported the results. Since we believed our scores on the other datasets were strong enough (without tuning hyperparameters), we thought showing a slight improvement in graph classification was a nice result.
>
> ### References ###
>
> [E] Gu et al., "Learning Mixed-Curvature Representations in Product Spaces", ICLR 2019
>
> [F] Bachmann et al., "Constant curvature graph convolutional networks", ICML 2020

---

> > ### Author Response · Authors · 2021-08-28
> > **Message to Reviewer KvL5**
> >
> > Dear Reviewer KvL5,
> >
> > Thanks for your initial review. We believe that we have addressed all your remarks in our response. In particular, we have added additional baselines that combine hyperbolic + spherical manifolds and provided an analysis. We would greatly appreciate if you could update your review with any further comments or any questions, we would be happy to answer them and we will update the draft accordingly. Thanks again for your time!

---

> > ### Comment · Reviewer_KvL5 · 2021-08-29
> > **Response**
> >
> > I appreciate the response to my concerns; in particular, thank you for providing the relevant experiments comparing products of constant curvature spaces with ultrahyperbolic space. I find the results to be strong enough that my concerns, for the most part, are addressed. Please incorporate what you have demonstrated here into the final version of the paper. I've also reviewed the other reviews and the corresponding responses. I've decided to update my score from a 5 to a 7. I believe this paper will be a positive contribution to the conference, granted that the clarifications made (together with relevant results) are included in the final version.

---

### Author Response · Authors · 2021-08-10
**General comment**

We thank the reviewers for their valuable feedback.

### Summary of the paper

We propose an optimization framework to learn parametric models (e.g., neural networks) that map data to a specific family of (non-Riemannian) pseudo-Riemannian manifolds of constant nonzero curvature.
The main motivation is that such manifolds generalize hyperbolic and elliptic geometries (see Section 2 for details) into a single non-Riemannian manifold and can describe relationships specific to both hyperbolic and elliptic geometries (i.e., tree-like and cyclic relationships, respectively).

Learning nonparametric embeddings lying on non-Riemannian manifolds was already proposed in the machine learning literature (e.g., ref [C]).
However, to the best of our knowledge, our paper is the first one that introduces a method to learn a parametric model that maps to a non-Riemannian manifold.
It also elegantly uses pseudo-Riemannian optimization tools that are intrinsic to the considered manifold.

One main difficulty for optimizing on non-Riemannian manifolds is that the negative of the (pseudo-Riemannian) gradient is not a descent direction due to the lack of positive definiteness of the metric tensor.
Moreover, for many non-Riemannian manifolds such as the one considered in [C], there exist pairs of points that cannot be joined by an unbroken geodesic so parallel translation and the logarithm map (that are standard tools to optimize Riemannian neural networks) are difficult to extend.

In this paper, we propose to consider the quotient space $\mathcal{P}_r^{p,q} = \mathcal{S}_r^{p,q} / \pm 1$ where $\mathcal{S}_r^{p,q}$ is the pseudo-sphere considered in [C] (we recall that  $\mathcal{S}_r^{p,q}$ generalizes hyperbolic and spherical geometries).
$\mathcal{P}_r^{p,q} = \mathcal{S}_r^{p,q} / \pm 1$ is defined such that every point $\textbf{x}$ of  $\mathcal{S}_r^{p,q}$ is equivalent to its antipodal point $-\textbf{x}$.
We also explain in Section 2 how $\mathcal{P}_r^{p,q}$ generalizes hyperbolic and elliptic geometries.
This construction ensures that every pair of points of $\mathcal{P}_r^{p,q}$ can be joined by at least one unbroken geodesic.
Due to the quotient nature of the manifold, we propose to use differential geometry tools such as the horizontal lift operator (see Section 3.1) so that every "abstract" tangent vector of $\mathcal{P}_r^{p,q}$ can be equivalently written as a function of a tangent vector of $\mathcal{S}_r^{p,q}$.
Thanks to the proposed bijection, we are able to define all the necessary differential geometry tools in closed form (i.e., geodesic, exponential map, logarithm map, geodesic distance and parallel transport) and work equivalently with $\mathcal{S}_r^{p,q}$.

### Generality and flexibility

Our optimization framework is general (see Section 3) and can be used for any kind of task or model. Since the negative of the pseudo-Riemannian gradient is not a descent direction, we show in our message "Comments about originality" to Reviewer J8on that the chosen search direction is a descent direction that allows to optimize any kind of neural network via standard backpropagation.
In Section 4, we also propose an extension to Graph convolutional networks and show how it can easily generalize hyperbolic approaches. Other types of models could be extended with our approach.

### Simplicity

The simplicity of our approach to optimize this kind of problem was well received by reviewers KvL5 and xWxM, but it seemed too natural for Reviewer v58v and straightforward for Reviewer J8on.
We believe that proposing a simple and elegant approach to optimize parametric models on non-Riemannian manifolds is a nice contribution since the original problem was difficult and never solved before to the best of our knowledge. This shows that our framework was well explained. Moreover, we managed to use only tools that are intrinsic to the considered manifold $\mathcal{P}_r^{p,q}$.

### We included baselines requested by reviewers.

In the paper, we proposed to represent data in a space form (i.e., a manifold of constant curvature). We did not compare to products of spherical and hyperbolic manifolds because these constructed product manifolds do not have constant curvature and we could similarly consider products of pseudo-spheres of same dimension to add more complexity (see lines 284-285), which would have made the paper hard to read.

Nonetheless, we understand the concern of the reviewers saying that, since our manifold contains hyperbolic and elliptic parts, it should be compared to products of spherical and hyperbolic manifolds of same topological dimension.

We then included a comparison to baselines using products of space forms in our message "Comparisons with products of manifold baselines (+ weakness of Table 5)" to Reviewer KvL5. We also included an interpretation of the results.

### Difference with the manifold used in [C]

We gave an explanation of how our manifold is different from the manifold considered in [C] in our message "Quality and other comments" to Reviewer J8on.
In particular, $\mathcal{P}_r^{p,q}$ is a projective space that generalizes elliptic geometry. In elliptic geometry, "elliptic straight lines" and "elliptic cycles" behave very similarly as they pass multiple times through their original point if they are extended indefinitely. Our manifold can then describe cyclic relationships.

We addressed all the other questions individually and we would be happy to give more details (even in the proofs) if requested.




[C] Law and Stam, "Ultrahyperbolic representation learning", NeurIPS 2020.

---

### Decision · Program_Chairs · 2021-09-27

**Decision:**

Accept (Spotlight)

**Comment:**

There is a lot of support for this paper, which proposes a mathematical elegant and convincing method for embedding graphs in non-Euclidean spaces, namely ultrahyperbolic spaces. The experiments - despite some caveats mentioned - provide sufficient evidence for the validity of the approach.